# AdaFocal: Calibration-aware Adaptive Focal Loss

**Arindam Ghosh**
3M Health Info. Systems
Pittsburgh, PA 15217
aghosh4@mmm.com

**Thomas Schaaf**
3M Health Info. Systems
Pittsburgh, PA 15217
tschaaf@mmm.com

**Matt Gormley**
Carnegie Mellon University
Pittsburgh, PA 15213
mgormley@cs.cmu.edu

## Abstract

Much recent work has been devoted to the problem of ensuring that a neural network's confidence scores match the true probability of being correct, i.e. the calibration problem. Of note, it was found that training with focal loss leads to better calibration than cross-entropy while achieving similar level of accuracy [19]. This success stems from focal loss regularizing the entropy of the model's prediction (controlled by the parameter $\gamma$), thereby reining in the model's overconfidence. Further improvement is expected if $\gamma$ is selected independently for each training sample (Sample-Dependent Focal Loss (FLSD-53) [19]). However, FLSD-53 is based on heuristics and does not generalize well. In this paper, we propose a calibration-aware adaptive focal loss called AdaFocal that utilizes the calibration properties of focal (and inverse-focal) loss and adaptively modifies $\gamma_t$ for different groups of samples based on $\gamma_{t-1}$ from the previous step and the knowledge of model's under/over-confidence on the validation set. We evaluate AdaFocal on various image recognition and one NLP task, covering a wide variety of network architectures, to confirm the improvement in calibration while achieving similar levels of accuracy. Additionally, we show that models trained with AdaFocal achieve a significant boost in out-of-distribution detection.

## 1 Introduction

Neural networks have found tremendous success in almost every field including computer vision, natural language processing, and speech recognition. Over time, these networks have grown complex and larger in size to achieve state-of-the-art performance and they continue to evolve in that direction. Along with these advances it has also been well established that these networks suffer from poor calibration [4], i.e. the confidence scores of the predictions do not reflect the real world probabilities of those predictions being true. For example, if the network assigns $0.8$ confidence to a set of predictions, we should expect $80\%$ of those predictions to be correct. However, this is far from reality since modern networks tend to be grossly over-confident. This is of great concern, particularly for mission-critical applications such as autonomous driving or medical diagnosis, wherein the downstream decision making relies not only on the predictions but also on their confidences.

In recent years, there has been a growing interest in developing methods for neural network calibration. These roughly fall into two categories (1) post hoc approaches that perform calibration after training (2) methods that calibrate the model during training. The first category includes methods such as temperature scaling [4], histogram binning [31], isotonic regression [32], Bayesian binning and averaging [22, 21], Dirichlet scaling [10], mix-n-match methods [34], and spline fitting [5]. Methods in the second category focus on designing objective functions that account for calibration during training, such as Maximum Mean Calibration Error (MMCE) [13], Label smoothing [20], and recently focal loss [19]. These aim to inherently calibrate the model during training, yet when combined with post hoc calibration further improvement is often obtained.

36th Conference on Neural Information Processing Systems (NeurIPS 2022).

**Contribution**   Our work belongs to the second category. We first show that while regular focal loss (with fixed $\gamma$) improves the overall calibration by preventing samples from being over-confident, it also leaves other samples under-confident. To address this issue, we propose a modification to the focal loss, while utilizing inverse-focal loss [30, 17], named AdaFocal that adjusts the $\gamma$ for each training sample (or rather a group of samples) separately by taking into account the model's under/over-confidence about a corresponding sample (or group) in the validation set. AdaFocal also adaptively switches from focal to inverse focal loss when focal loss fails to overcome the level of under-confidence. We evaluate our method on four image classification tasks (CIFAR-10, CIFAR-100, Tiny-ImageNet and ImageNet) and one text classification task (20 Newsgroup) using various model architectures. We find that AdaFocal substantially outperforms regular focal loss and other state-of-the-art calibration-during-training techniques in the literature. Models calibrated by AdaFocal benefit more from post hoc calibration techniques to further reduce the calibration error. Finally, we study the performance of AdaFocal on an out-of-distribution detection task and find a substantial improvement in performance.

## 2   Problem Setup and Definitions

For a classification problem with training data $\{(\mathbf{x}_n, y_{\text{true},n})\}$, where $\mathbf{x}_n$ is the input and $y_{\text{true},n} \in \mathcal{Y} = \{1, 2, \ldots, K\}$ is the ground-truth, we train a model $f$ that outputs a probability vector $\hat{\mathbf{p}}$ over the $K$ classes. We further assume access to a validation set for hyper-parameter tuning and a test set for evaluation. For example, $f_\theta(\cdot)$ can be a neural network with learnable parameters $\theta$, $\mathbf{x}$ is an image, and $\hat{\mathbf{p}}$ is the output of a *softmax layer* whose $k^{\text{th}}$ element $\hat{p}_k$ is the probability score for class $k$. We refer to $\hat{y} = \arg\max_{k \in \mathcal{Y}} \hat{p}_k$ as the network's prediction and the probability score $\hat{p}_{\hat{y}}$ as the confidence. Then, a model is said to be perfectly calibrated if the confidence score $\hat{p}_{\hat{y}}$ matches the probability of the model classifying $\mathbf{x}$ correctly i.e. $\mathbb{P}(\hat{y} = y_{\text{true}} \mid \hat{p}_{\hat{y}} = p) = p, \ \forall p \in [0, 1]$ [4]. Continuing our example, if the network assigns an average confidence score of $0.8$ to a set of predictions then we should expect $80\%$ of those to be correct.

To quantify calibration, we use *Calibration Error* as $\mathcal{E} = \hat{p}_{\hat{y}} - \mathbb{P}(\hat{y} = y_{\text{true}} \mid \hat{p}_{\hat{y}})$ and the *Expected Calibration Error* as $\mathbb{E}_{\hat{p}_{\hat{y}}}[\mathcal{E}] = \mathbb{E}_{\hat{p}_{\hat{y}}}[\,|\hat{p}_{\hat{y}} - \mathbb{P}(\hat{y} = y_{\text{true}} \mid \hat{p}_{\hat{y}})|\,]$ [4]. Since the true calibration error cannot be computed empirically with a finite sized dataset, the following approximations are generally used in the literature. For a dataset $\{(\mathbf{x}_n, y_{\text{true},n})\}_{n=1}^N$, (1) $\text{ECE}_{\text{EW}} = \sum_{i=1}^M \frac{|B_i|}{N}|C_i - A_i|$ [4], where $B_i$ is equal-width (EW) bin that contains all examples $j$ with $\hat{p}_{\hat{y},j}$ in the range $[\frac{i}{M}, \frac{i+1}{M})$, $C_i = \frac{1}{|B_i|}\sum_{j \in B_i} \hat{p}_{\hat{y},j}$ is the average confidence and $A_i = \frac{1}{|B_i|}\sum_{j \in B_i} \mathbb{1}(\hat{y}_j = y_{\text{true},j})$ is the bin accuracy. Note that $E_i = C_i - A_i$ is the empirical approximation of the calibration error $\mathcal{E}$, (2) $\text{ECE}_{\text{EM}} = \sum_{i=1}^M \frac{|B_i|}{N}\,|C_i - A_i|$ [24], where $\forall i, j \ |B_i| = |B_j|$ are equal-mass (EM) bins. Furthermore, as ECE has been shown to be a biased estimate of the true calibration error [29], we additionally use $\text{ECE}_{\text{DEBIAS}}$ [11] and $\text{ECE}_{\text{SWEEP}}$ [26] to corroborate the results in the paper.

## 3   Calibration Properties of Focal Loss

Focal loss [16] $\mathcal{L}_{FL}(p) = -(1 - p)^\gamma \log p$ was originally proposed to improve the accuracy of classifiers by focusing on hard examples and down-weighting well classified examples. Recently, it was shown that focal loss can also be used to improve calibration [19]. This is because, based on the relation $\mathcal{L}_{FL} \geq KL(q||\hat{\mathbf{p}}) - \gamma\mathbb{H}(\hat{\mathbf{p}})$ ($q$ is the one-hot target vector), focal loss while minimising the KL divergence objective also increases the entropy of the prediction $\hat{\mathbf{p}}$. This prevents the network from being overly confident on wrong predictions thereby improving calibration.

However, as we show next, focal loss with fixed $\gamma$ falls short of achieving the best calibration. In Figure 1, we plot the calibration behaviour of ResNet50 in different probability regions (bins) when trained on CIFAR-10 with different focal losses. The $i$th bin's calibration error $E_{val,i} = C_{val,i} - A_{val,i}$ is computed on the validation set using 15 equal-mass binning. The figure plots a lower (bin-0), middle (bin-7) and higher bin (bin-14) (rest of the bins and their bin boundaries are shown in Appendix B). We see that, although focal loss $\gamma = 4$ achieves the overall lowest $\text{ECE}_{\text{EM}}$, there's no single $\gamma$ that performs the best across all the bins. For example, in bin-0, $\gamma = 4, 5$ achieves better calibration whereas $\gamma = 0, 3$ are over-confident. On the other hand, in bin-7 $\gamma = 3$ seems to be better calibrated whereas $\gamma = 4, 5$ are under-confident and $\gamma = 0$ is over-confident.

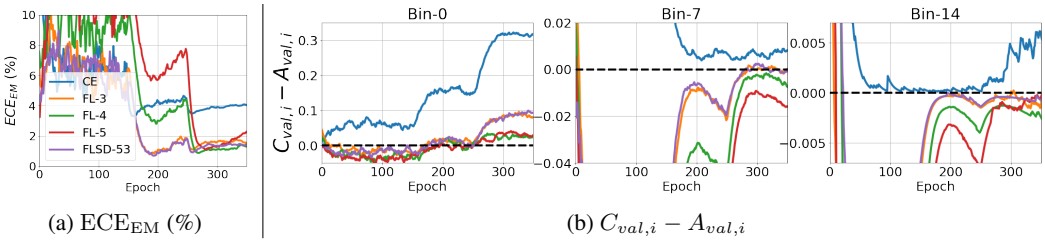

(a) $\text{ECE}_{\text{EM}}$ (%)             (b) $C_{val,i} - A_{val,i}$

Figure 1: Calibration behaviour of ResNet-50 trained on CIFAR-10 with cross entropy (CE), focal loss $\gamma = 3, 4, 5$ (FL-3/4/5) and FLSD-53. These are computed using 15 equal-mass binning on the validation set. (a) $\text{ECE}_{\text{EM}}$, and (b) Calibration error $C_{val,i} - A_{val,i}$ for a lower (bin-0), middle (bin-7), and upper (bin-14) bin. The black horizontal lines in (b) represent $E_{val,i} = 0$. These show that although $\gamma = 4$ achieves the overall lowest calibration error, the best performing $\gamma$ is different for different bins.

This shows that using different $\gamma$s for different bins can bring further improvement. Such an attempt called the Sample-Dependent Focal Loss (FLSD-53) is presented in [19] that assigns $\gamma = 5$ if the training sample's true class posterior $\hat{p}_{y_{\text{true}}} \in [0, 0.2)$ and $\gamma = 3$ if $\hat{p}_{y_{\text{true}}} \in [0.2, 1]$. However, from Figure 1(b), FLSD-53 is also not the best calibrated method across all bins. It is a strategy based on fixed heuristics of choosing higher $\gamma$ for smaller values of $\hat{p}_{y_{\text{true}}}$ and relatively lower $\gamma$ for higher values of $\hat{p}_{y_{\text{true}}}$.

This clearly motivates the design of a strategy that can assign appropriate $\gamma_i$s for each bin-$i$ based on the magnitude and sign of $E_{val,i}$. To design such a strategy one, however, faces two challenges:

1. How do we find some correspondence between the confidence of training samples (which we can manipulate during training using the parameter $\gamma$) and the confidence of the validation/test samples (which are our actual target but we do not have direct control over them)? In other words, to indirectly control the confidence of a group of validation samples, how do we know which particular group of training samples' confidence to be manipulated?

2. Given that some correspondence is established, how do we arrive at the appropriate values of $\gamma$ that will lead to the best calibration?

We try to answer the first question in the next section and answer to the second question leads to the main contribution of the paper: AdaFocal.

Additionally, alongside focal loss, we make use of the inverse-focal loss [30, 17] for cases where regular focal loss fails to overcome under-confidence. See, for example, ResNet-50 trained on ImageNet in section 6 and Fig. 5 where even cross entropy ($\gamma = 0$) can not reach the desired level of confidence. Inverse-focal loss, plotted in Fig. 3(a), and given by

$$\mathcal{L}_{InvFL}(p) = -(1 + p)^{\gamma} \log p, \tag{1}$$

serves the opposite purpose of focal loss. While focal loss reduces the over-confidence of the network, inverse-focal loss helps recover from under-confidence by providing larger gradients to the samples with higher confidences (easy samples), thereby pushing their scores even further.

## 4   Correspondence between Confidence of Training and Validation Samples

One way to check for any correspondences is to simply group the validation samples into $M$ equal-mass bins (henceforth called validation-bins) and compare the confidence with the training samples that fall into the respective validation-bin boundaries. Before proceeding further, we first clarify a few notations of interest.

**Quantities of interest**    For binning the validation samples, we look at the confidence of the top predicted class $\hat{y}$ denoted by $\hat{p}_{val,top}$ (bin average: $C_{val,top}$). For training samples, on the other hand, instead of the confidence of the top predicted class $\hat{y}$ denoted by $\hat{p}_{train,top}$ (bin average: $C_{train,top}$), we will focus on the confidence of the true class $y_{\text{true}}$ denoted by $\hat{p}_{train,true}$ (average: $C_{train,true}$) because during training we only care about $\hat{p}_{train,true}$ that gets manipulated by some loss function

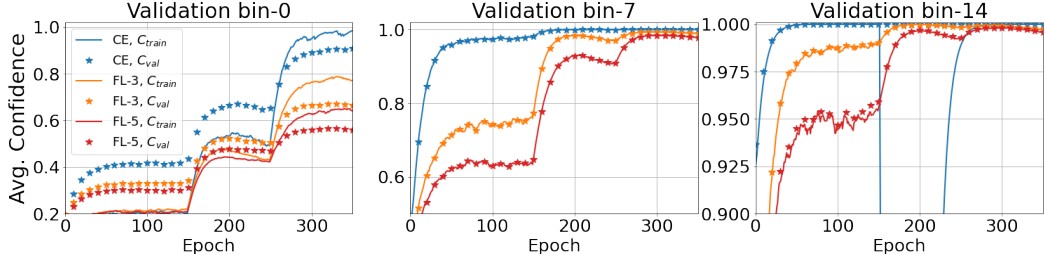

Figure 2: Correspondence between confidence of training ($C_{train}$) and validation samples ($C_{val}$) for ResNet-50 trained on CIFAR-10 with focal loss $\gamma = 0$ (CE), $3, 5$. The binning involves 15 equal-mass bins where training samples are grouped using validation-bin boundaries. A lower (bin-0), middle (bin-7) and upper (bin-14) bins are shown here (with the rest shown in Fig. 10 in Appendix C).

(Figure 8 in Appendix C compares $C_{train,true}$ and $C_{train,top}$ to show that as the training accuracy approaches $100\%$, the top predicted class and the true class become the same). For brevity, we will henceforth refer to $C_{train} \equiv C_{train,true}$ and $C_{val} \equiv C_{val,top}$.

In Figure 2, we compare $C_{train,i}$ in validation-bin-$i$ [1] with $C_{val,i}$ and find that there is indeed a good correspondence between the two quantities. For example in Figure 2, as $\gamma$ increases from 0 (CE), to 3 to 5, the solid-line $C_{train,i}$ gets lower, and the same behaviour is observed for the starred-line $C_{val,i}$. For more evidence refer to Fig. 11, 12, and 13 in Appendix C where similar behaviour is observed for ResNet-50 and WideResNet on CIFAR-100, and ResNet-50 on TinyImageNet, respectively.

We also look at the case when training samples and validation samples are grouped independently into their respective training-bins and validation-bins. Figure 9 in Appendix C compares $C_{train,i}$ in training-bin-$i$ with $C_{val,i}$ in validation-bin-$i$. Again, we observe a similar correspondence. Note here that, since the binning is independent, the boundaries of training-bin-$i$ will not be exactly the same as that of validation-bin-$i$, but, as shown in Figure 9, they are very close to each other.

These observations, therefore, are very encouraging as now we have a way to indirectly control $C_{val,i}$ by manipulating $C_{train,i}$, i.e. we can expect (even if loosely) that if we increase/decrease the confidence of a group of training samples in some lower (or middle, or higher) probability region then the same will get reflected on the validation samples in a similar lower (or middle, or higher) probability region. From a calibration point of view, our strategy going forward would be to exploit this correspondence to push $C_{train,i}$ (which we have control over during training) closer to $A_{val,i}$ (the validation set accuracy in validation-bin-$i$) so that $C_{val,i}$ also gets closer to $A_{val,i}$, and, therefore, achieve a very low calibration error $C_{val,i} - A_{val,i}$. For simplifying the design of AdaFocal, we will employ the first method of common binning i.e. using validation-bins to bin the training samples.

## 5 Proposed Method

Let's denote the $n$th training sample's true class posterior $\hat{p}_{y_{true}}$ by $p_n$ and $p_n$ falls into validation-bin $b$. Our goal then is to keep $p_n$ (or its averaged equivalent $C_{train,b}$) closer to $A_{val,b}$ so that the same is reflected on $C_{val,b}$. For manipulating $p_n$, we will utilize the regularization effect of focal loss's parameter $\gamma$. At this point, one can choose to update the $\gamma$ of validation-bin-$b$ denoted by $\gamma_b$ either based on (1) how far $p_n$ is from $A_{val,b}$ i.e. $\gamma = g(p_n - A_{val,b})$ or (2) how far $C_{val,b}$ is from $A_{val,b}$ i.e. $\gamma = g(C_{val,b} - A_{val,b})$. Such a $\gamma$-update-rule should ensure that whenever the model is over-confident, i.e. $p_n > A_{val,b}$ (or $C_{val,b} > A_{val,b}$), $\gamma$ is increased so that the gradients get smaller to prevent $p_n$ from increasing further. On the other hand, when $p_n < A_{val,b}$ (or $C_{val,b} < A_{val,b}$), i.e. the model is under-confident, we decrease $\gamma$ so as to get larger gradients that in turn will increase $p_n$ [2].

---

[1] It may happen that no training sample falls into a particular validation-bin. In that case, $C_{train,i}$ is shown to drop to zero, for example in bin-14 in Figure 2.

[2] Note that for focal loss increasing $\gamma$ does not always lead to smaller gradients. This mostly holds true in the region $p_n$ approximately $> 0.2$ (see Figure 3(a) in [19]). However, in practice and as shown by the training-bin boundaries of bin-0 and bin-1 in Figure in Figure 9 Appendix C, we find majority of the training samples to lie above 0.2 during the majority of the training, and therefore, for the experiments in this paper, we simply stick to the rule of increasing $\gamma$ to decrease gradients and stop $p_n$ from increasing and vice versa.

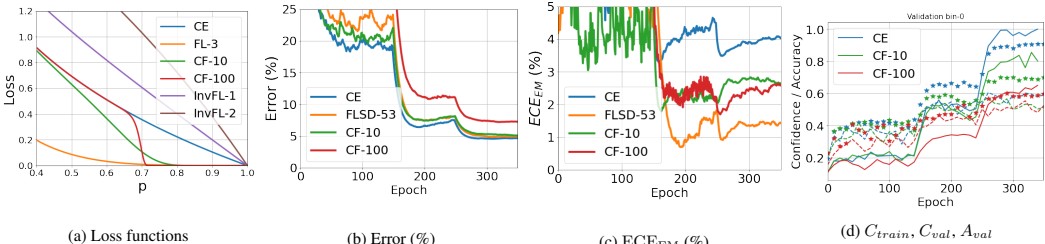

(a) Loss functions      (b) Error (%)      (c) ECE$_{EM}$ (%)      (d) $C_{train}, C_{val}, A_{val}$

Figure 3: (a) plots different loss functions: Cross Entropy (CE), Focal loss (FL-$\gamma$), Inverse-Focal loss (InvFL-$\gamma$), CalFocal (CF-$\lambda$) with $A_{val,b} = 0.8$. Subfigures (b) and (c) plot error and ECE$_{EM}$, respectively, for ResNet-50 trained on CIFAR-10 with CalFocal. (d) compares $C_{train}$ (solid lines), $C_{val}$ (starred lines) and $A_{val}$ (dashed lines) in validation bin-0 to show that when CalFocal brings $C_{train}$ closer to $A_{val}$, $C_{val}$ also approaches $A_{val}$.

To test this strategy, we first design a calibration-aware $\gamma$-update method (called CalFocal), which with some additional modifications will lead to the final AdaFocal algorithm.

## 5.1 Calibration-aware Focal Loss (CalFocal)

**Case 1:** $\gamma = g(p_n - A_{val,b})$    Treating $A_{val,b}$ as the point that we want $p_n$ to not deviate from, we make the focal loss parameter $\gamma$ a function of $p_n - A_{val,b}$ to get

$$\mathcal{L}_{CalFocal}(p_n) = -(1-p_n)^{\gamma_n} \log p_n, \quad \text{where } \gamma_n = \exp(\lambda(p_n - A_{val,b})), \quad (2)$$

and $b$ is the validation-bin in which $p_n$ falls. The hyper-parameter $\lambda$ is the scaling factor which combined with the exponential function helps to quickly ramp up/down $\gamma$. The exponential function adheres to the $\gamma$-*update rule* mentioned earlier. Figure 3(a) plots $\mathcal{L}_{CalFocal}$ vs. $p_n$ for $A_{val,b} = 0.8$. We see that based on the strength of $\lambda$, the loss drastically drops near $p_n = 0.8$ and thereafter remains close to zero. This shows that $\mathcal{L}_{CalFocal}$ aims is to first push $p$ towards $0.8$ and then slow its growth towards overconfidence. Next, in Figure 3(c), we find that CalFocal with $\lambda = 10, 100$ is able to reduce the calibration error (ECE$_{EM}$) but it is still far from FLSD-53's performance. Also note in Figure 3(b) that too high $\lambda$ (=100) affects the accuracy of model. Most interesting is Fig. 3(d) which compares $C_{train,i}$ with $C_{val,i}$ (and also $A_{val,i}$) for bin-0, where we find evidence that the strategy of bringing $p_n$ or $C_{train,i}$ (solid lines) closer to $A_{val,i}$ (dashed lines) results in $C_{val,i}$ (starred lines) getting closer to $A_{val,i}$ as well, thus reducing the calibration error $E_{val,i} = C_{val,i} - A_{val,i}$.

**Case 2:** $\gamma = g(C_{val,b} - A_{val,b})$    Note that Eq. 2 assigns a different $\gamma_n$ for each training sample $p_n$. To reduce computation and avoid keeping track of $\gamma_n$ for each training sample, we can assign one $\gamma_b$ to each validation-bin-$b$ by making it a function of $C_{val,b} - A_{val,b}$ (instead of $p_n - A_{val,b}$). Then all $p_n$ that fall into the validation-bin-$b$ are assigned $\gamma_b$ and the loss function is modified to

$$\mathcal{L}_{CalFocal}(p_n) = -(1-p_n)^{\gamma_b} \log p_n, \quad \text{where } \gamma_b = \exp(\lambda(C_{val,b} - A_{val,b})) \quad (3)$$

$b$ is the validation-bin in which $p_n$ falls. The performance of this strategy, as shown in Appendix P, is very similar (or slightly better than) Eq. 2. Further, it makes more sense to update $\gamma$ based on how far $C_{val,b}$ is from $A_{val,b}$ instead of how far $p_n$ is from $A_{val,b}$ because, as shown in Figure 3(d) bin-0, one may find $C_{val,b}$ (starred lines) quite closer to $A_{val,b}$ (dashed lines) even when $p_n$ or $C_{train}$ (solid lines) is still far from $A_{val,b}$. At this point where $C_{val,b} = A_{val,b}$, we should stop updating $\gamma$ further, even if $p_n - A_{val,b} \neq 0$, as we have reached our goal of making $E_{val,b} = C_{val,b} - A_{val,b} = 0$. Therefore, for AdaFocal we will use Eq. 3 as the base for AdaFocal loss function.

**Limitations of CalFocal**    (1) Let's say at some step of training, a high $\gamma_b$ over some epochs reduces the error $C_{val,b} - A_{val,b}$. Then, it is desirable to continue training with the same high $\gamma_b$. However, note CalFocal's update rule in Eq. 3 which will reduce $\gamma \to 1$ as the $C_{val,b} - A_{val,b} \to 0$. (2) Let's say, at some point $C_{val,b} - A_{val,b}$ is quite high. This will set $\gamma_b$ to some high value depending on the hyper-parameter $\lambda$. Assuming this $\gamma_b$ is still not high enough to bring down the confidence, we would want a way to further increase $\gamma_b$. However, CalFocal is incapable of doing so as it will continue to hold at $\gamma_b = \exp(\lambda(C_{val,b} - A_{val,b}))$.

## 5.2 Calibration-aware Adaptive Focal Loss (AdaFocal)

We propose to address the above limitations by making $\gamma_{t,b}$ depend on $\gamma_{t-1,b}$ from previous time step

$$\gamma_{t,b} = \gamma_{t-1,b} * \exp(\lambda(C_{val,b} - A_{val,b})). \tag{4}$$

This update rule address the limitations of CalFocal in the following way: let's say at some point we observe over-confidence i.e. $E_{val,b} = C_{val,b} - A_{val,b} > 0$. Then, in the next step $\gamma_b$ will be increased. In the subsequent steps, it will continue to increase unless the calibration error $E_{val,b}$ starts decreasing (this additional increase in $\gamma$ was not possible with CalFocal). At this point, if we find $E_{val,b}$ to start decreasing, that would reduce the increase in $\gamma_b$ over the next epochs and $\gamma_b$ will ultimately settle down to a value when $E_{val,b} = 0$ (CalFocal at $E_{val,b} = 0$ will cause $\gamma$ to go down to 1). Next, if this current value of $\gamma_b$ starts causing under-confidence i.e. $C_{val,b} - A_{val,b} < 0$, then the update rule will kick in to reduce $\gamma$ thus allowing $C_{val,b}$ to be increased back to $A_{val,b}$. This oscillating behaviour of AdaFocal around the desired point of $C_{val,b} = A_{val,b}$ is its main strength in reducing the calibration error in every bin.

Next, to deal with cases where even cross entropy suffers from under-confidence, we switch to inverse-focal loss which can further increase the confidence of the predictions. For the switch between focal and inverse-focal loss, we simply set a threshold $S_{th}$ below which if gamma falls, we switch to the other loss function. Note here that, for notational purpose, we will denote the inverse-focal loss by a negative $\gamma$ i.e. $\gamma_{t,b} > 0$ means focal loss with parameter $\gamma_{t,b}$ whereas $\gamma_{t,b} < 0$ implies inverse-focal loss with parameter $|\gamma_{t,b}|$. The complete gamma-update rule and the switching criteria is given in Algorithm 1. If not stated explicitly, we use $S_{th} = 0.2$ for all AdaFocal experiments. For reference, we also plot results with $S_{th} = 0.5$ for ImageNet, ResNet-50 in Fig. 4 (d) and Fig. 5(b).

Finally, note the unbounded exponential update in Eq. 4 which is an undesirable property. This may easily cause $\gamma_t$ to explode as when expanded $\gamma_t = \gamma_{t-1} \exp(E_{val,t}) = \gamma_0 \exp(E_{val,0} + E_{val,1} + ... + E_{val,t-1} + E_{val,t})$, and if $E_{val,t} > 0$ for quite a few number of epochs, $\gamma_t$ will become so large that even if $E_{val,t} < 0$ in the subsequent epochs, it may not reduce to a desired level. We remedy this by constraining $\gamma_t$ to an upper bound $\gamma_{\max}$ when $\gamma > 0$ (focal loss) and lower bound of $\gamma_{\min}$ when $\gamma < 0$ (inverse-focal loss). Therefore, the final AdaFocal loss is given by

$$\mathcal{L}_{AdaFocal}(p_n, t) = \begin{cases} -(1-p_n)^{\gamma_{t,b}} \log p_n, & \text{if } \gamma_{t,b} \geq 0 \\ -(1+p_n)^{|\gamma_{t,b}|} \log p_n, & \text{if } \gamma_{t,b} < 0, \end{cases} \tag{5}$$

and the complete algorithm along with the gamma-update rules is given in Algorithm 1. A discussion on the selection of hyper-parameters is presented in the next section.

## 6 Experiments

We evaluate the performance of our proposed method on image and text classification tasks. For image classification, we use CIFAR-10, CIFAR-100 [9], Tiny-ImageNet [2], and ImageNet [27] to analyze the calibration of ResNet50, ResNet-100 [6], Wide-ResNet-26-10 [33], and DenseNet-121 [8] models. For text classification, we use the 20 Newsgroup dataset [14] and train a global-pooling CNN [15] and fine-tune a pre-trained BERT model [3]. More details about the datasets, models and experimental configurations are given in Appendix D. In the main paper, we report results using only $\text{ECE}_{\text{EM}}$, whereas other ECE metrics are reported in Appendix.

**Baseline** Among calibration-during-training methods we use MMCE [13], Brier loss [1], Label smoothing (LS-0.05) [20] and sample-dependent focal loss FLSD-53 as baselines. For post hoc calibration, we report the effect of temperature scaling, ensemble temperature scaling (ETS) [34] and spline fitting [5] on top of these methods. For temperature scaling, we select the optimal temperature $\in (0, 10]$ (step size 0.1) that gives the lowest $\text{ECE}_{\text{EM}}$ on the validation set.

**Results** In Figure 4, we compare AdaFocal against cross entropy (CE) and FLSD-53, for ResNet-50 trained on various small to large-scale image datasets. Among various focal losses, we chose FLSD-53 as our baseline because it was shown to be consistently better than MMCE, Brier Loss and Label smoothing [19] across many datasets-model pairs. The figure plots the test set error and $\text{ECE}_{\text{EM}}$. In Figure 5, for ResNet-50 on CIFAR-10 and ImageNet, we plot (1) the calibration statistics $E_{val} = C_{val} - A_{val}$ used by AdaFocal during training and (2) the dynamics of $\gamma_t$ for a few bins in lower, middle, and higher probability regions.

---

**Algorithm 1:** AdaFocal

---

1 **Input**: $D_{train} = \{\mathbf{x}_n, y_{true}\}_{n=1}^{N_{train}}$, $D_{val} = \{\mathbf{x}_n, y_{true}\}_{n=1}^{N_{val}}$;

2 **Bin initialization**: **for** $i = 1$ **to** $M$ **do**

3     $B_{t=0,i} = \left(\frac{i-1}{M}, \frac{i}{M}\right]$,    $\gamma_{t=0,i} = 1$    // Initialize validation-bins to equal-width with gamma set to 1;

4 **Training**: **for** $t = 0$ **to** $T$ **do**

5     $Loss_t = 0$;

6     **for** $n = 1$ **to** $N_{train}$ **do**

7        $p_n = f_{\theta_t}(\mathbf{x}_n)$    // Denoting $p_{y_{true,n}}$ by $p_n$;

8        $b = $ get_bin_index$(p_n, \{B_{t,i}\})$    // Bin in which $p_n$ lies;

9        **if** $\gamma_{t,b} \geq 0$ **then**

10          $Loss_t \mathrel{+}= -(1 - p_n)^{\gamma_{t,b}} \log p_n$    // Focal loss;

11        **else if** $\gamma_{t,b} < 0$ **then**

12          $Loss_t \mathrel{+}= -(1 + p_n)^{|\gamma_{t,b}|} \log p_n$    // Inverse-focal loss;

13     $\theta_{t+1} = $ gradient_update$(\theta_t, Loss_t)$;

14     $\gamma$**-update step**: **for** $i = 1$ **to** $M$ **do**

15        Re-compute $B_{t+1,i}$, $C_{val,t+1,i}$ and $A_{val,t+1,i}$ using the updated model $f_{\theta_{t+1}}$ and $D_{val}$;

16        **if** $\gamma_{t,i} \geq 0$ **then**

17          $\gamma_{t+1,i} = \min\left\{\gamma_{\max}, \gamma_{t,i} * e^{\lambda(C_{val,t+1,i} - A_{val,t+1,i})}\right\}$    // Focal loss $\gamma$-update;

18          **if** $|\gamma_{t+1,i}| < S_{th}$ **then**

19            $\gamma_{t+1,i} = -S_{th}$    // Switch to inverse-focal loss;

20        **else if** $\gamma_{t,i} < 0$ **then**

21          $\gamma_{t+1,i} = \max\left\{\gamma_{\min}, \gamma_{t,i} * e^{-\lambda(C_{val,t+1,i} - A_{val,t+1,i})}\right\}$    // Inverse-focal $\gamma$-update;

22          **if** $|\gamma_{t+1,i}| < S_{th}$ **then**

23            $\gamma_{t+1,i} = S_{th}$    // Switch to focal loss;

---

First, we observe that for CIFAR-10, CIFAR-100 and Tiny-ImageNet, FLDS-53 is much better calibrated than CE. This is because, as shown in Fig. 5(a) for ResNet-50, CIFAR-10, CE is over-confident compared to FLSD-53 in almost every bin. For ImageNet, however, the behaviour is reversed: FLSD-53 is poorly calibrated than CE. This, as shown in Figure 5(b), is due to the use of high values of $\gamma$ ($= 5, 3$) by FLSD-53 which makes the model largely under-confident in each bin, leading to a overall high calibration error. AdaFocal, on the other hand, maintains a well calibrated model throughout the training for all cases.

Next, for ResNet-50 trained on CIFAR-10, we find $\gamma_t$ to be closer to 1 for higher bins and closer to 20 for lower bins. These $\gamma_t$s result in better calibration than $\gamma_t = 5, 3$ of FLSD-53. For ImageNet, on the other hand, we find, except bin-14, AdaFocal switches to inverse-focal loss at some point in the training. This makes sense because for ImageNet even cross entropy ($\gamma_t = 0$) suffers from under-confidence, therefore, AdaFocal, starting from $\gamma_t = 1$, first approaches cross entropy ($\gamma = 0$) to ultimately switch to inverse-focal loss ($\gamma_t < 0$) at $S_{th} = 0.2$. Since, in retrospect, we already know that switching to inverse focal loss is beneficial for ImageNet, switching early at $S_{th} = 0.5$ helps to reach the same level of calibration early. Overall, these experiments confirm that during training AdaFocal being aware of the network's current under/over-confidence is able to guide the $\gamma_t$s to values that maintain a well calibrated model at all times. Also note that for an unseen dataset-model pair there's no way to know apriori which $\gamma$ will perform better but AdaFocal will automatically find these appropriate values thus avoiding an expensive and extensive hyper-parameter search.

Comparison of $\mathrm{ECE_{EM}}$ and test set error on for rest of the experiments are shown in Table 1 and 2 respectively. From Table 1, we observe that prior to temperature scaling, AdaFocal outperforms the other methods in 14 out of 15 cases. Post-temperature scaling, AdaFocal achieves the lowest calibration error in 12 out of the 15 cases. Further, observe that the optimal temperatures are mostly close to 1 indicating that AdaFocal produces inherently calibrated models from training itself. Effects of other post hoc calibration methods (ETS and spline fitting) are shown in Appendix E. Again, we observe that pre-calibrated models from AdaFocal benefit from post processing to further lower the overall calibration error. Besides $\mathrm{ECE_{EM}}$, the consistency of the results across other calibration

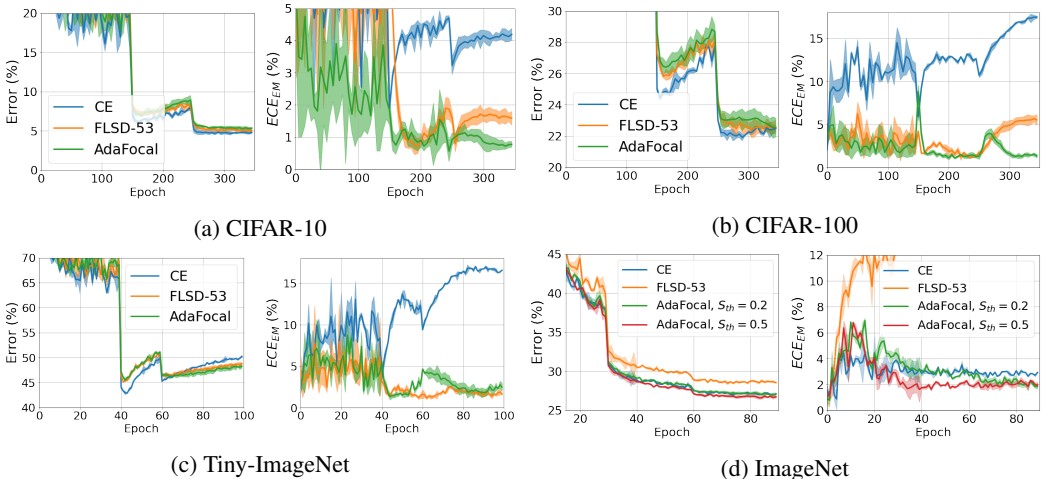

(a) CIFAR-10

(b) CIFAR-100

(c) Tiny-ImageNet

(d) ImageNet

Figure 4: ResNet-50 trained with cross entropy (CE), FLSD-53 and AdaFocal. In each subfigure, **left**: Error (%), **right**: $ECE_{EM}$ (%) on the test set are plotted for mean and standard deviation over 5 runs. We observe that throughout the training AdaFocal maintains a low calibration error while achieving similar accuracy.

| Dataset | Model | Pre Temperature scaling | | | | | | Post Temperature scaling | | | | | |
|---|---|---|---|---|---|---|---|---|---|---|---|---|---|
| | | CE | Brier | MMCE | LS | FLSD-53 | AdaFocal | CE | Brier | MMCE | LS | FLSD-53 | AdaFocal |
| CIFAR-10 | ResNet50 | 4.24 | 1.78 | 4.52 | 3.86 | 1.63 | **0.66** | 2.11(2.52) | 1.24(1.11) | 2.12(2.65) | 2.97(0.92) | 1.42(1.08) | **0.44(1.06)** |
| | ResNet110 | 4.39 | 2.63 | 5.16 | 4.44 | 1.90 | **0.71** | 2.27(2.74) | 1.75(1.21) | 2.53(2.83) | 4.44(1.00) | 1.25(1.20) | **0.73(1.02)** |
| | WideResNet | 3.42 | 1.72 | 3.31 | 4.26 | 1.82 | **0.64** | 1.87(2.16) | 1.72(1.00) | 1.6(2.22) | 2.44(0.81) | 1.57(0.94) | **0.44(1.06)** |
| | DenseNet121 | 4.26 | 2.09 | 5.05 | 4.40 | 1.40 | **0.62** | 2.21(2.33) | 2.09(1.00) | 2.26(2.52) | 3.31(0.94) | 1.40(1.00) | **0.59(1.02)** |
| CIFAR-100 | ResNet50 | 17.17 | 6.57 | 15.28 | 7.86 | 5.64 | **1.36** | 3.71(2.16) | 3.66(1.13) | 2.32(1.80) | 4.10(1.13) | 2.97(1.17) | **1.36(1.00)** |
| | ResNet110 | 19.44 | 7.70 | 19.11 | 11.18 | 7.08 | **1.40** | 6.11(2.28) | 4.55(1.18) | 4.88(2.32) | 8.58(1.09) | 3.85(1.20) | **1.40(1.00)** |
| | WideResNet | 14.83 | 4.27 | 13.12 | 5.10 | 2.25 | **1.95** | 3.23(2.12) | 2.85(1.08) | 4.23(1.91) | 5.10(1.00) | 2.25(1.00) | **1.95(1.00)** |
| | DenseNet121 | 19.82 | 5.14 | 19.16 | 12.81 | 2.58 | **1.73** | 3.62(2.27) | 2.58(1.09) | 3.11(2.13) | 8.95(1.19) | 1.80(1.10) | **1.73(1.00)** |
| TinyImageNet | ResNet50 | 7.81 | 3.42 | 8.49 | 9.12 | 2.86 | **2.61** | 3.73(1.45) | 2.98(0.93) | 4.25(1.36) | 4.66(0.78) | 2.48(1.05) | **2.29(0.96)** |
| | ResNet110 | 8.11 | 3.74 | 7.40 | 9.36 | 1.88 | **1.85** | 1.93(1.20) | 2.83(0.91) | 1.95(1.20) | 4.51(0.83) | 1.88(1.00) | **1.85(1.00)** |
| ImageNet | ResNet50 | 2.93 | 3.91 | 9.30 | 10.05 | 16.77 | **1.87** | **1.50(0.88)** | 3.59(0.92) | 4.22(1.34) | 4.53(0.82) | 2.62(0.74) | 1.87(1.00) |
| | ResNet110 | 1.28 | 3.98 | 1.83 | 4.02 | 18.66 | **1.17** | 1.28(1.00) | 2.87(0.90) | 1.83(1.00) | 2.76(0.90) | 2.51(0.70) | **1.17(1.00)** |
| | DenseNet121 | 1.82 | 2.94 | **1.22** | 5.30 | 19.19 | 1.50 | 1.82(1.00) | 2.21(0.90) | **1.22(1.00)** | 1.42(0.90) | 2.24(0.70) | 1.50(1.00) |
| 20Newsgroup | CNN | 18.57 | 13.52 | 15.23 | 4.36 | 8.86 | **2.62** | 4.08(3.78) | 3.13(2.33) | 6.45(2.21) | 2.62(1.12) | **2.13(1.58)** | 2.46(1.10) |
| | BERT | 8.47 | 5.91 | 8.30 | 6.01 | 8.63 | **3.96** | 4.46(1.44) | 4.40(1.24) | 4.60(1.46) | 5.69(1.14) | 3.91(0.80) | **3.73(1.04)** |

Table 1: Test $ECE_{EM}$ (%) averaged over 5 runs. Bold marks the lowest in pre and post temperature scaling groups separately. Optimal temperature, given in brackets, is cross-validated on $ECE_{EM}$.

metrics is shown through $ECE_{DEBIAS}$, $ECE_{SWEEP}$ (equal-width and equal-mass) in Appendix F. Statistical significance of the results is confirmed through $ECE_{EW}$ error bars in Appendix G where mean and standard deviations are plotted over 5 runs.

**Out-of-Distribution (OOD) detection** Following [19], we report the performance of AdaFocal on an OOD detection task. We train ResNet-110 and Wide-ResNet26-10 on CIFAR-10 as the in-distribution data and test on SVHN [23] and CIFAR-10-C [7] (with level 5 Gaussian noise corruption) as OOD data. Using entropy of the softmax as the measure of uncertainty, the corresponding ROC plots are shown in Figure 6 and AUROC scores are reported in Table 8 in Appendix K. We see that models trained with AdaFocal outperform focal loss $\gamma = 3$ (FL-3) and FLSD-53. For the exact AUROC scores, please refer to Appendix K. These results further highlight the benefits of inherently calibrated model produced by AdaFocal as post-hoc calibration techniques such as temperature scaling, as shown in the figure, is ineffective under distributional shift [28].

**Hyper-parameters** The hyper-parameters introduced by AdaFocal are $\lambda$, $\gamma_{\min/\max}$, and $S_{th}$. However, these do not require an extensive hyper-parameter search and are much easier to select compared to $\gamma$ (which, otherwise, needs to be searched for every bin at every time step). Based on our experiments,

- $\lambda$ is redundant and one may choose to ignore it as for all our experiments $\lambda = 1$ worked very well. However, for an unknown dataset-model pair, if increasing/decreasing the rate of change of $\gamma$ improves calibration (or helps to reach the desired level faster), then one can use $\lambda$ to achieve so.

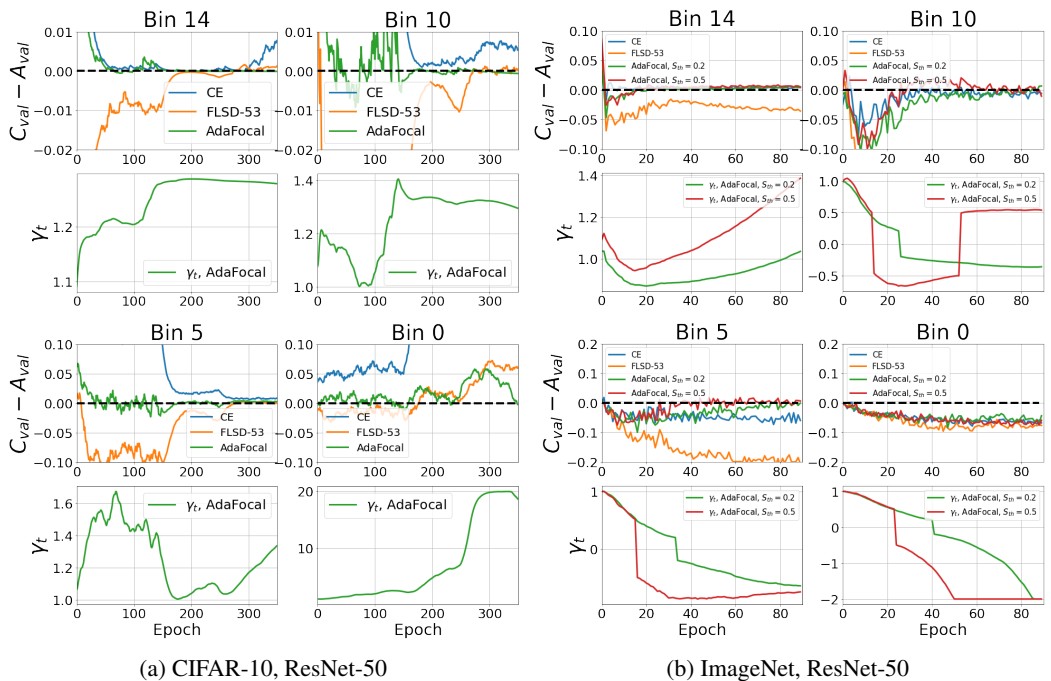

(a) CIFAR-10, ResNet-50                    (b) ImageNet, ResNet-50

Figure 5: Dynamics of $\gamma$s in different validation-bins when ResNet-50 is trained on CIFAR-10 and ImageNet with AdaFocal. For each bin, **top**: $C_{val} - A_{val}$ from the validation set, **bottom**: $\gamma_t$ vs. epochs. Black dotted line in top plot represent zero calibration error. We observe that for each bin AdaFocal is able to find the gammas that result in low calibration error. For CIFAR-10, $\gamma_t > 1$ whereas for ImageNet, starting from $\gamma_t = 1$ (focal loss), AdaFocal ultimately switches to inverse-focal loss ($\gamma_t < 0$) at $S_{th} = 0.2$ (or 0.5) for some of the bins.

| Dataset | Model | CE | Brier | MMCE | LS | FLSD-53 | AdaFocal |
|---|---|---|---|---|---|---|---|
| CIFAR-10 | ResNet50 | **4.95** | 5.00 | 4.99 | 5.29 | 4.98 | 5.30 |
| | ResNet110 | **4.89** | 5.48 | 5.40 | 5.52 | 5.42 | 5.27 |
| | WideResNet | **3.86** | 4.08 | 3.91 | 4.20 | 4.01 | 4.50 |
| | DenseNet121 | **5.00** | 5.11 | 5.41 | 5.09 | 5.46 | 5.20 |
| CIFAR-100 | ResNet50 | 23.30 | 23.39 | 23.20 | 23.43 | 23.22 | **22.60** |
| | ResNet110 | 22.73 | 25.10 | 23.07 | 23.43 | **22.51** | 22.79 |
| | WideResNet | 20.70 | 20.59 | 20.73 | 21.19 | 20.11 | **20.07** |
| | DenseNet121 | 24.52 | 23.75 | 24.0 | 24.05 | 22.67 | **22.22** |
| Tiny-ImageNet | ResNet50 | **42.90** | 46.27 | 45.96 | 44.42 | 45.12 | 45.49 |
| | ResNet110 | 42.53 | 45.47 | **42.22** | 44.13 | 44.88 | 44.55 |
| ImageNet | ResNet50 | 27.08 | 28.80 | 27.12 | 28.43 | 28.53 | **27.07** |
| | ResNet110 | 23.77 | 24.07 | 23.72 | 23.84 | 25.17 | **23.66** |
| | DenseNet121 | 27.84 | 28.02 | 27.87 | 27.79 | 29.12 | **27.74** |
| 20Newsgroup | CNN | 26.68 | 27.06 | 27.23 | **26.03** | 27.98 | 28.53 |
| | BERT | **16.05** | 16.52 | 16.16 | 16.18 | 17.57 | 17.22 |

Table 2: Test set error (%). Lowest error is marked in bold.

- $\gamma_{\min / \max}$ do not require any special fine-tuning as their sole purpose is to stop $\gamma$ from exploding in either directions. This is similar to the common practice of gradient clipping for stable training. For all our experiments, we use $\gamma_{\max} = 20$, but any reasonable value around that range should also work well in practice. For comparison of results with $\gamma_{\max} = 20$, $\gamma_{\max} = 50$ and $\gamma_{\max} = \infty$, please refer to Appendix N.

- $\gamma_{\min} = -2$ is selected based on the observation that values beyond $-2$ led to unstable training. However, if, for a new untested dataset-model pair, $\gamma_{\min} = -2$ turns out to be unsuitable, it should

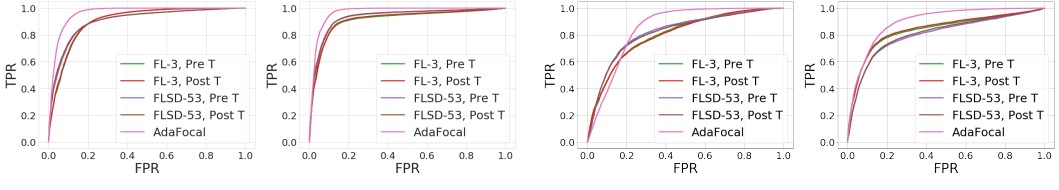

(a) SVHN: ResNet-110, WideResNet      (b) CIFAR-10-C: ResNet-110, WideResNet

Figure 6: ROC for ResNet-110 and Wide-ResNet-26-10 trained on in-distribution CIFAR-10 and tested on out-of-distribution (a) SVHN and (b) CIFAR-10-C. Pre/Post T refers to pre and post temperature scaling.

still be fairly easy to select a new threshold by simply looking at the "dynamics of $\gamma$" plots (similar to Fig. 5) at the time step the training becomes unstable.

- We use $S_{th} = 0.2$ for all our experiments. This also does not require extensive tuning and can be easily selected based on the evolution of $\gamma$ in various bins for a trial run of AdaFocal ( without switching to inverse focal loss). For example, for ImageNet (Fig. 5 ), where some of the bins are always under-confident, AdaFocal decreases $\gamma$ (starting at $\gamma = 1$) towards negative values. In this case, it makes more sense to have a higher $S_{th}$ ($= 0.5$) so that AdaFocal can switch early to inverse-focal loss and overcome the under-confidence (see ImageNet results in Fig 4 and 5).

- **Number of bins.** We experimented with AdaFocal using 5, 10, 15, 20, 30, and 50 equal-mass bins during training to draw calibration statistics form the validation set. As reported in Appendix H, the best results are for number of bins in the range of 10 to 20. Performance degrades when the number of bins are too small ($< 10$) or too large ($> 20$). Therefore, we use 15 bins for all AdaFocal trainings. Note that for computing ECE metrics as well, we use 15 bins so as to be consistent with previous works in literature [19, 4].

**Best choice of AdaFocal + post-hoc calibration** From the results in Table 1, 3, and 4 for temperature scaling, ETS and Spline fitting, respectively, there is not a clear choice of post-hoc calibration method that gives the best results across all dataset-model pairs when combined with AdaFocal. However, we do observe that in almost all cases it is the pre-calibrated model by AdaFocal that gives the lowest ECE when combined with one of these post-hoc calibration techniques. Therefore, for an unknown dataset-model pair, the choice of the best post-hoc calibration method (to be used on top of AdaFocal) might require more investigation and is a different study in itself. Overall, the evidence in our paper show that pre-calibration from training leads to even better calibrated models post hoc, and AdaFocal is much better in producing such pre-calibrated models.

## 7 Conclusion

In this work, we first revisit the calibration properties of regular focal loss to highlight the downside of using a fixed $\gamma$ for all samples. Particularly, by studying the calibration behaviour of different samples in different probability region, we find that there's no single $\gamma$ that achieves the best calibration over the entire region. We use this observation to motivate the selection of $\gamma$ independently for each sample (or group of samples) based on the knowledge of models's under/over-confidence from the validation set. We propose a calibration-aware adaptive strategy called AdaFocal that accounts for such information and updates the $\gamma_t$ at every step based on $\gamma_{t-1}$ from the previous step and the magnitude of the model's under/over-confidence. We find AdaFocal to perform consistently better across different dataset-model pairs producing inherently calibrated models that benefit further from post-hoc calibration techniques. Additionally, we find that models trained with AdaFocal are much better at out-of-distribution detection.

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
