# Appendix

## A    Broader Impact

Overconfidence in deep neural networks could easily lead to deployments where predictions are made that should have been withheld. For example, in medical diagnosis applications the Bayes optimal decision depends heavily on the model accurately modeling the distribution over its output classes. We hope that our work is a small step towards avoiding high loss errors in decision making applications. The metrics used in this paper assume that improving average calibration is the goal; but other metrics should be considered if we want to, for example, ensure good average calibration across different strata (e.g. if instances correspond to users of different social strata).

## B    Calibration Behaviour of Focal Loss in Different Bins

In the main paper, we have showed the calibration behavior of different focal losses for ResNet50 trained on CIFAR-10 for only a few bins. For completeness, the rest of the bins and their calibration error $E_i = C_{val,i} - A_{val,i}$ are shown in Figure 7 for focal losses with $\gamma = 0, 3, 4, 5$. We observe that there's no single $\gamma$ that performs the best across all the bins. Rather, every bin has a particular $\gamma$ that achieves the best calibration.

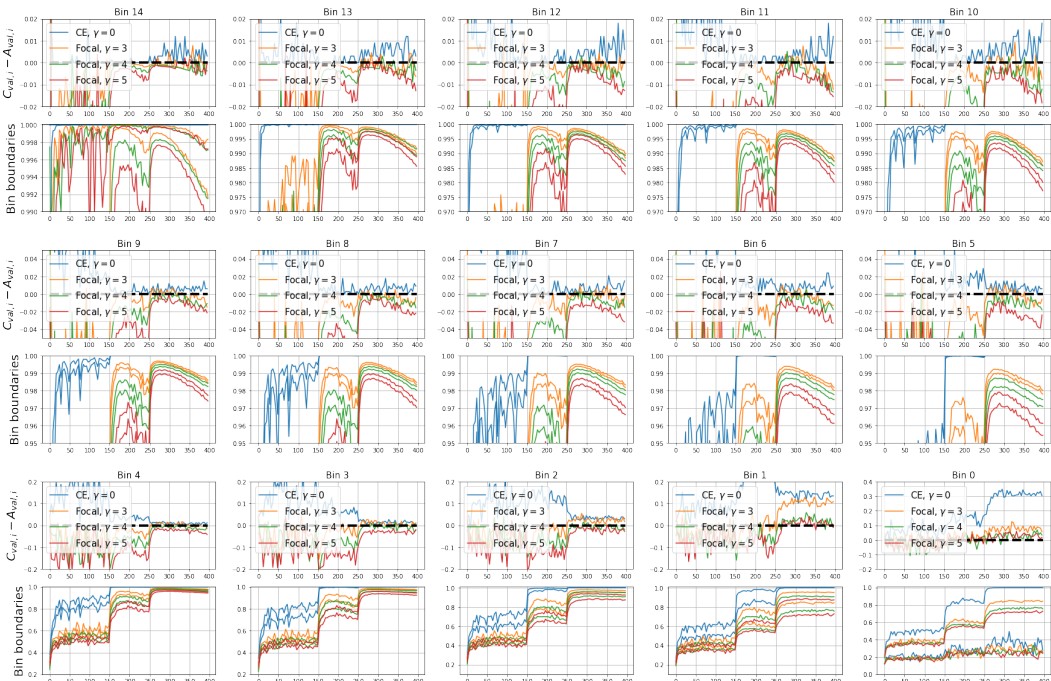

Figure 7: ResNet-50 trained on CIFAR-10 using focal loss $\gamma = 0, 3, 4, 5$. **Top**: $E_{val,i} = C_{val,i} - A_{val,i}$, **Bottom**: bin boundaries. The statistics are computed on the validations set using 15 equal-mass bins. The black horizontal dashed line in every top subfigure represents zero calibration error $E_{val,i} = 0$.

# C   Correspondence between Confidence of Training and Validation Samples

## C.1   Closeness of $C_{train,true}$ and $C_{train,top}$ as Training Progresses

For a training sample, the confidence of the true class $y_{\text{true}}$ is denoted by $\hat{p}_{train,true}$ and the average equivalent in a bin by $C_{train,true}$. Similarly, the confidence of the top predicted class $\hat{y}$ (for the training sample) is denoted by $\hat{p}_{train,top}$ and the average equivalent in a bin by $C_{train,top}$. For the training set, we care only about the confidence of the "true class" $\hat{p}_{train,true}$ as that is the quantity which gets manipulated by some loss function. For validation set, on the other hand, we care about the confidence of the "top predicted class". Therefore, it would be more natural to look for correspondence between similar quantities, particularly $C_{train,top,i}$, across the two datasets. However, as we shown in Fig. 8, $C_{train,true,i}$ and $C_{train,top,i}$ are almost the same during major part of the training. This is because as the model approaches towards $100\%$ accuracy on the training set, the top predicted class and the true class for a training sample become the same. Therefore we can directly compare the two different quantities $C_{train,true,i}$ and $C_{train,top,i}$ across the training and validation set.

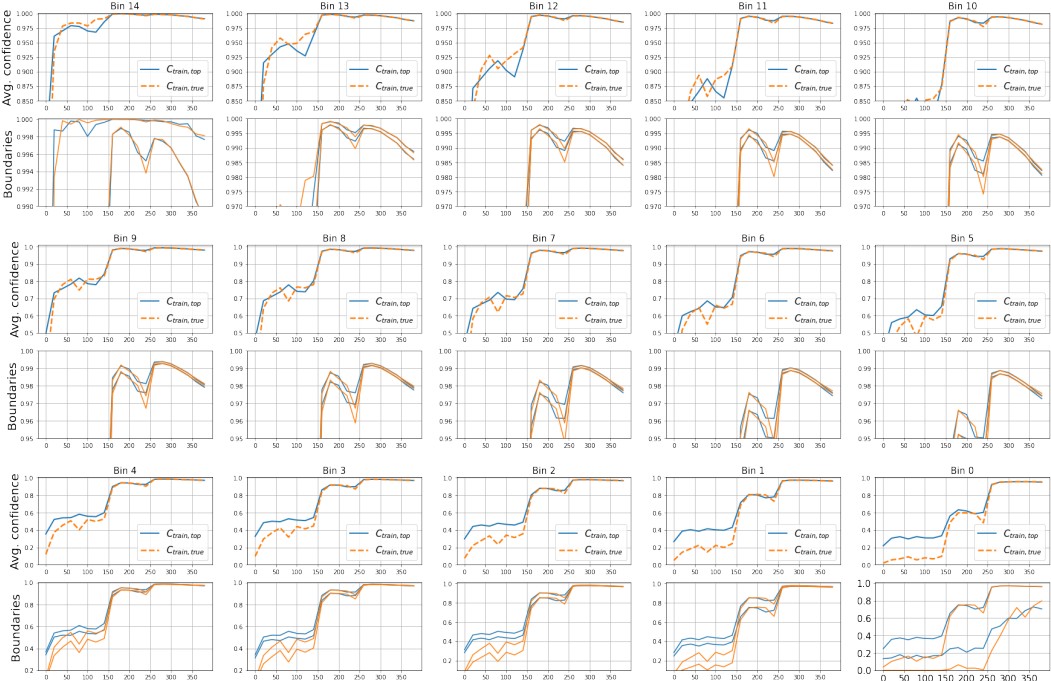

Figure 8: ResNet50 trained on CIFAR-10 with focal loss $\gamma = 3$. The figure shows the closeness of $C_{train,true,i}$ (**orange**) and $C_{train,top,i}$ (**blue**) for training samples as training progresses towards $100\%$ accuracy on the training set.

## C.2 CIFAR-10, ResNet-50: Correspondence between $C_{train}$ and $C_{val}$

Fig. 9 and 10 show the correspondence between the confidence of training samples $C_{train} \equiv C_{train,true}$ and the confidence of the validation samples $C_{val} \equiv C_{val,top}$ for the dataset-model pair CIFAR-10, ResNet-50, under following two cases:

- **Independent binning**: when training samples and validation samples are grouped independently into their respective training-bins and validation-bins (Fig. 9).
- **Common binning**: when training samples are grouped using the common bin boundaries of the validation-bins that were formed by binning the validation bins (Fig. 10).

### C.2.1 Independent binning

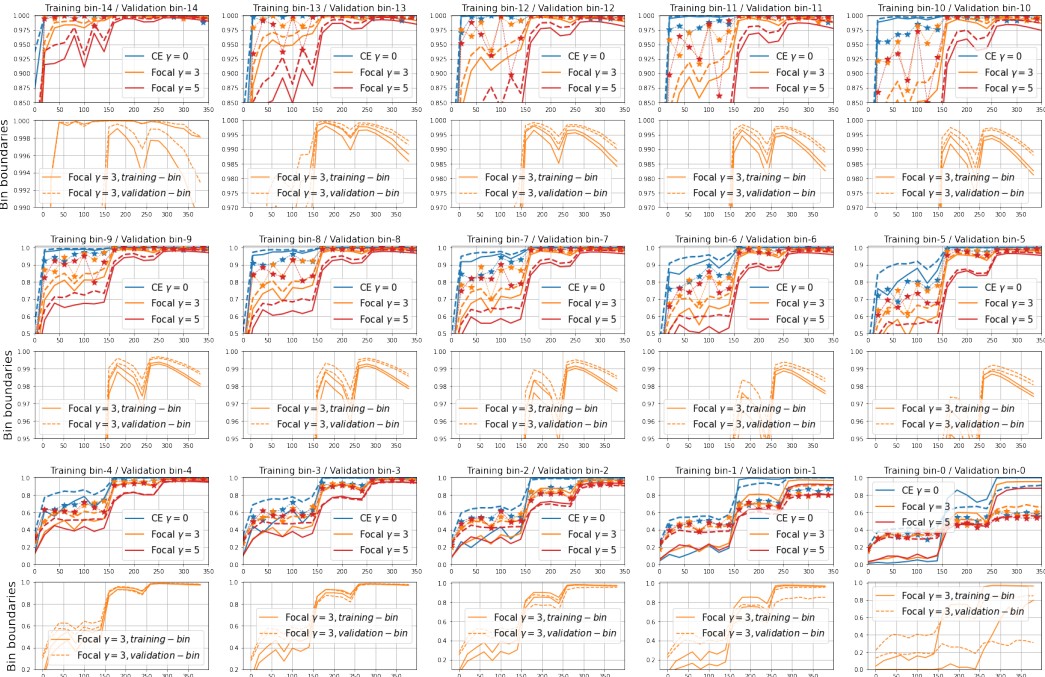

Figure 9: **Independent binning**: training samples and validation samples are grouped independently into training-bin and validation-bin respectively. The top subfigure for each bin shows the correspondence between average confidence of a group of training samples $C_{train,true,i}$ and a group of validation samples $C_{val,top,i}$ when ResNet-50 is trained on CIFAR-10 with focal loss $\gamma = 0, 3, 5$. The binning is adaptive with 15 equal-mass bins. **Solid line**: $C_{train,true,i}$ in training-bin $i$, **Dashed line**: $C_{val,top,i}$ and **Star-dashed line**: $A_{val,i}$ in validation-bin $i$. The bottom subfigure shows the bin boundaries for focal loss $\gamma = 3$ as an example.

### C.2.2 Common binning

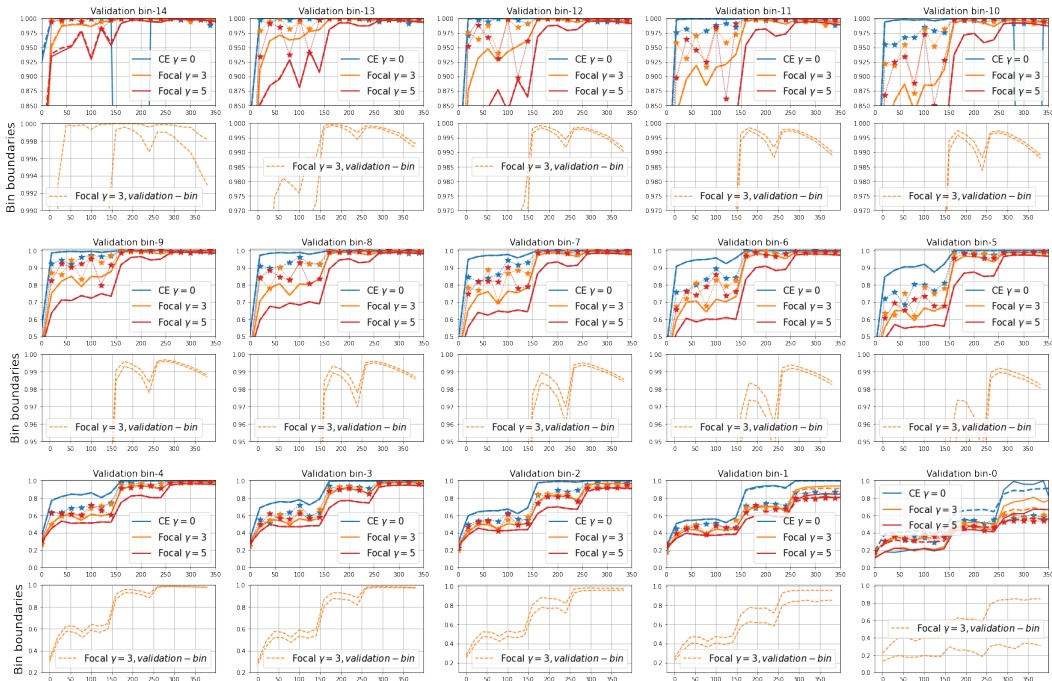

Figure 10: **Common binning**: training samples are grouped using the bin boundaries of the validation-bins. The top subfigure for each bin shows the correspondence between average confidence of a group of training samples $C_{train,true,i}$ and a group of validation samples $C_{val,top,i}$ when ResNet-50 is trained on CIFAR-10 with focal loss $\gamma = 0, 3, 5$. The binning is adaptive with 15 equal-mass bins. **Solid line**: $C_{train,true,i}$ in validation-bin $i$, **Dashed line**: $C_{val,top,i}$ and **Star-dashed line**: $A_{val,i}$ in validation-bin $i$. The bottom subfigure shows the bin bin boundaries for focal loss $\gamma = 3$ as an example.

## C.3 CIFAR-100, ResNet-50: Correspondence between $C_{train}$ and $C_{val}$

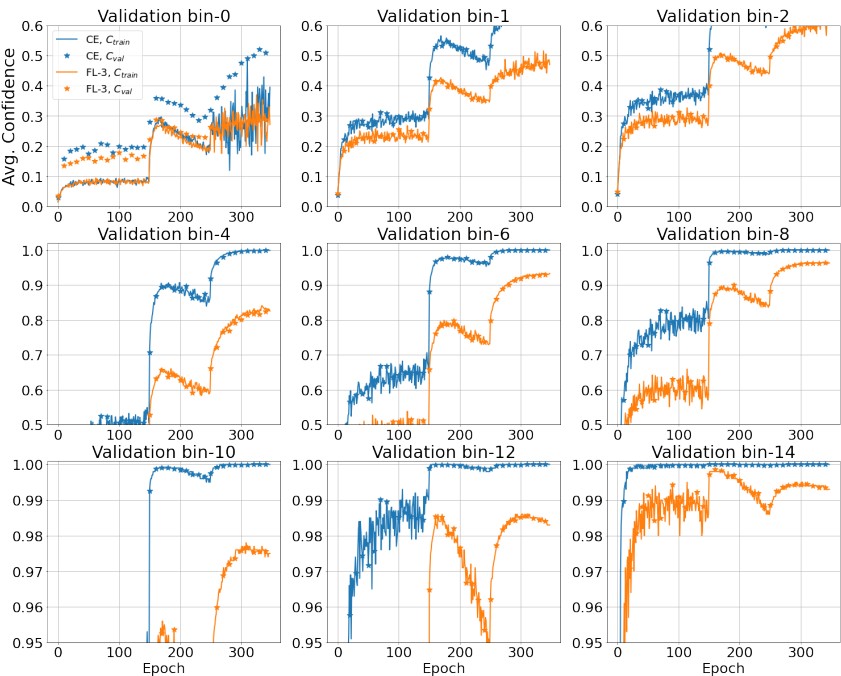

Figure 11: **Common binning**: $C_{train}$ (solid) and $C_{val}$ (star) both binned using validation-bin boundaries. Show here for focal loss $\gamma = 0$ (CE) and $\gamma = 3$ (FL-3).

## C.4 CIFAR-100, WideResNet: Correspondence between $C_{train}$ and $C_{val}$

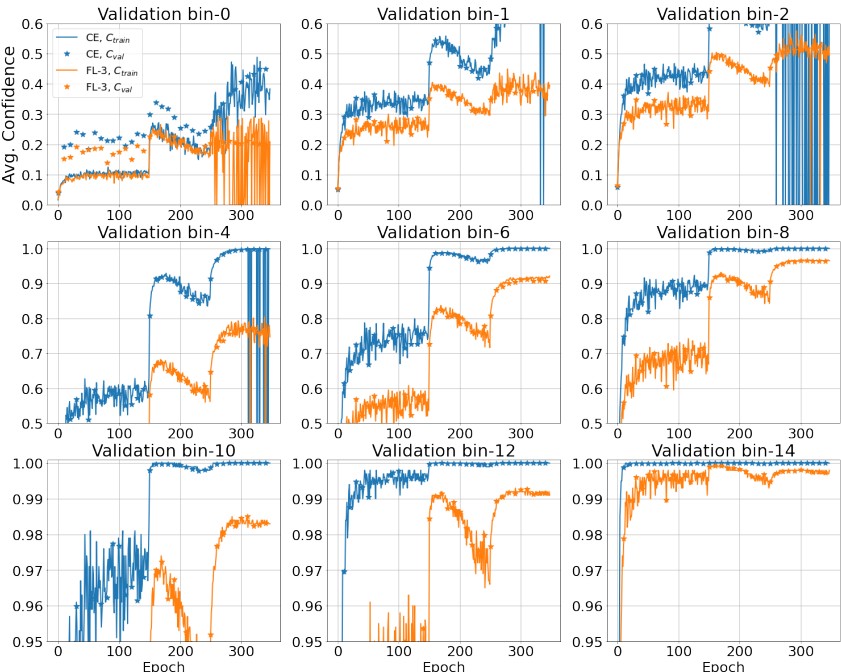

Figure 12: **Common binning**: $C_{train}$ (solid) and $C_{val}$ (star) both binned using validation-bin boundaries. Show here for focal loss $\gamma = 0$ (CE) and $\gamma = 3$ (FL-3).

## C.5   TinyImageNet, ResNet-50: Correspondence between $C_{train}$ and $C_{val}$

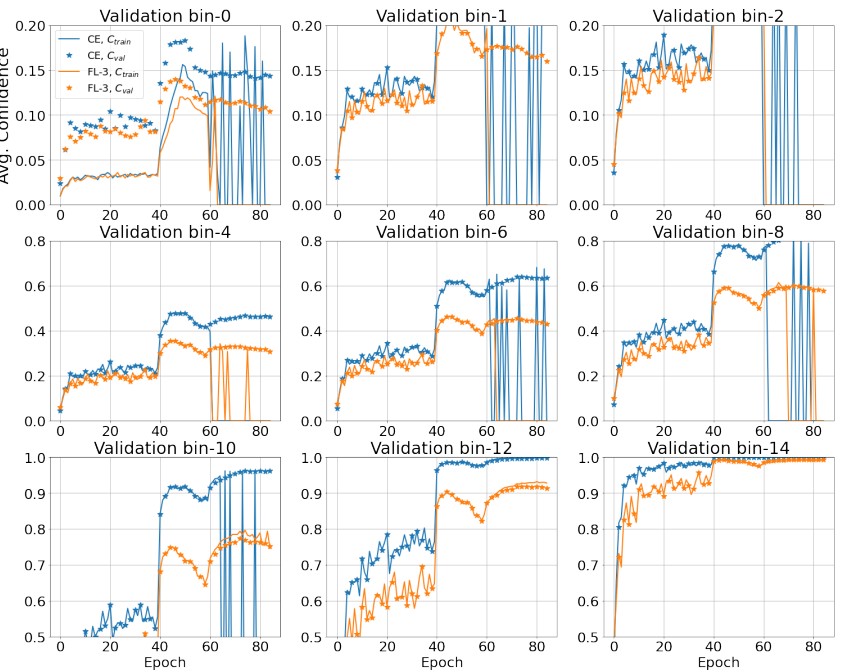

Figure 13: **Common binning**: $C_{train}$ (solid) and $C_{val}$ (star) both binned using validation-bin boundaries. Show here for focal loss $\gamma = 0$ (CE) and $\gamma = 3$ (FL-3).

## C.6   20 Newsgroups, CNN: Correspondence between $C_{train}$ and $C_{val}$

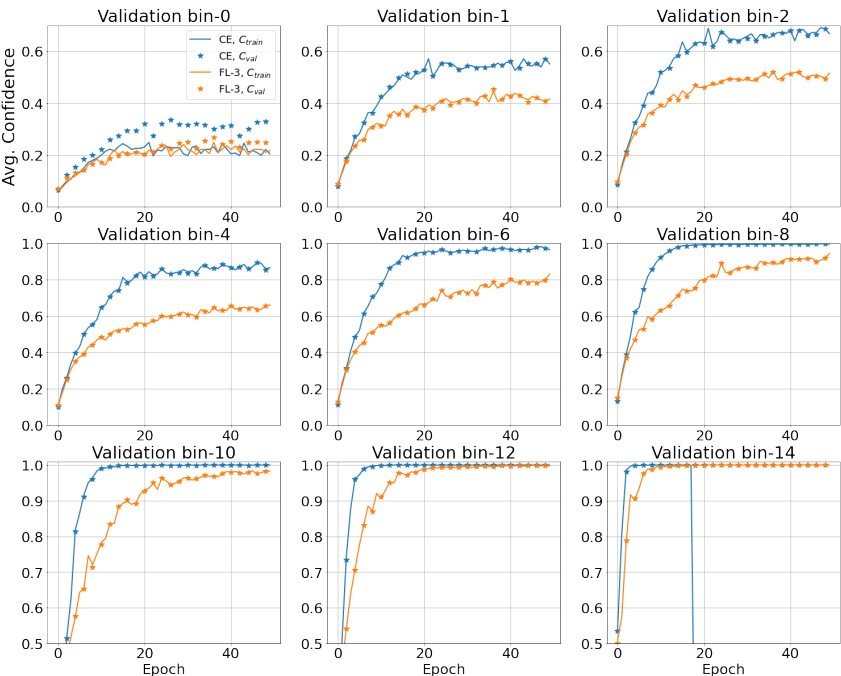

Figure 14: **Common binning**: $C_{train}$ (solid) and $C_{val}$ (star) both binned using validation-bin boundaries. Show here for focal loss $\gamma = 0$ (CE) and $\gamma = 3$ (FL-3).

# D    Datasets and Experiments

## D.1    Dataset Description

**CIFAR-10** [9]: This dataset contains $60,000$ coloured images of size $32 \times 32$, which are equally divided into 10 classes. A split of $45,000/5,000/10,000$ images is used as train/validation/test sets respectively.

**CIFAR-100** [9]: This dataset contains $60,000$ coloured images of size $32 \times 32$, which are equally divided into 100 classes. A split of $45,000/5,000/10,000$ images is used as train/validation/test sets respectively.

**ImageNet** [27]: ImageNet Large Scale Visual Recognition Challenge (ILSVRC) 2012-2017 is an image classification and localization dataset. This dataset spans 1000 object classes and contains 1,281,167 training images and 50,000 validation images.

**Tiny-ImageNet** [2]: It is a subset of the ImageNet dataset with $64 \times 64$ dimensional images and 200 classes. It has 500 images per class in the training set and 50 images per class in the validation set.

**20 Newsgroups** [14]: This dataset contains $20,000$ news articles, categorised evenly into 20 different newsgroups. Some of the newsgroups are very closely related to each other (e.g. comp.sys.ibm.pc.hardware / comp.sys.mac.hardware), while others are highly unrelated (e.g misc.forsale / soc.religion.christian). We use a train/validation/test split of $15,098/900/3,999$ documents.

## D.2    Experiment Configurations

For our experiments, we have used Nvidia Titan X Pascal GPU with 12 GB of memory. Training configuration for each dataset is given below.

**CIFAR-10** and **CIFAR-100**: We use SGD with a momentum of 0.9 as our optimiser, and train the networks for 350 epochs, with a learning rate of 0.1 for the first 150 epochs, 0.01 for the next 100 epochs, and 0.001 for the last 100 epochs. We use a training batch size of 128. The training data is augmented by applying random crops and random horizontal flips.

**Tiny-ImageNet**: We use SGD with a momentum of 0.9 as our optimiser, and train the models for 100 epochs with a learning rate of 0.1 for the first 40 epochs, 0.01 for the next 20 epochs and 0.001 for the last 40 epochs. We use a training batch size of 64. Note that we use 50 samples per class (i.e. a total of 10000 samples) from the training set as the validation set. Hence, the training is only on 90000 images. We use the Tiny-ImageNet validation set as our test set.

**ImageNet**: We use SGD as our optimiser with momentum of 0.9 and weight decay $10^{-4}$, and train the models for 90 epochs with a learning rate of 0.01 for the first 30 epochs, 0.001 for the next 30 epochs and 0.0001 for the last 30 epochs. We use a training batch size of 128. We divide the 50,000 validation images into validation and test set of 25,000 images each.

**20 Newsgroups, CNN**: We train the Global Pooling CNN Network [15] using the Adam optimiser, with learning rate 0.001, and default betas 0.9 and 0.999. We used Glove word embeddings [25] to train the network. We train the model for 50 epochs and use the model at the end to evaluate the performance.

**20 Newsgroups, BERT**: We fine-tune a BERT model by adding a single linear classification layer on top of pre-trained "bert-base-uncased" model (12-layer, 768-hidden, 12-heads, 110M parameters) [3], using the AdamW optimiser (Adam with weight decay), with batch size of 32, learning rate $2e - 5$, weight decay of 0.01 and warm up steps of 0.2 the number of batches in the training set. We limit the length of the input sequence to 128 for training and 512 for testing. We train the model for 10 epochs and select the model that has the lowest error on validation set.

The experiments are implemented using PyTorch library. The hyperparameters that are not explicitly mentioned above are set to their default values in PyTorch. For CIFAR-10/100 and Tiny-ImageNet, AdaFocal is implemented on top of the base code available at [18]. The code for 20 Newsgroups is implemented in PyTorch by adapting the TensorFlow code available at [12].

## D.3 Model Selection

For all experiments, except Tiny-ImageNet, we select the model at the end of the training mainly to be consistent with [19] i.e. the work we are trying to improve upon in this paper. As confirmed by the authors of [19], they use the model at the end of the training to report results in the paper. Therefore, for the following datasets, the error and ECE results are reported for the model at

- CIFAR-10: 350 epochs
- CIFAR-100: 350 epochs
- ImageNet: 90 epochs
- 20 NewsGroups, CNN: 50 epochs

For Tiny-ImageNet and BERT, we have reported the model that has the lowest error on the validation set.

## E Other Post Hoc Calibration Techniques

### E.1 Ensemble Temperature Scaling (ETS)

| Dataset | Model | Cross Entropy | FLSD-53 | AdaFocal |
|---|---|---|---|---|
| CIFAR-10 | ResNet-50 | 2.97 | 1.71 | **0.55** |
| | ResNet-110 | 3.18 | 1.79 | **0.57** |
| | Wide-ResNet-26-10 | 2.55 | 2.00 | **0.49** |
| | DenseNet-121 | 3.40 | 1.64 | **0.57** |
| CIFAR-100 | ResNet-50 | 3.38 | 2.46 | **1.33** |
| | ResNet-110 | 4.60 | 3.87 | **1.24** |
| | Wide-ResNet-26-10 | 2.91 | 2.07 | **1.79** |
| | DenseNet-121 | 4.48 | **1.21** | 1.86 |
| Tiny-ImageNet | ResNet-50 | 3.02 | 1.46 | **1.23** |
| | ResNet-110 | 1.26 | 1.22 | **0.62** |
| ImageNet | ResNet-50 | **0.90** | 2.13 | 1.13 |
| | ResNet-110 | 1.38 | 2.25 | **1.28** |
| | DenseNet-121 | **1.07** | 2.36 | 1.40 |
| 20 Newsgroups | CNN | 2.46 | 2.50 | **2.29** |
| | BERT | 5.34 | **3.91** | 4.30 |

Table 3: $ECE_{EW}$ (%) after post hoc calibration with Ensemble Temperature Scaling.

### E.2 Spline Fitting

| Dataset | Model | Cross Entropy | FLSD-53 | AdaFocal |
|---|---|---|---|---|
| CIFAR-10 | ResNet-50 | 1.69 | **0.60** | 0.65 |
| | ResNet-110 | 1.88 | 0.61 | **0.58** |
| | Wide-ResNet-26-10 | 1.17 | 0.65 | **0.45** |
| | DenseNet-121 | 1.48 | 0.97 | **0.53** |
| CIFAR-100 | ResNet-50 | 2.56 | 1.07 | **1.01** |
| | ResNet-110 | 3.36 | 1.33 | **1.29** |
| | Wide-ResNet-26-10 | 2.20 | **1.08** | 1.53 |
| | DenseNet-121 | 2.83 | **1.03** | 1.36 |
| Tiny-ImageNet | ResNet-50 | 1.44 | 1.91 | **1.39** |
| ImageNet | ResNet-50 | 0.82 | 0.87 | **0.66** |
| | ResNet-110 | **0.60** | 0.69 | 0.62 |
| | DenseNet-121 | 0.72 | **0.66** | 0.75 |
| 20 Newsgroups | Global-pool CNN | 1.97 | 1.38 | **1.12** |

Table 4: $ECE_{EW}$ (%) after post hoc calibration with spline fitting.

## F Debiased Estimates of ECE: $ECE_{DEBIAS}$ and $ECE_{SWEEP}$

As shown in [26], binning-based estimators $ECE_{EW}$ and $ECE_{EM}$ may suffer from statistical bias ($ECE_{EM}$ has lower bias than $ECE_{EW}$) and if the bias is strong enough it may lead to mis-estimation

of calibration error and a wrong model selection. Therefore, to confirm that the results presented in the paper using $ECE_{EM}$ and $ECE_{EW}$ are consistent and reliable, we additionally present here $ECE_{DEBIAS}$ [11] and $ECE_{SWEEP}$ [26] (equal-mass) as debiased estimates of ECE.

| Dataset | Model | Pre Temperature scaling | | | Post Temperature scaling | | |
|---|---|---|---|---|---|---|---|
| | | CE | FLSD-53 | AdaFocal | CE | FLSD-53 | AdaFocal |
| CIFAR-10 | ResNet-50 | 4.05 | 1.62 | **0.47** | 1.70(2.5) | 1.62(1.0) | **0.82(0.9)** |
| | ResNet-110 | 4.38 | 1.82 | **0.32** | 2.20(2.7) | 1.30(1.1) | **0.32(1.0)** |
| | Wide-ResNet-26-10 | 3.52 | 2.01 | **0.59** | 1.89(2.2) | 1.50(0.9) | **0.25(1.1)** |
| | DenseNet-121 | 4.26 | 1.56 | **0.42** | 2.15(2.3) | 1.93(0.9) | **0.42(1.0)** |
| CIFAR-100 | ResNet-50 | 17.73 | 5.52 | **1.46** | 3.86(2.2) | 2.92(1.1) | **1.46(1.0)** |
| | ResNet-110 | 19.44 | 7.31 | **1.35** | 6.01(2.3) | 3.55(1.2) | **1.35(1.0)** |
| | Wide-ResNet-26-10 | 14.91 | 2.53 | **2.12** | 3.32(2.1) | 2.53(1.0) | **2.12(1.0)** |
| | DenseNet-121 | 19.82 | 2.29 | **1.27** | 3.44(2.3) | 2.12(1.1) | **1.27(1.0)** |
| Tiny-ImageNet | ResNet-50 | 7.95 | 2.90 | **2.69** | 3.86(1.44) | 2.61(1.06) | **2.31(0.96)** |
| | ResNet-110 | 8.09 | 1.65 | **1.50** | 1.23(1.20) | 1.65(1.00) | **1.50(1.0)** |
| ImageNet | ResNet-50 | 2.89 | 16.76 | **1.74** | **1.42(0.90)** | 2.58(0.70) | 1.74(1.00) |
| | ResNet-110 | 1.14 | 18.65 | **1.04** | 1.14(1.00) | 2.41(0.70) | **1.04(1.00)** |
| | DenseNet-121 | 1.74 | 19.18 | **1.30** | 1.74(1.00) | 2.17(0.70) | **1.30(1.00)** |
| 20 Newsgroups | Global-pool CNN | 18.36 | 8.94 | **1.84** | 5.23(4.1) | **0.94(1.6)** | 1.84(1.0) |

Table 5: Test set $ECE_{DEBIAS}(\%)$ 15 bins. Optimal temperature, shown in brackets, are selected based on the lowest $ECE_{EW}$ on the validation set.

| Dataset | Model | Pre Temperature scaling | | | Post Temperature scaling | | |
|---|---|---|---|---|---|---|---|
| | | CE | FLSD-53 | AdaFocal | CE | FLSD-53 | AdaFocal |
| CIFAR-10 | ResNet-50 | 4.05 | 1.54 | **0.04** | 1.43(2.5) | 1.54(1.0) | **0.70(0.9)** |
| | ResNet-110 | 4.38 | 1.83 | **0.40** | 1.34(2.7) | 1.32(1.1) | **0.40(1.0)** |
| | Wide-ResNet-26-10 | 3.53 | 1.64 | **0.38** | 1.41(2.2) | 1.55(0.9) | **0.32(1.1)** |
| | DenseNet-121 | 4.27 | 1.58 | **0.34** | 2.17(2.3) | 1.98(0.9) | **0.34(1.0)** |
| CIFAR-100 | ResNet-50 | 17.72 | 5.51 | **1.89** | **0.51(2.2)** | 2.36(1.1) | 1.89(1.0) |
| | ResNet-110 | 19.44 | 7.34 | **1.58** | 3.71(2.3) | 3.65(1.2) | **1.58(1.0)** |
| | Wide-ResNet-26-10 | 14.92 | 2.62 | **2.25** | 2.62(2.1) | 2.62(1.0) | **2.25(1.0)** |
| | DenseNet-121 | 19.82 | 2.25 | **1.47** | 3.12(2.3) | 2.31(1.1) | **1.47(1.0)** |
| Tiny-ImageNet | ResNet-50 | 7.98 | 2.99 | **2.78** | 3.96(1.44) | 2.83(1.06) | **2.56(0.96)** |
| | ResNet-110 | 8.11 | 2.01 | **1.97** | 1.81(1.20) | 2.01(1.00) | **1.97(1.00)** |
| ImageNet | ResNet-50 | 2.93 | 16.77 | **1.98** | **1.63(0.90)** | 2.58(0.70) | 1.98(1.00) |
| | ResNet-110 | 1.15 | 18.66 | **1.08** | 1.15(1.00) | 2.51(0.70) | **1.08(1.00)** |
| | DenseNet-121 | 1.80 | 19.19 | **1.40** | 1.80(1.00) | 2.29(0.70) | **1.40(1.00)** |
| 20 Newsgroups | Global-pool CNN | 18.38 | 8.95 | **2.22** | 5.53(4.1) | **2.13(1.6)** | 2.22(1.0) |

Table 6: Test set $ECE_{SWEEP}(\%)$ equal-mass. Optimal temperature, shown in brackets, are selected based on the lowest $ECE_{EW}$ on the validation set.

# G  $ECE_{EW}$ **error bars**

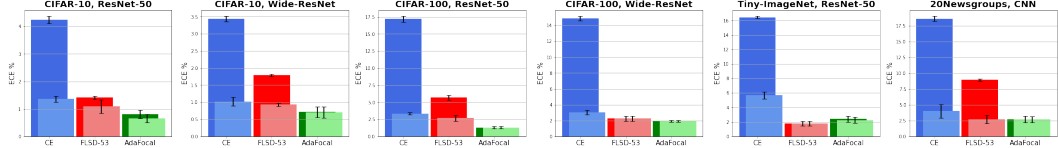

Figure 15: Test set $ECE_{EW}$ (%) error bars with mean and standard deviation computed over 5 runs with different initialization seed. Dark and light shades of a color show pre and post temperature scaling results respectively. Optimal temperatures are cross-validated based on $ECE_{EW}$.

# H  **Number of bins used for AdaFocal training**

Experiment details:

1. ResNet-50 trained on CIFAR-10 for 350 epochs.
2. The reported results below are without temperature scaling.

3. We compare AdaFocal with 5, 10, 15, 20, 30, and 50 equal mass bins vs FLSD-53.

Note that there are two types of binning involved:

- **For training**: the binning that is performed on the validation set from where AdaFocal draws calibration related information to adjust $\gamma$. These correspond to the columns in the table 7.
- **For evaluation**: once we have a trained model, the binning that is used to compute the ECE metric. These correspond to the rows in the table below.

| Evaluation Metric | FLSD-53 | Number of bins used for AdaFocal training | | | | | |
| | | 5 bins | 10 bins | 15 bins | 20 bins | 30 bins | 50 bins |
|---|---|---|---|---|---|---|---|
| $ECE_{EW}$ (15bins) | 1.35 | 0.76 | 0.53 | 0.51 | 0.60 | 0.82 | 1.16 |
| $ECE_{EM}$ (15bins) | 1.67 | 0.63 | 0.53 | 0.56 | 0.40 | 0.84 | 1.10 |
| $ECE_{DEBIAS}$ (15bins) | 1.62 | 0.50 | 0.44 | 0.47 | 0.25 | 0.79 | 1.07 |
| $ECE_{DEBIAS}$ (30bins) | 1.57 | 0.73 | 0.43 | 0.46 | 0.27 | 0.72 | 1.06 |
| $ECE_{SWEEP-EW}$ | 1.31 | 0.66 | 0.43 | 0.48 | 0.48 | 0.80 | 1.08 |
| $ECE_{SWEEP-EM}$ | 1.54 | 0.53 | 0.21 | 0.04 | 0.38 | 0.07 | 1.08 |

Table 7: ECE (%) performance for ResNet-50 trained on CIFAR-10 when AdaFocal training uses different number of equal-mass bins. We observe that the best results are for number of bins in the range of 10 to 20. Performance degrades when the number of bins are too less ($< 10$) or too many ($\geq 30$).

# I   Frequency of $\gamma$-update

In this section, we study how the frequency of the $\gamma$-update affect the performance of AdaFocal on the 20Newsgroup dataset (with CNN and BERT models) i.e if the validation-bin boundaries and $\gamma$ are updated every mini-batch or a few times per training epoch.

Intuitively, one would expect that if the validation-bin boundaries are updated more frequently then AdaFocal would be able to more closely track the changes in the calibration behaviour of the validation set and accordingly adjust it's $\gamma$s to better respond to the changes. This is supported by the experiments on 20 Newsgroup dataset using CNN and BERT model as shown in Fig. 16 and 17. In these figures, AdaFocal on its own means that $\gamma$ is updated at the end of every epoch. AdaFocal-$n$, where $n = 100, 50, 10, 1$, means that $\gamma$ is updated every $n$ mini-batches. Since the number of training batches for 20Newsgroup with CNN is 118, AdaFocal-50(respectively 10, 1) means $\gamma$ is updated 2 (respectively 11, 118) times per epoch. Similarly, for 20Newsgroup and BERT, as the the number of training batches is 472, AdaFocal-100(respectively 10, 1) means $\gamma$ is updated 4 (respectively 47, 472) times per epoch.

From these experiments, we observe that

1. **Frequent updates keep the model better calibrated at all time steps and prevent it from getting mis-calibrated**. For example in Fig. 16(b), AdaFocal and AdaFocal-50 are miscalibrated at the start of the training and at around 15-20 epoch, whereas AdaFocal-10 and AdaFocal-1 remains very well calibrated at all epochs. We observe the same in Fig. 17(b), where AdaFocal and AdaFocal-100 are miscalibrated at the start of the training and around epoch $6 - 9$, whereas AdaFocal-10 and AdaFocal-1 remains well calibrated at all times.

2. **For cases where updating $\gamma$ once per epoch leads to only a few $\gamma$ updates in total, frequent updates may lead to improved ECE performance**. We observe this in Fig. 17(b) where AdaFocal (with only 10 updates as BERT is trained for only 10 epochs) is unable to reach the calibration level of AdaFocal-10 and AdaFocal-1. For 20Newsgroup and CNN, as the model is trained for 50 epochs, AdaFocal, first unable to keep up with AdaFocal-10 and AdaFocal-1, is able to ultimately reach their level of calibration with enough updates.

**Drawback: Increased training time**. Updating $\gamma$ more frequently comes at the cost of increased training time. For example, as discussed in J, standard training of ResNet-50 on CIFAR-10 takes

79.1s per epoch and $\gamma$-update step takes 2.8s. When trained for 350 epochs, updating $\gamma$ every epoch adds 16 minutes to the the total training time of 7.7h. Therefore, increasing the update frequency will linearly increase the additional overhead as well.

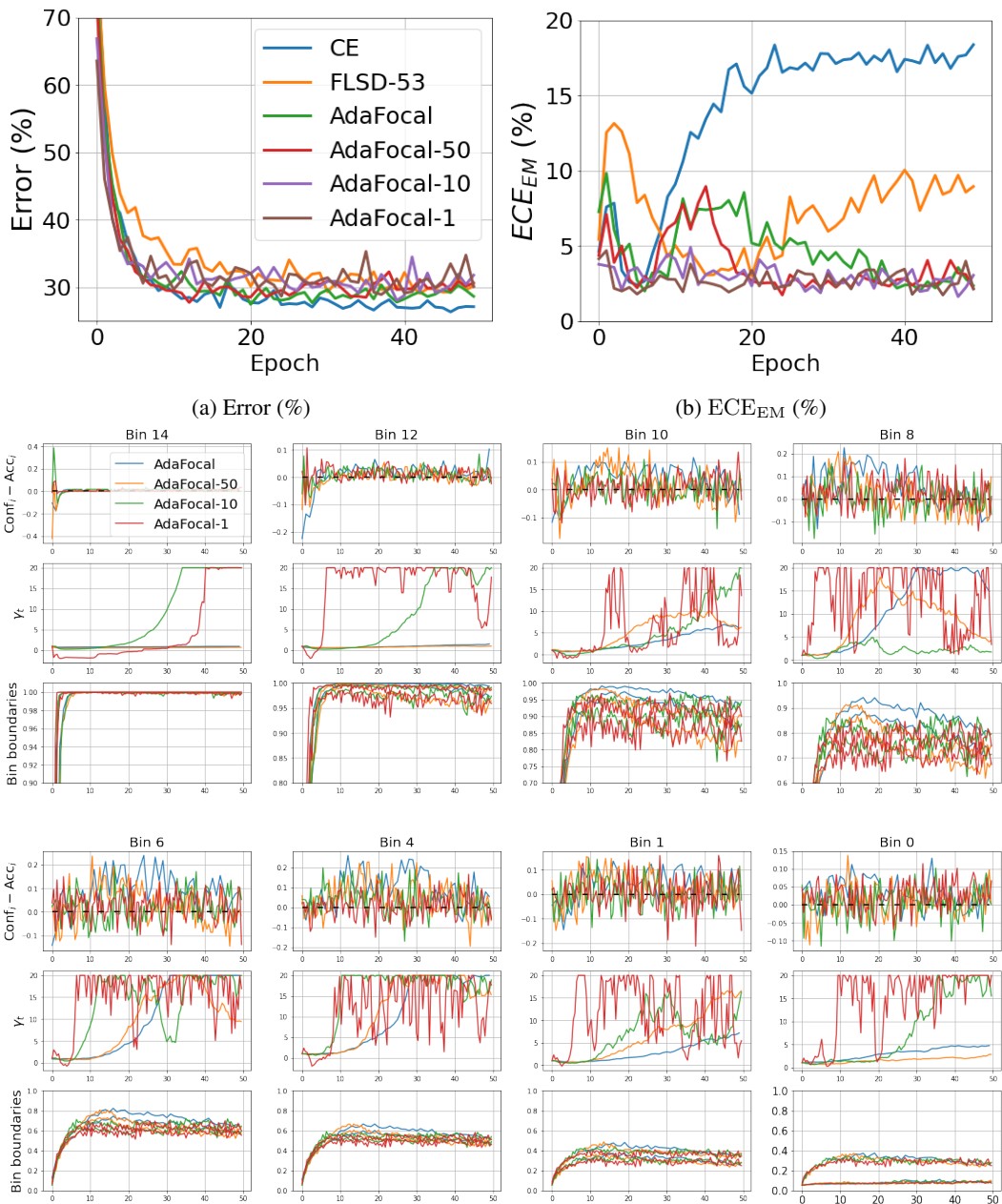

(a) Error (%)

(b) $ECE_{EM}$ (%)

(c) Dynamics of $\gamma$ and calibration behaviour in different bins. Each bin has three subplots: **top**: $E_{val,i} = C_{val,i} - A_{val,i}$, **middle**: evolution of $\gamma_t$, and **bottom**: bin boundaries. Black dashed line in top plot represent zero calibration error.

Figure 16: CNN trained on 20Newsgroups with cross entropy (CE), FLSD-53, and AdaFocal. AdaFocal on its own means that $\gamma$ is updated at the end of every epoch. AdaFocal-$n$, where $n = 50, 10, 1$, means that $\gamma$ is updated after every $n$ mini-batches. Since the number of training batches for 20Newsgroup with CNN is 118, AdaFocal-50 (respectively 10, 1) means $\gamma$ is updated 2 (respectively 11, 118) times per epoch

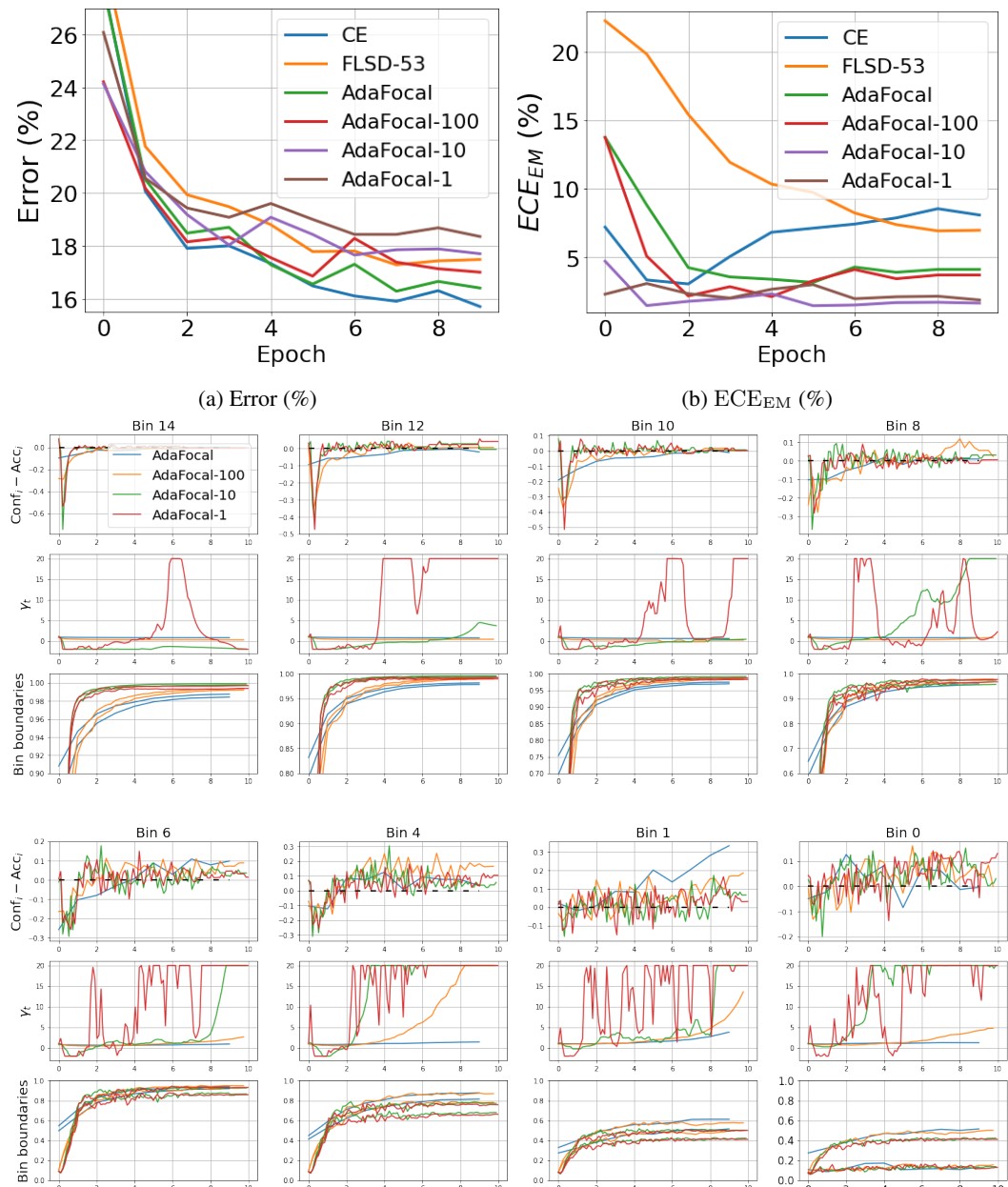

(a) Error (%)

(b) ECE$_{\text{EM}}$ (%)

(c) Dynamics of $\gamma$ and calibration behaviour in different bins. Each bin has three subplots: **top**: $E_{val,i} = C_{val,i} - A_{val,i}$, **middle**: evolution of $\gamma_t$, and **bottom**: bin boundaries. Black dashed line in top plot represent zero calibration error.

Figure 17: Pre-trained BERT fine-tuned on 20 Newsgroups with cross entropy (CE), FLSD-53, and AdaFocal. AdaFocal on its own means that $\gamma$ is updated at the end of every epoch. AdaFocal-$n$, where $n = 100, 10, 1$, means that $\gamma$ is updated after every $n$ mini-batches. Since the number of training batches for 20Newsgroup with BERT is $472$, AdaFocal-100 (respectively 10, 1) means $\gamma$ is updated $4$ (respectively $47, 472$) times per epoch.

## J   Computation overhead for AdaFocal

To update $\gamma$ of AdaFocal, the extra operations that are required are

1. Forward pass on the validation set to compute the logits/softmaxes.

2. Compute bin statistics and update $\gamma$.

In general, if we update $\gamma$ at the end of every epoch, then compared to the time it takes to train the model for one whole epoch, these two overheads are quite negligible. For example, for ResNet-50 trained on CIFAR-10 (train set contains $45000$ examples, val set contains $5000$ examples) using Nvidia Titan X Pascal GPU with 12GB memory,

- Training for one epoch = $79,123$ ms = $79.1$ s

- Forward pass on validation set = $2,886$ ms = $2.8$ s

- Compute bin statistics and update $\gamma = 8$ ms

So if the standard training with cross entropy, without any involvement of a validation set, for 350 epochs requires in total $79.1 \times \frac{350}{3600} = 7.7$ hours, then AdaFocal will add $2.808 \times \frac{350}{60} = 16$ minutes on top of the entire training. Naturally, if we update $\gamma$ more often during the epoch then this overhead will increase and may become significant. However, for all our experiments we update $\gamma$ at the end of an epoch and that works quite well. Nonetheless, for a comparison of performance of AdaFocal when the update frequency of $\gamma$ is varied, please refer to Appendix I.

# K   AUROC for Out-of-Distribution Detection

For ResNet110 trained on in-distribution CIFAR-10 and tested on out-of-distribution SVHN, we were not able to reproduce the reported results of 96.74, 96.92 for focal loss $\gamma = 3$ (FL-3) as given in [19]. Instead we found those values to be 90.27, 90.39 and report the same in Table 8 below.

| Dataset | Model | Cross Entropy | | Brier Loss | | MMCE | | LS-0.05 | | FL-3 | | FLSD-53 | | AdaFocal |
|---|---|---|---|---|---|---|---|---|---|---|---|---|---|---|
| | | Pre T | Post T | Pre T | Post T | Pre T | Post T | Pre T | Post T | Pre T | Post T | Pre T | Post T | Pre T |
| CIFAR-10 / SVHN | ResNet-110 | 61.71 | 59.66 | 94.80 | 95.13 | 85.31 | 85.39 | 68.68 | 68.68 | 90.27 | 90.39 | 90.33 | 90.49 | **96.09** |
| | Wide-ResNet-26-10 | 96.82 | 97.62 | 94.51 | 94.51 | 97.35 | 97.95 | 84.63 | 84.66 | 90.92 | 91.30 | 93.08 | 93.11 | **96.63** |
| CIFAR-10 / CIFAR-10-C | ResNet-110 | 77.53 | 75.16 | 84.09 | 83.86 | 71.96 | 70.02 | 72.17 | 72.18 | 80.11 | 79.78 | 82.06 | 81.38 | **84.96** |
| | Wide-ResNet-26-10 | 81.06 | 80.68 | 85.03 | 85.03 | 82.17 | 81.72 | 71.10 | 71.16 | 83.33 | 84.00 | 80.00 | 80.76 | **89.52** |

Table 8: AUROC (%) of models trained on CIFAR-10 as the in-distribution data and tested on SVHN and CIFAR-10-C as out-of-distribution data. Temperature scaling is based on $\text{ECE}_{\text{EW}}$.

# L   Reliability Diagrams

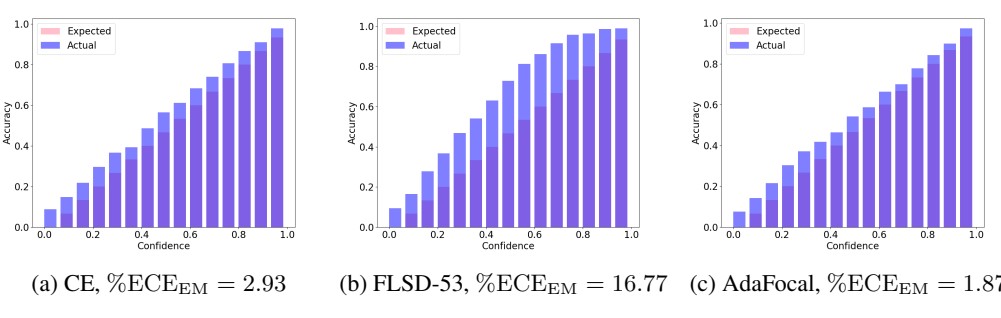

(a) CE, $\%\text{ECE}_{\text{EM}} = 2.93$    (b) FLSD-53, $\%\text{ECE}_{\text{EM}} = 16.77$    (c) AdaFocal, $\%\text{ECE}_{\text{EM}} = 1.87$

Figure 18: ImageNet, ResNet-50.

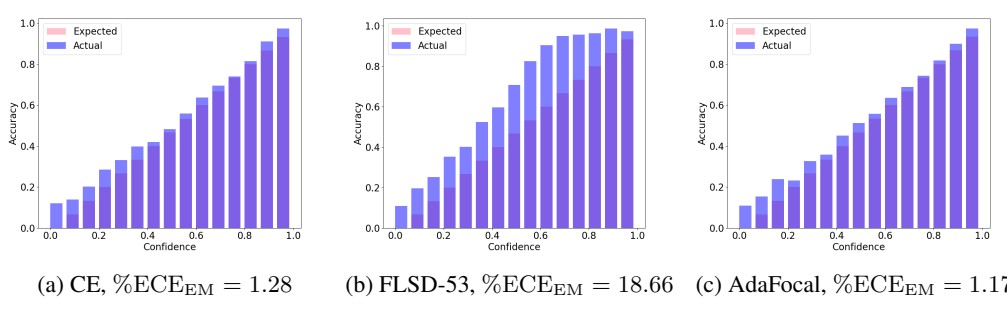

(a) CE, $\%\text{ECE}_{\text{EM}} = 1.28$     (b) FLSD-53, $\%\text{ECE}_{\text{EM}} = 18.66$     (c) AdaFocal, $\%\text{ECE}_{\text{EM}} = 1.17$

Figure 19: ImageNet, ResNet-110.

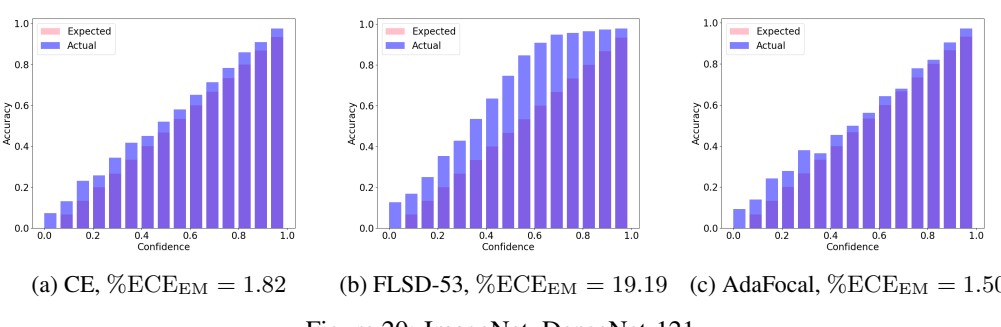

(a) CE, $\%\text{ECE}_{\text{EM}} = 1.82$     (b) FLSD-53, $\%\text{ECE}_{\text{EM}} = 19.19$     (c) AdaFocal, $\%\text{ECE}_{\text{EM}} = 1.50$

Figure 20: ImageNet, DenseNet-121.

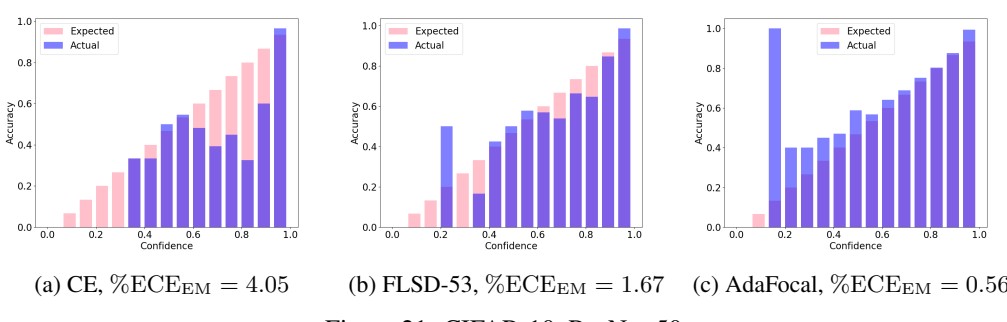

(a) CE, $\%\text{ECE}_{\text{EM}} = 4.05$     (b) FLSD-53, $\%\text{ECE}_{\text{EM}} = 1.67$     (c) AdaFocal, $\%\text{ECE}_{\text{EM}} = 0.56$

Figure 21: CIFAR-10, ResNet-50.

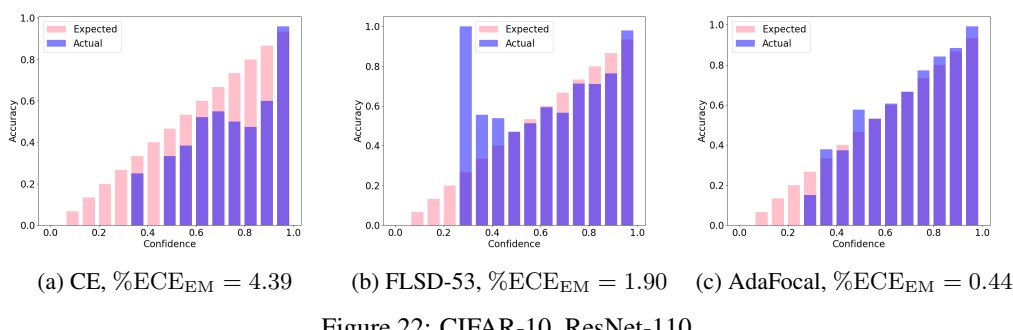

(a) CE, $\%\text{ECE}_{\text{EM}} = 4.39$     (b) FLSD-53, $\%\text{ECE}_{\text{EM}} = 1.90$     (c) AdaFocal, $\%\text{ECE}_{\text{EM}} = 0.44$

Figure 22: CIFAR-10, ResNet-110.

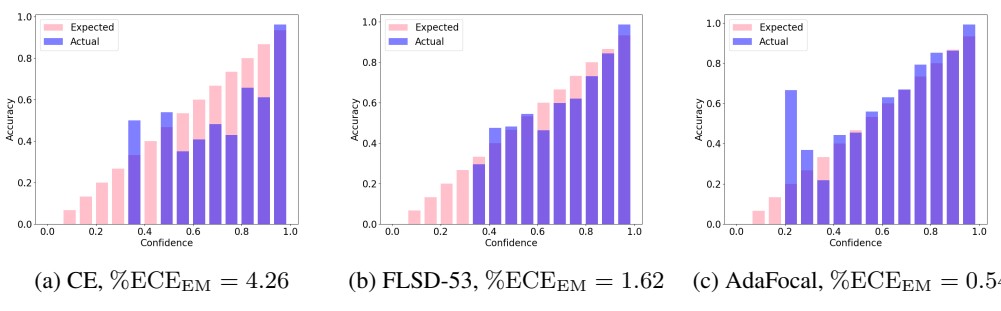

(a) CE, %ECE$_{\text{EM}}$ = 4.26    (b) FLSD-53, %ECE$_{\text{EM}}$ = 1.62    (c) AdaFocal, %ECE$_{\text{EM}}$ = 0.54

Figure 23: CIFAR-10, DenseNet-121.

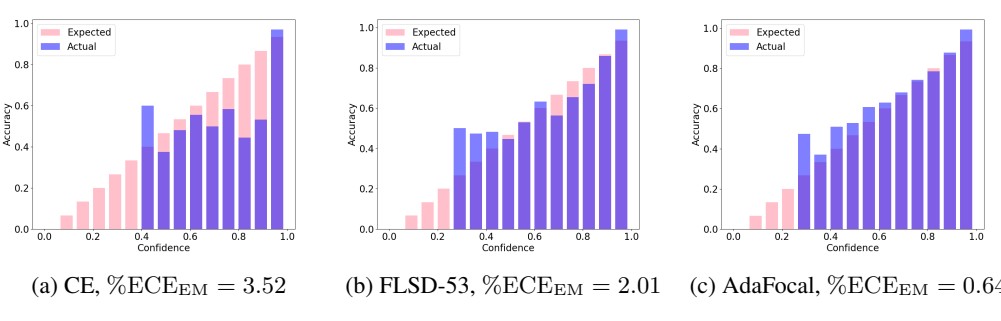

(a) CE, %ECE$_{\text{EM}}$ = 3.52    (b) FLSD-53, %ECE$_{\text{EM}}$ = 2.01    (c) AdaFocal, %ECE$_{\text{EM}}$ = 0.64

Figure 24: CIFAR-10, Wide-ResNet.

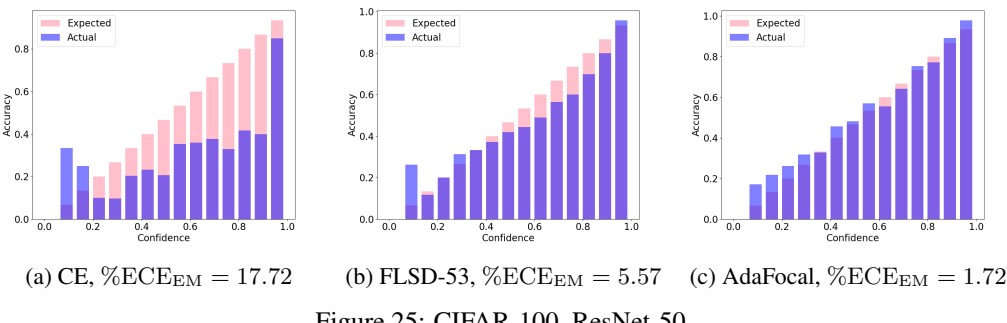

(a) CE, %ECE$_{\text{EM}}$ = 17.72    (b) FLSD-53, %ECE$_{\text{EM}}$ = 5.57    (c) AdaFocal, %ECE$_{\text{EM}}$ = 1.72

Figure 25: CIFAR-100, ResNet-50.

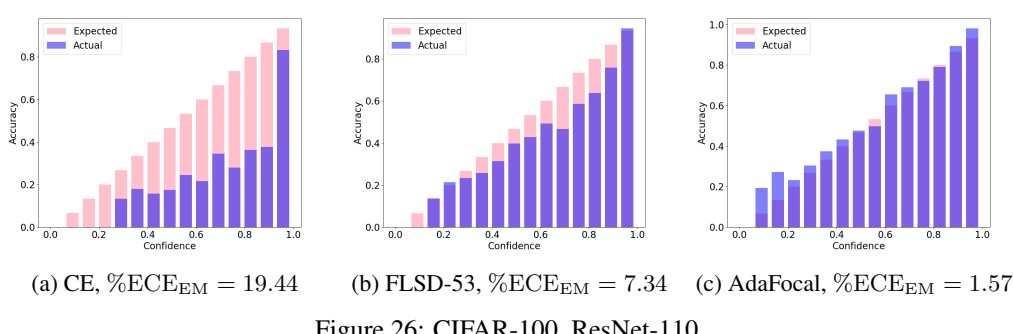

(a) CE, %ECE$_{\text{EM}}$ = 19.44    (b) FLSD-53, %ECE$_{\text{EM}}$ = 7.34    (c) AdaFocal, %ECE$_{\text{EM}}$ = 1.57

Figure 26: CIFAR-100, ResNet-110.

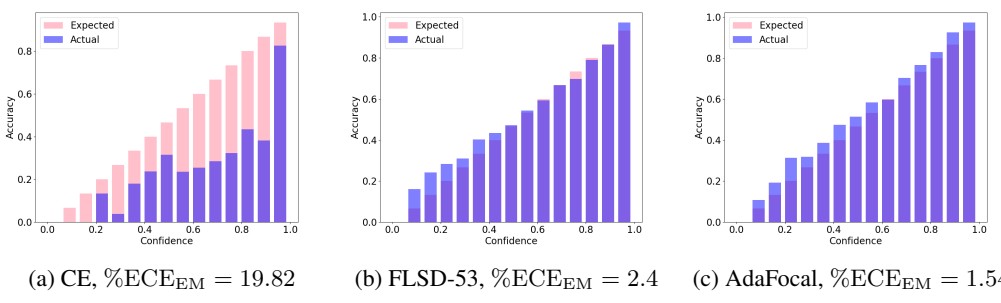

(a) CE, $\%ECE_{EM} = 19.82$     (b) FLSD-53, $\%ECE_{EM} = 2.4$     (c) AdaFocal, $\%ECE_{EM} = 1.54$

Figure 27: CIFAR-100, DenseNet-121.

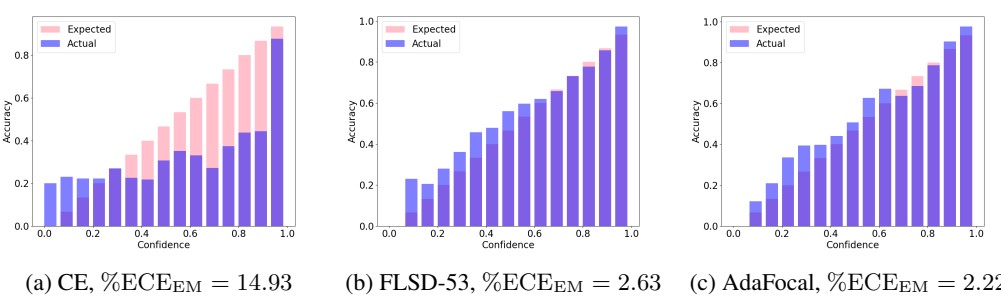

(a) CE, $\%ECE_{EM} = 14.93$     (b) FLSD-53, $\%ECE_{EM} = 2.63$     (c) AdaFocal, $\%ECE_{EM} = 2.22$

Figure 28: CIFAR-100, Wide-ResNet.

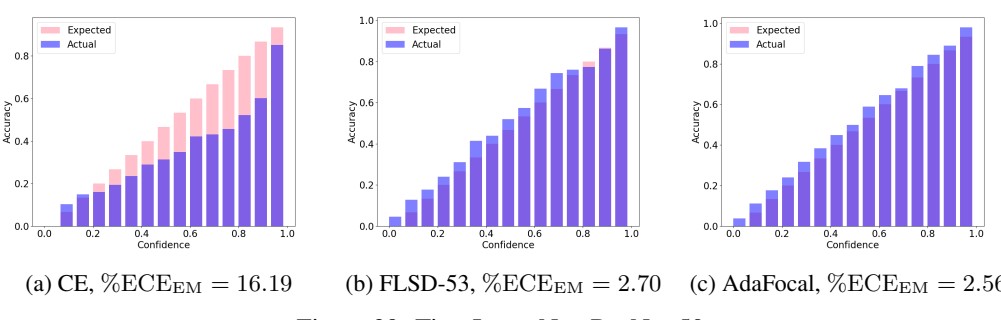

(a) CE, $\%ECE_{EM} = 16.19$     (b) FLSD-53, $\%ECE_{EM} = 2.70$     (c) AdaFocal, $\%ECE_{EM} = 2.56$

Figure 29: Tiny-ImageNet, ResNet-50.

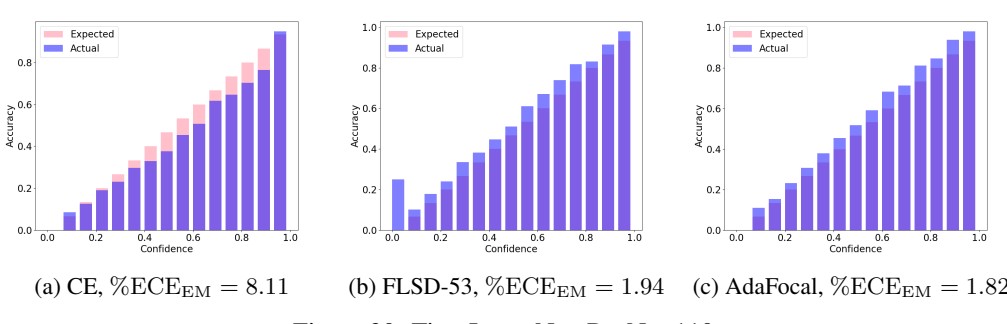

(a) CE, $\%ECE_{EM} = 8.11$     (b) FLSD-53, $\%ECE_{EM} = 1.94$     (c) AdaFocal, $\%ECE_{EM} = 1.82$

Figure 30: Tiny-ImageNet, ResNet-110.

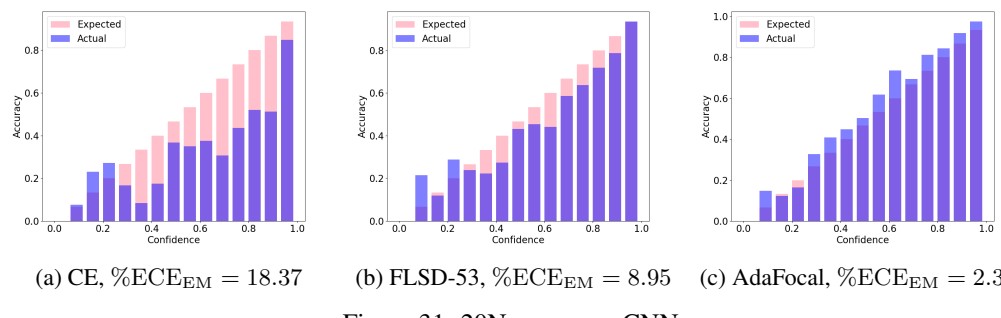

(a) CE, %ECE$_{EM}$ = 18.37    (b) FLSD-53, %ECE$_{EM}$ = 8.95    (c) AdaFocal, %ECE$_{EM}$ = 2.38

Figure 31: 20Newsgroup, CNN.

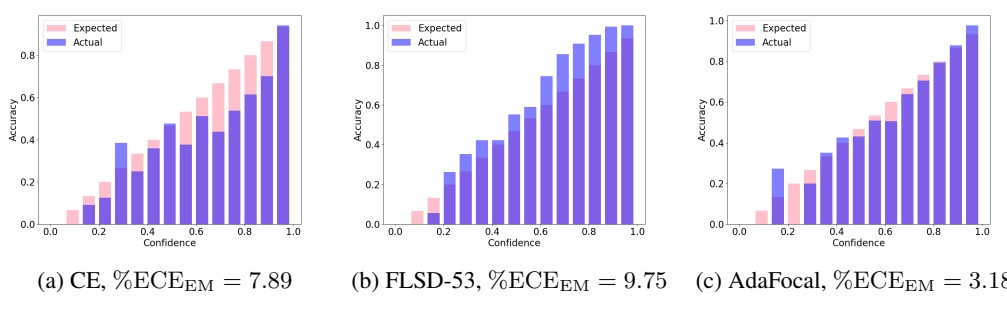

(a) CE, %ECE$_{EM}$ = 7.89    (b) FLSD-53, %ECE$_{EM}$ = 9.75    (c) AdaFocal, %ECE$_{EM}$ = 3.18

Figure 32: 20Newsgroup, BERT.

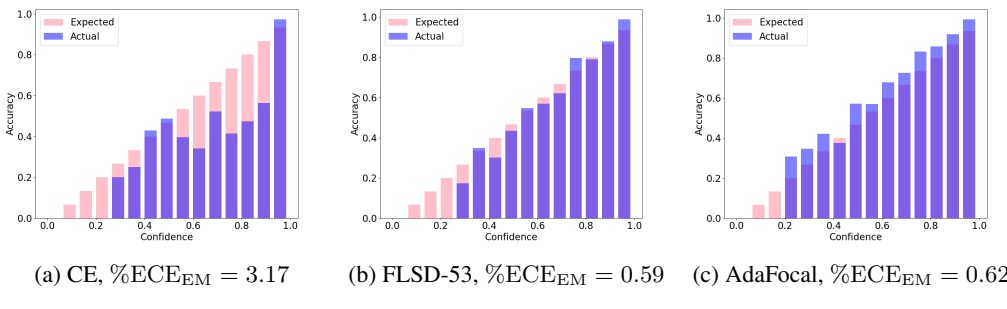

(a) CE, %ECE$_{EM}$ = 3.17    (b) FLSD-53, %ECE$_{EM}$ = 0.59    (c) AdaFocal, %ECE$_{EM}$ = 0.62

Figure 33: SVHN, ResNet-110.

## M    "Calibrate-able" Property

In Fig. 34, following [30], for ResNet-50 on CIFAR-10 and ResNet-50 on CIFAR-100, we plot the distribution of max-logits of training examples at the end of training (i.e. at epoch 350) grouped as per different "learned epochs".

**Observations**:

- Unlike for ResNet-32 in [30], we do not find cross entropy + Temperature Scaling to be better than focal loss + temperature scaling for ResNet-50 (the same is observed in [19]).

- Although the distribution of FLSD-53 is compressed, similar to what observed for focal loss in [30], the separation of samples is not seen here for cross entropy (CE).

- For AdaFocal, we see better separation of easy and hard examples grouped as per their "learned epoch". This makes AdaFocal more "calibrate-able" as per [30].

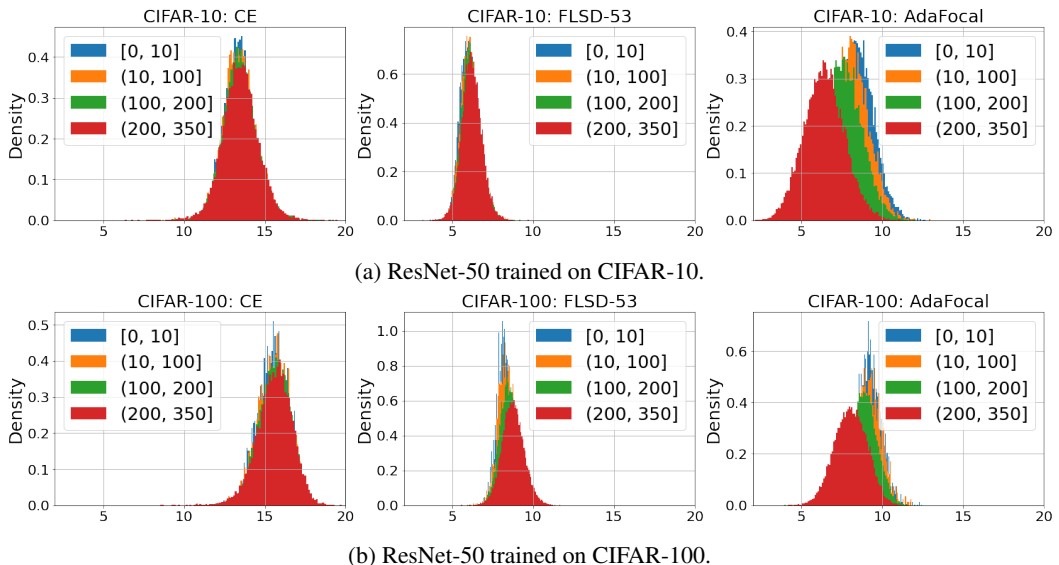

(a) ResNet-50 trained on CIFAR-10.

(b) ResNet-50 trained on CIFAR-100.

Figure 34: Comparison of the distribution of max-logits of training examples grouped as per different "learned epochs". In the legend, the groups are marked by intervals to which the "learned epoch" belongs.

# N   Multiple runs of AdaFocal with different $\gamma_{\max}$

Due to the stochastic nature of the experiments, AdaFocal $\gamma$s may end up following different trajectories across different runs (initialization), which in turn might lead to variations in the final results. In this section, we look at the extent of such variations for ResNet-50 trained on CIFAR-10 for $\gamma_{\max} = 20$, $\gamma_{\max} = 50$ and unconstrained $\gamma$ ($\gamma_{\max} = \infty$). For all these experiments, the minimum $\gamma$ for inverse-focal loss is set to $\gamma_{\min} = -2$ and the switching threshold is set to $S_{th} = 0.2$.

## N.1   AdaFocal $\gamma_{\max} = 20$

In Fig. 35, we observe that AdaFocal with $\gamma_{\max} = 20$ is consistently better than FLSD-53. Fig. 36 shows the variation in dynamics of $\gamma$ during training across different runs.

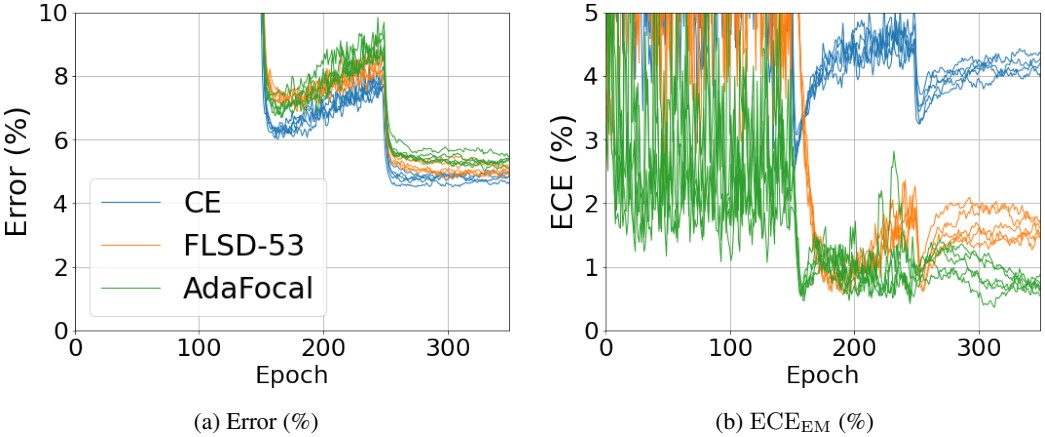

(a) Error (%)

(b) $\text{ECE}_{\text{EM}}$ (%)

Figure 35: Multiple runs of ResNet-50 trained on CIFAR-10 using cross entropy (CE), FLSD-53 and AdaFocal with $\gamma_{\max} = 20$ for different initialization seeds.

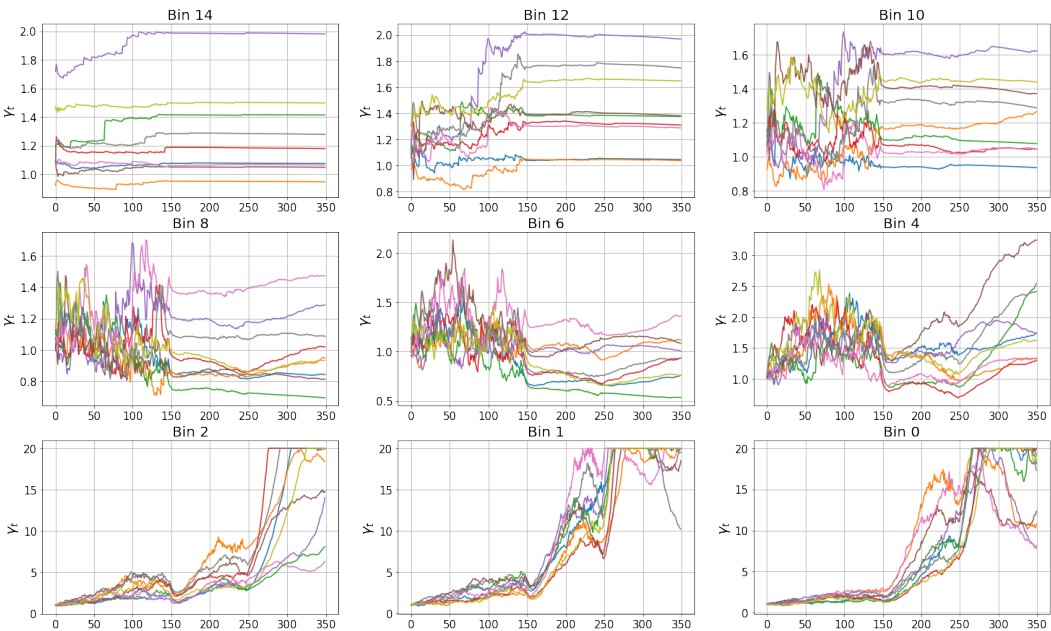

Figure 36: Dynamics of $\gamma_t$ for different runs of ResNet-50 trained on CIFAR-10 using AdaFocal $\gamma_{\max} = 20$.

## N.2 AdaFocal $\gamma_{\max} = 50$

In Fig. 37, we observe that AdaFocal with $\gamma_{\max} = 50$ has more variability than AdaFocal $\gamma_{\max} = 20$ but is mostly better than FLSD-53. Fig. 37 shows the variation in dynamics of $\gamma$ during training across different runs.

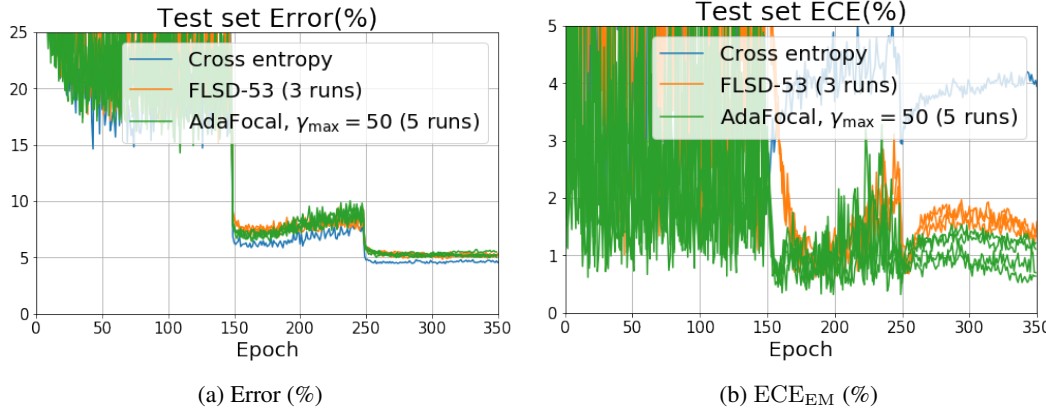

(a) Error (%)  (b) $\text{ECE}_{\text{EM}}$ (%)

Figure 37: Plots for ResNet-50 trained on CIFAR-10 using cross entropy (1 run), FLSD-53 (3 runs) and AdaFocal with $\gamma_{\max} = 50$ (5 runs). AdaFocal $\gamma_{\max} = 50$, although mostly better than FLSD-53, does exhibit greater variability than AdaFocal $\gamma_{\max} = 20$.

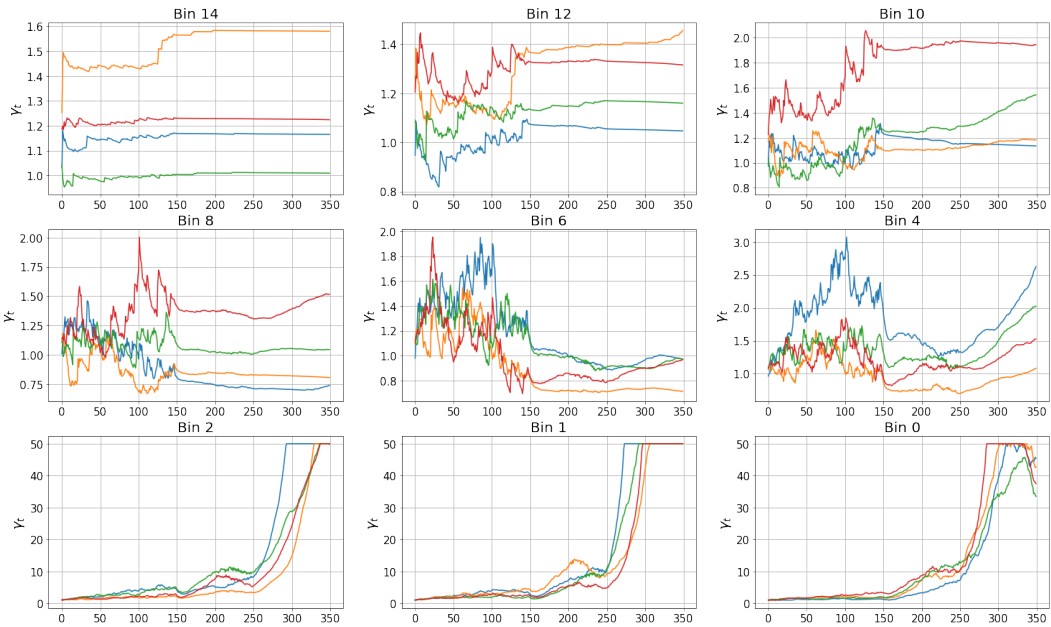

Figure 38: Dynamics of $\gamma_t$ for different runs of ResNet-50 trained on CIFAR-10 using AdaFocal $\gamma_{\max} = 50$.

### N.3    AdaFocal, unconstrained $\gamma$ ($\gamma_{\max} = \infty$)

In Fig. 39, we observe that AdaFocal with unconstrained $\gamma$ exhibit grater variability across different runs: 7 out of 9 times it performs better than FLSD-53 whereas the other two times it is similar or slightly worse.

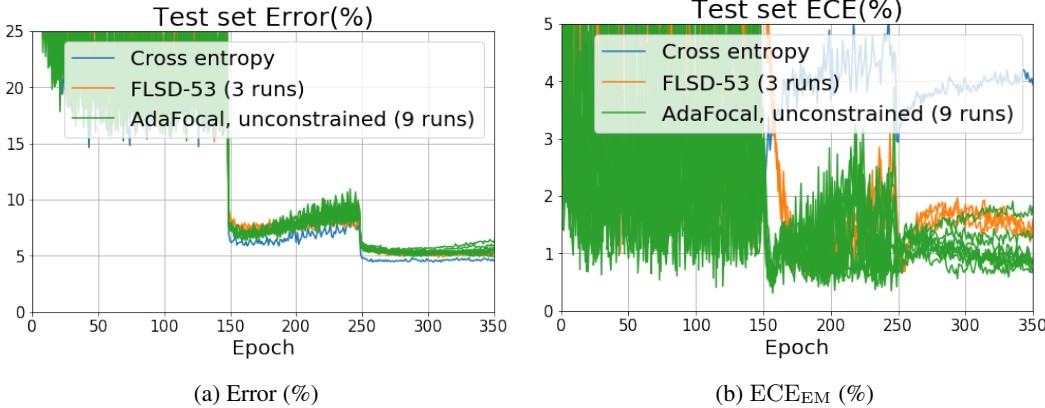

(a) Error (%)                (b) $\text{ECE}_{\text{EM}}$ (%)

Figure 39: Plots for ResNet-50 trained on CIFAR-10 using cross entropy (1 run), FLSD-53 (3 runs) and AdaFocal $\gamma_{\max} = \infty$ (9 runs). AdaFocal with unconstrained $\gamma$ exhibits much greater variability across different runs than $\gamma_{\max} = 20$ and $\gamma_{\max} = 50$.

The above behaviour is mostly due to large variations in the trajectory of $\gamma$s especially for lower bins as shown in Fig. 40. For higher bins, $\gamma$s do not explode and settle to similar nearby values, whereas, for lower bins, as the $\gamma$s are unconstrained they blow up to udesirably high values.

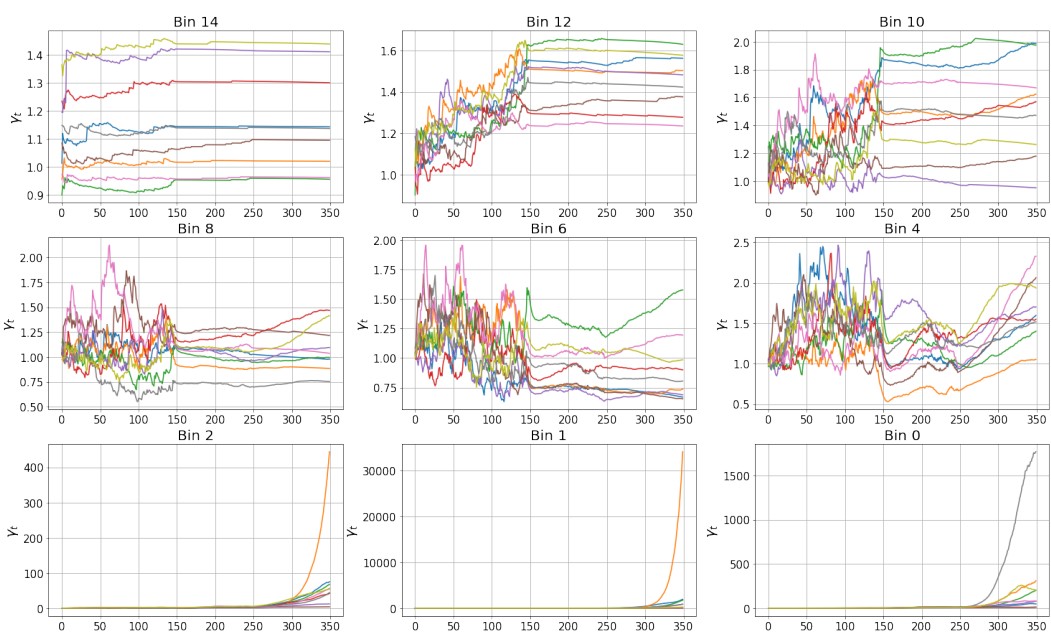

Figure 40: Dynamics of $\gamma_t$ for different runs of ResNet-50 trained on CIFAR-10 using unconstrained AdaFocal with $\gamma_{\max} = \infty$.

# O   Error, ECE, dynamics of $\gamma$, and bin statistics during training

## O.1   ImageNet, ResNet-50

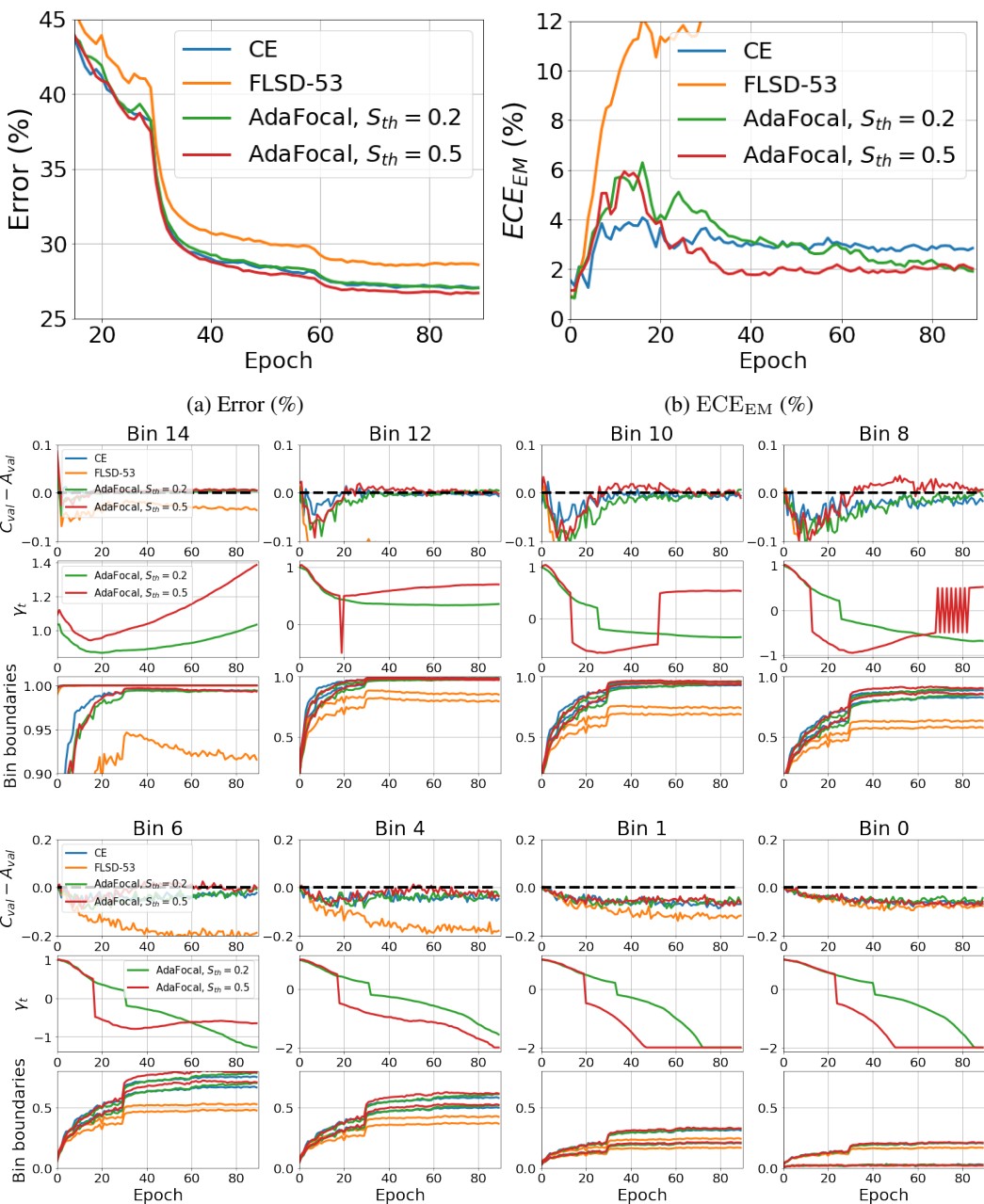

(a) Error (%)

(b) $\text{ECE}_{\text{EM}}$ (%)

(c) Dynamics of $\gamma$ and calibration behaviour in different bins. Each bin has three subplots: **top**: $E_{val,i} = C_{val,i} - A_{val,i}$, **middle**: evolution of $\gamma_t$, and **bottom**: bin boundaries. Black dashed line in top plot represent zero calibration error.

Figure 41: ResNet-50 trained on ImageNet with cross entropy (CE), FLSD-53, and AdaFocal with $S_{th} = 0.2$ and $0.5$.

## O.2 ImageNet, ResNet-110

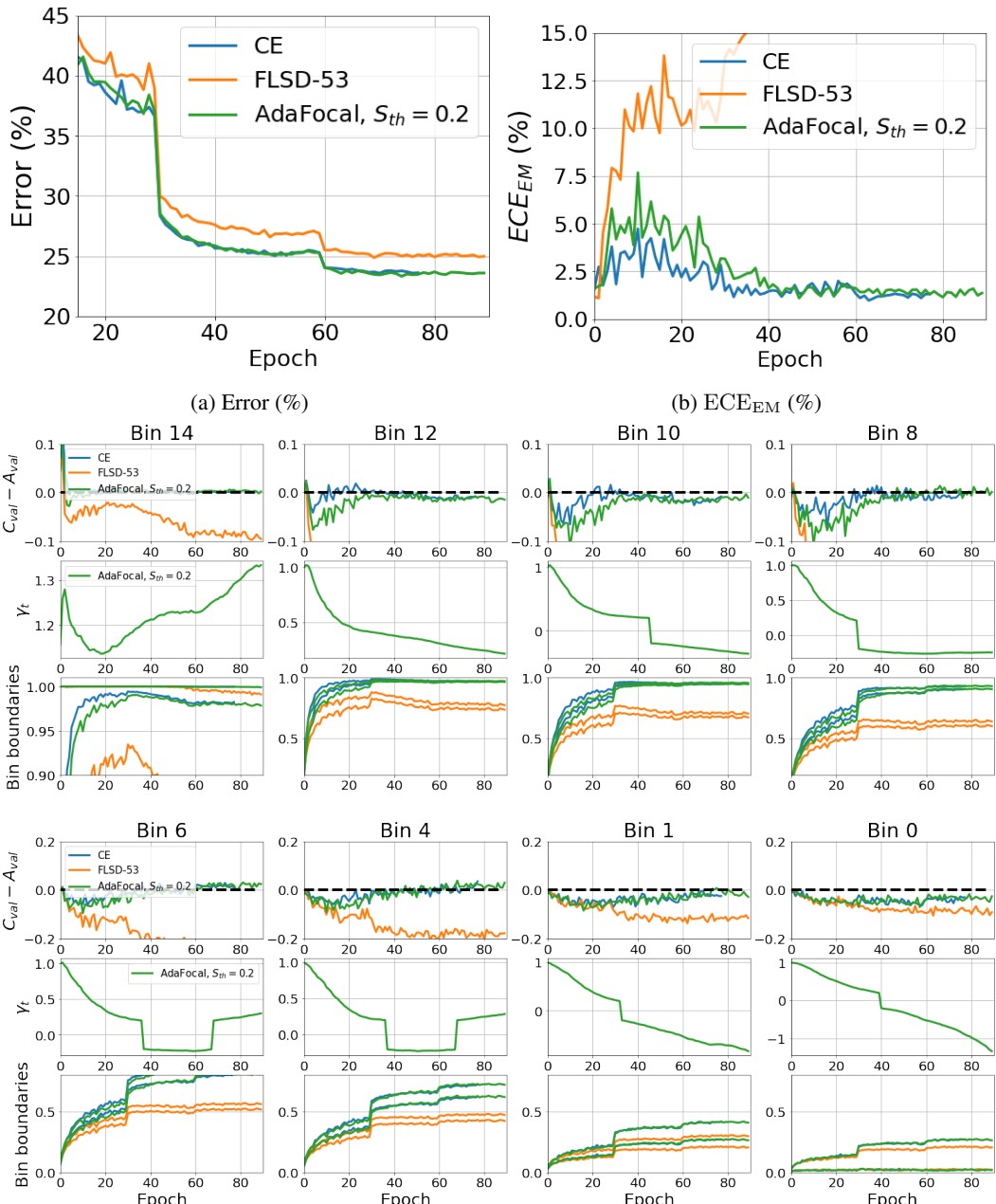

(a) Error (%)

(b) ECE$_{EM}$ (%)

(c) Dynamics of $\gamma$ and calibration behaviour in different bins. Each bin has three subplots: **top**: $E_{val,i} = C_{val,i} - A_{val,i}$, **middle**: evolution of $\gamma_t$, and **bottom**: bin boundaries. Black dashed line in top plot represent zero calibration error.

Figure 42: ResNet-110 trained on ImageNet with cross entropy (CE), FLSD-53, and AdaFocal with $S_{th} = 0.2$.

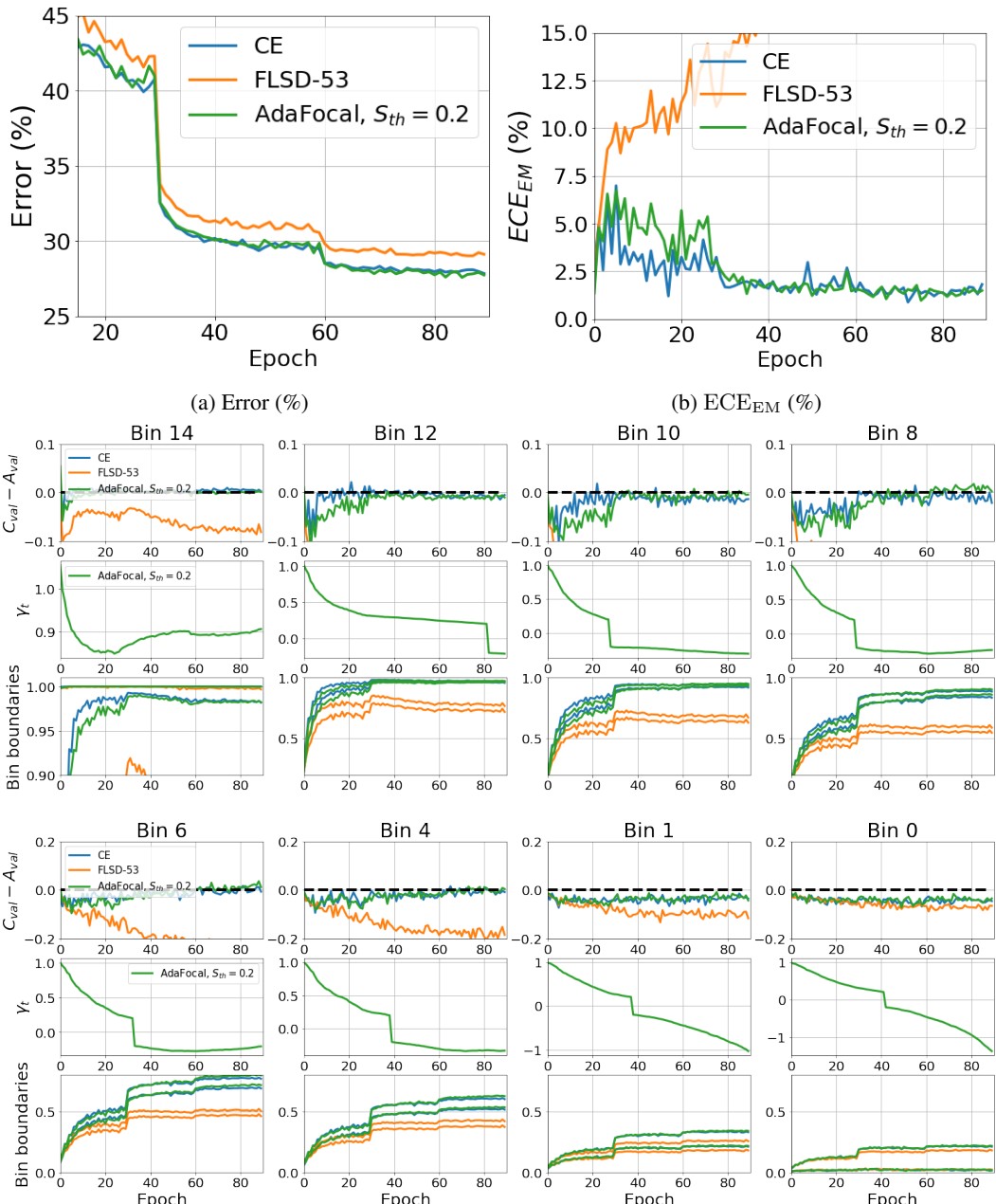

(a) Error (%)

(b) $\text{ECE}_{\text{EM}}$ (%)

(c) Dynamics of $\gamma$ and calibration behaviour in different bins. Each bin has three subplots: **top**: $E_{val,i} = C_{val,i} - A_{val,i}$, **middle**: evolution of $\gamma_t$, and **bottom**: bin boundaries. Black dashed line in top plot represent zero calibration error.

Figure 43: DenseNet-121 trained on ImageNet with cross entropy (CE), FLSD-53, and AdaFocal with $S_{th} = 0.2$.

## O.4 CIFAR-10, ResNet-110

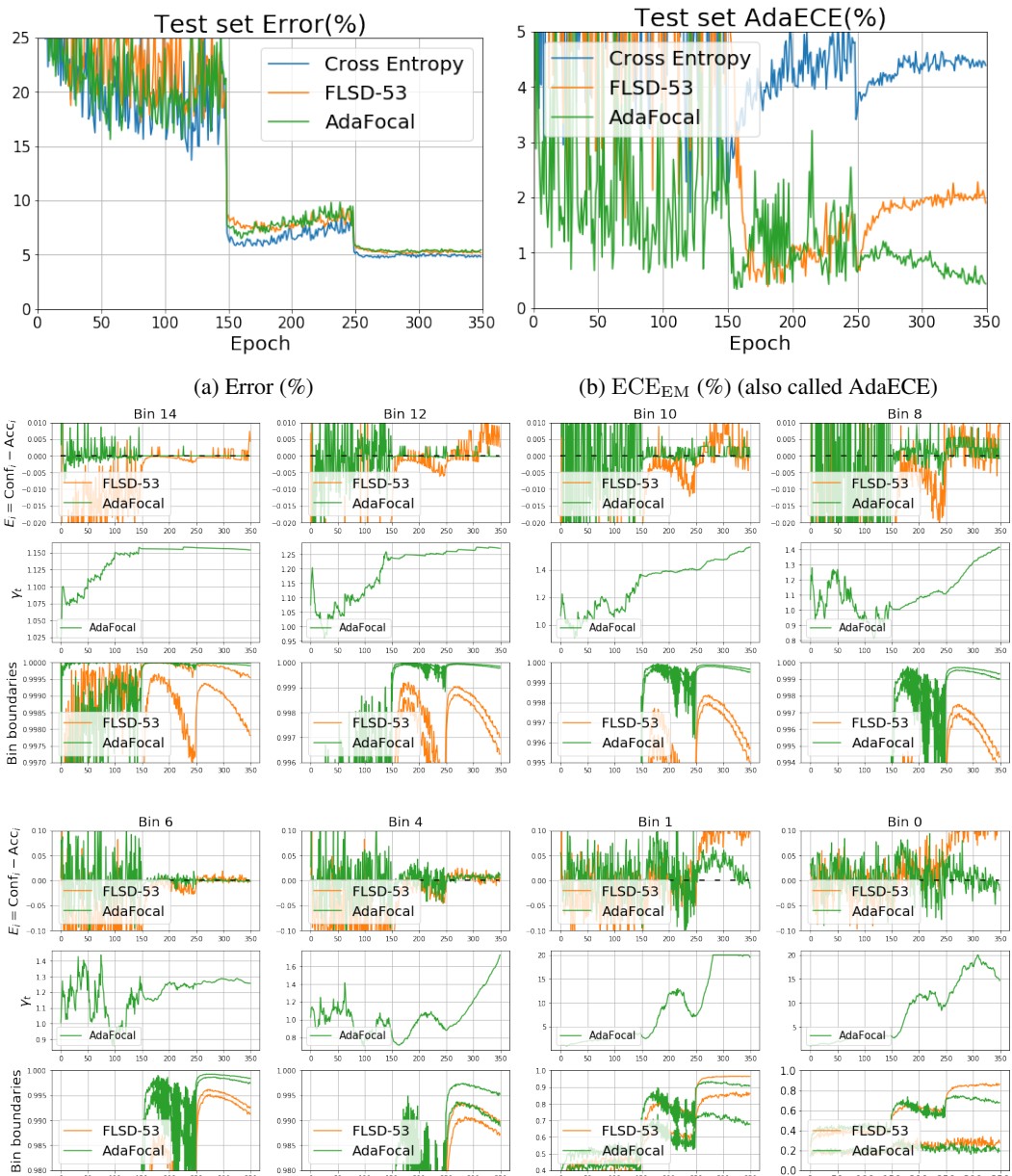

(a) Error (%)

(b) ECE$_{EM}$ (%) (also called AdaECE)

(c) Dynamics of $\gamma$ and calibration behaviour in different bins. Each bin has three subplots: **top**: $E_{val,i} = C_{val,i} - A_{val,i}$, **middle**: evolution of $\gamma_t$, and **bottom**: bin boundaries. Black dashed line in top plot represent zero calibration error.

Figure 44: ResNet-110 trained on CIFAR-10 with cross entropy (CE), FLSD-53, and AdaFocal.

## O.5 CIFAR-10, Wide-ResNet

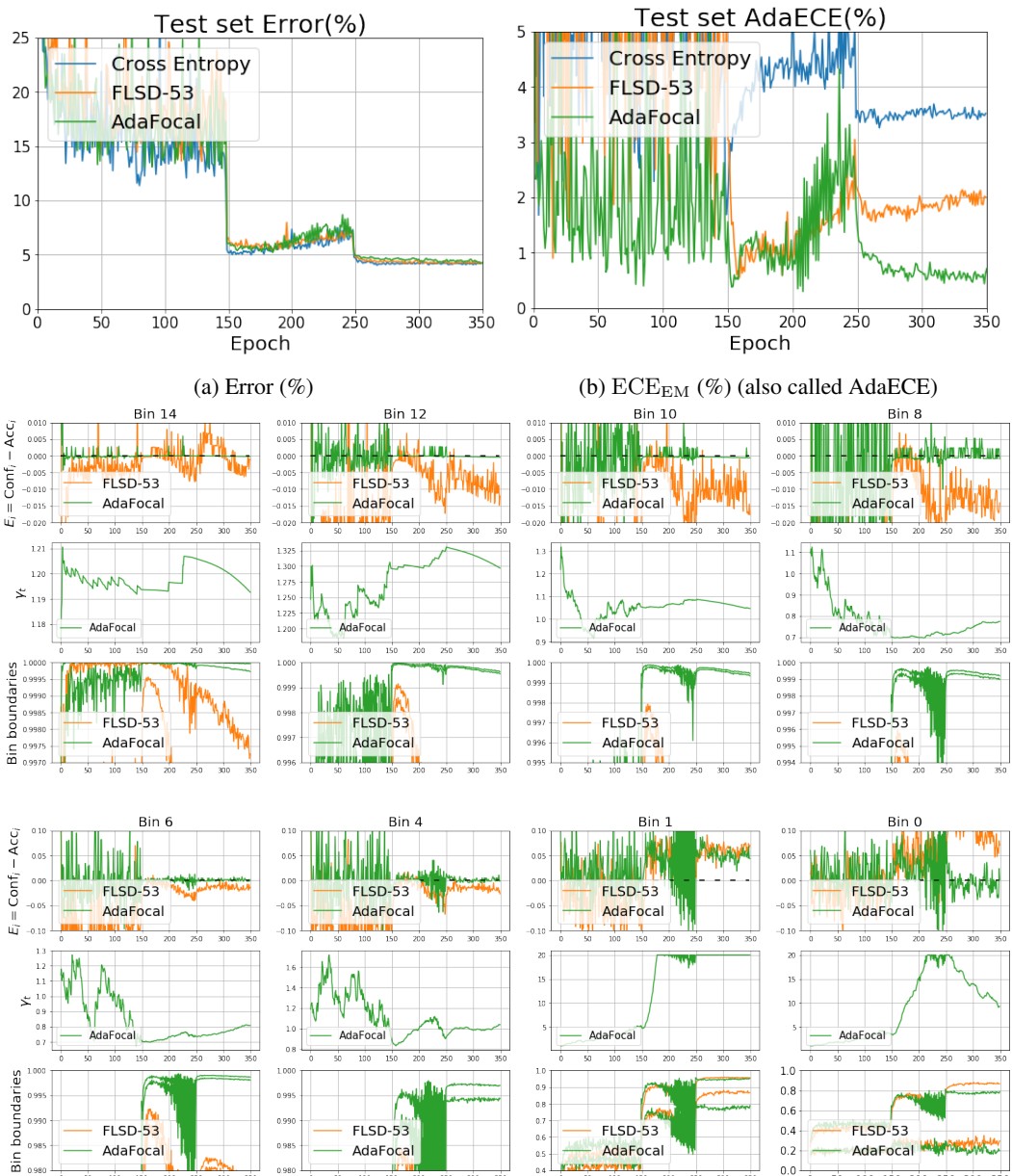

(a) Error (%)

(b) $\text{ECE}_{\text{EM}}$ (%) (also called AdaECE)

(c) Dynamics of $\gamma$ and calibration behaviour in different bins. Each bin has three subplots: **top**: $E_{val,i} = C_{val,i} - A_{val,i}$, **middle**: evolution of $\gamma_t$, and **bottom**: bin boundaries. Black dashed line in top plot represent zero calibration error.

Figure 45: Wide-ResNet trained on CIFAR-10 with cross entropy (CE), FLSD-53, and AdaFocal.

## O.6 CIFAR-10, DenseNet-121

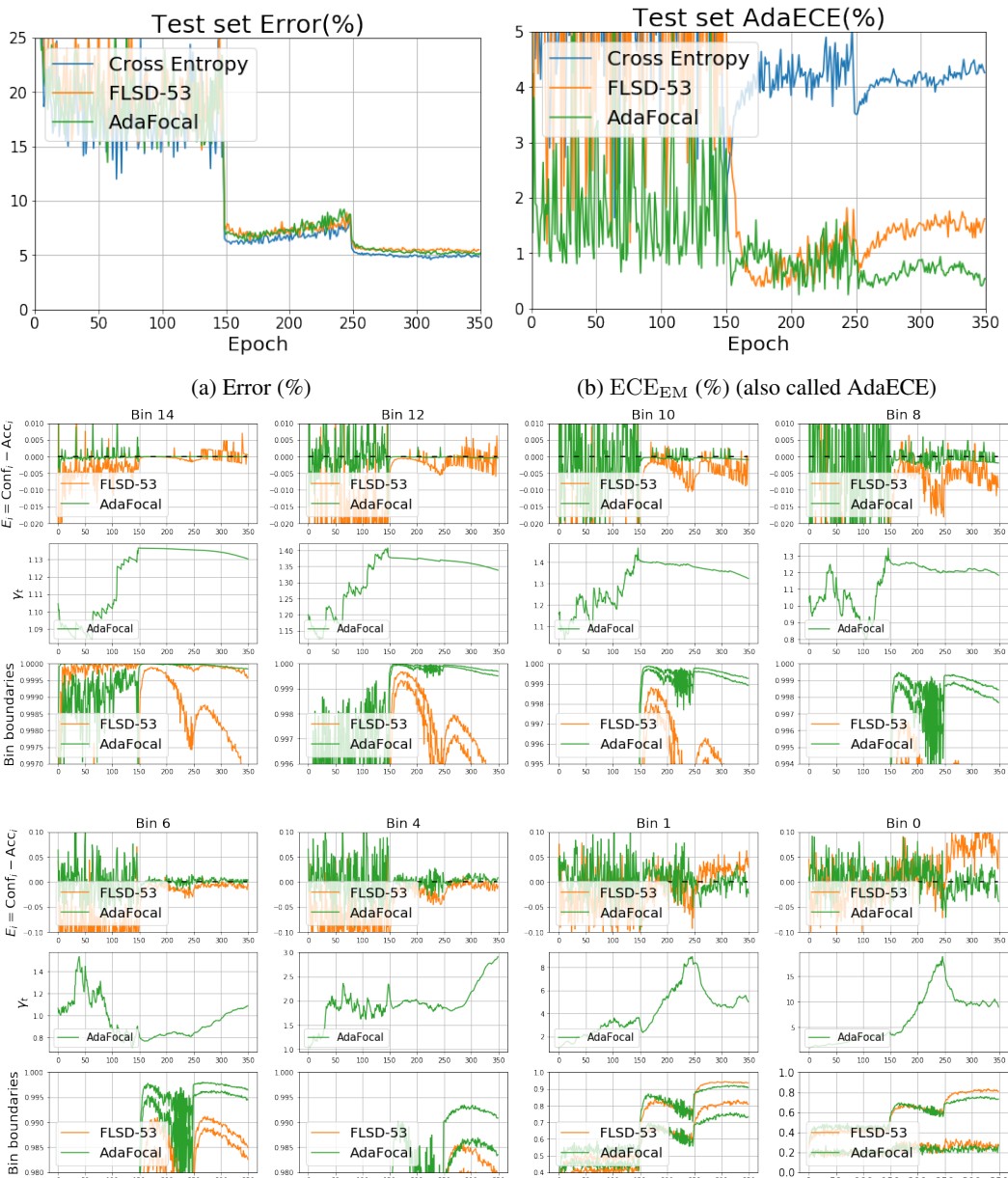

(a) Error (%)

(b) $ECE_{EM}$ (%) (also called AdaECE)

(c) Dynamics of $\gamma$ and calibration behaviour in different bins. Each bin has three subplots: **top**: $E_{val,i} = C_{val,i} - A_{val,i}$, **middle**: evolution of $\gamma_t$, and **bottom**: bin boundaries. Black dashed line in top plot represent zero calibration error.

Figure 46: DenseNet-121 trained on CIFAR-10 with cross entropy (CE), FLSD-53, and AdaFocal.

## O.7 CIFAR-100, ResNet-50

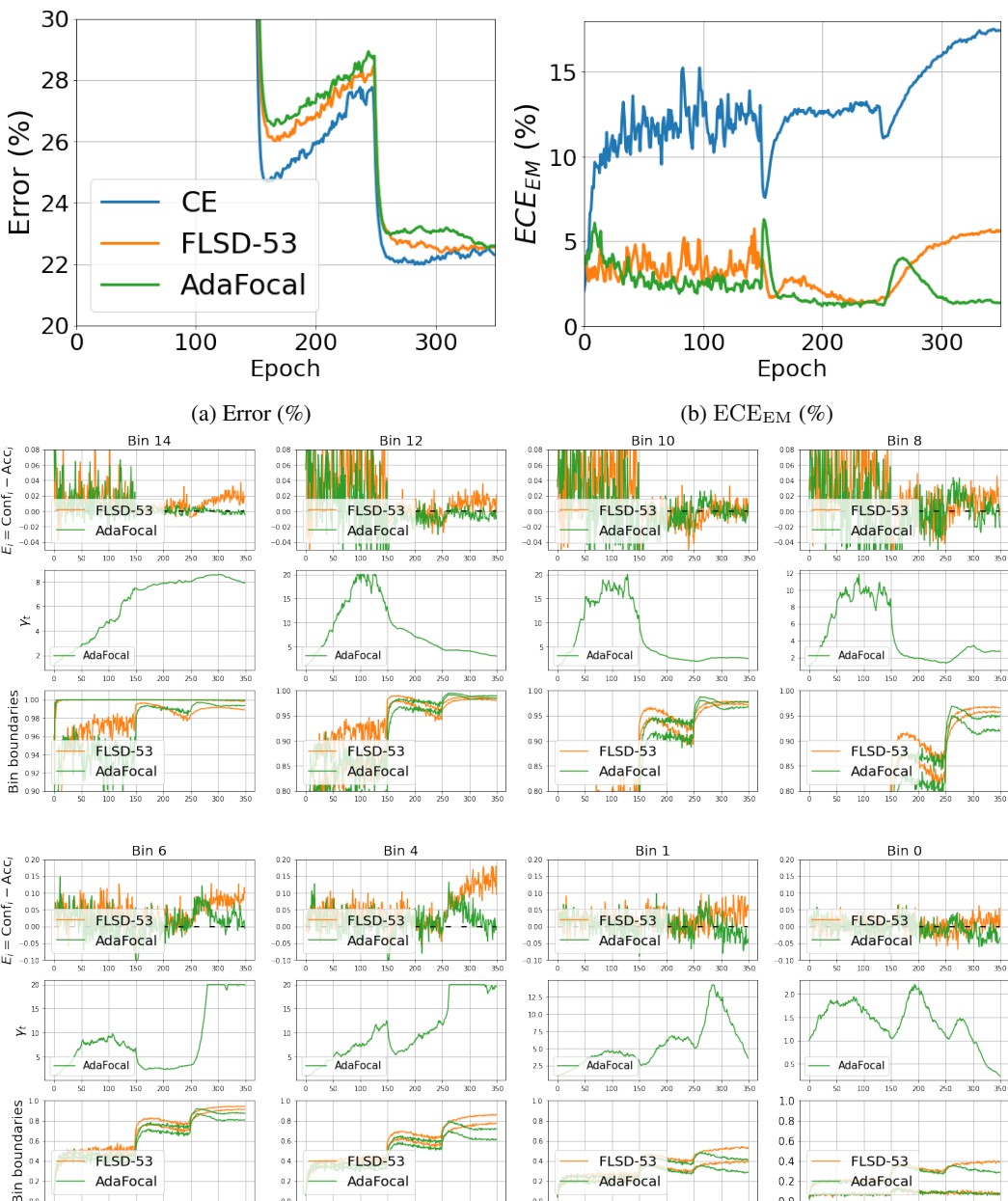

(a) Error (%)

(b) $\text{ECE}_{\text{EM}}$ (%)

(c) Dynamics of $\gamma$ and calibration behaviour in different bins. Each bin has three subplots: **top**: $E_{val,i} = C_{val,i} - A_{val,i}$, **middle**: evolution of $\gamma_t$, and **bottom**: bin boundaries. Black dashed line in top plot represent zero calibration error.

Figure 47: ResNet-50 trained on CIFAR-100 with cross entropy (CE), FLSD-53, and AdaFocal.

## O.8 CIFAR-100, ResNet-110

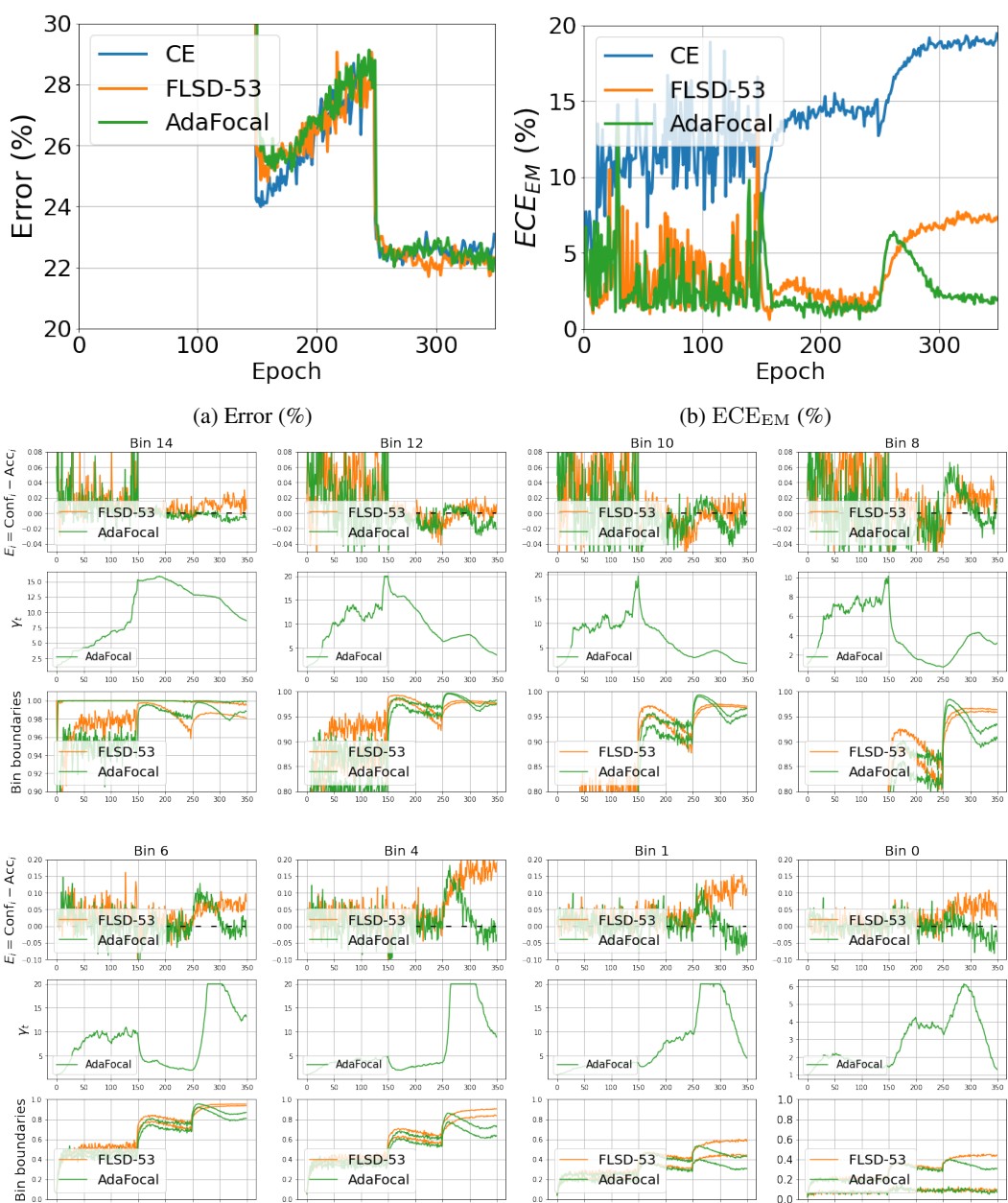

(a) Error (%)          (b) $ECE_{EM}$ (%)

(c) Dynamics of $\gamma$ and calibration behaviour in different bins. Each bin has three subplots: **top**: $E_{val,i} = C_{val,i} - A_{val,i}$, **middle**: evolution of $\gamma_t$, and **bottom**: bin boundaries. Black dashed line in top plot represent zero calibration error.

Figure 48: ResNet-110 trained on CIFAR-100 with cross entropy (CE), FLSD-53, and AdaFocal.

## O.9 CIFAR-100, Wide-ResNet

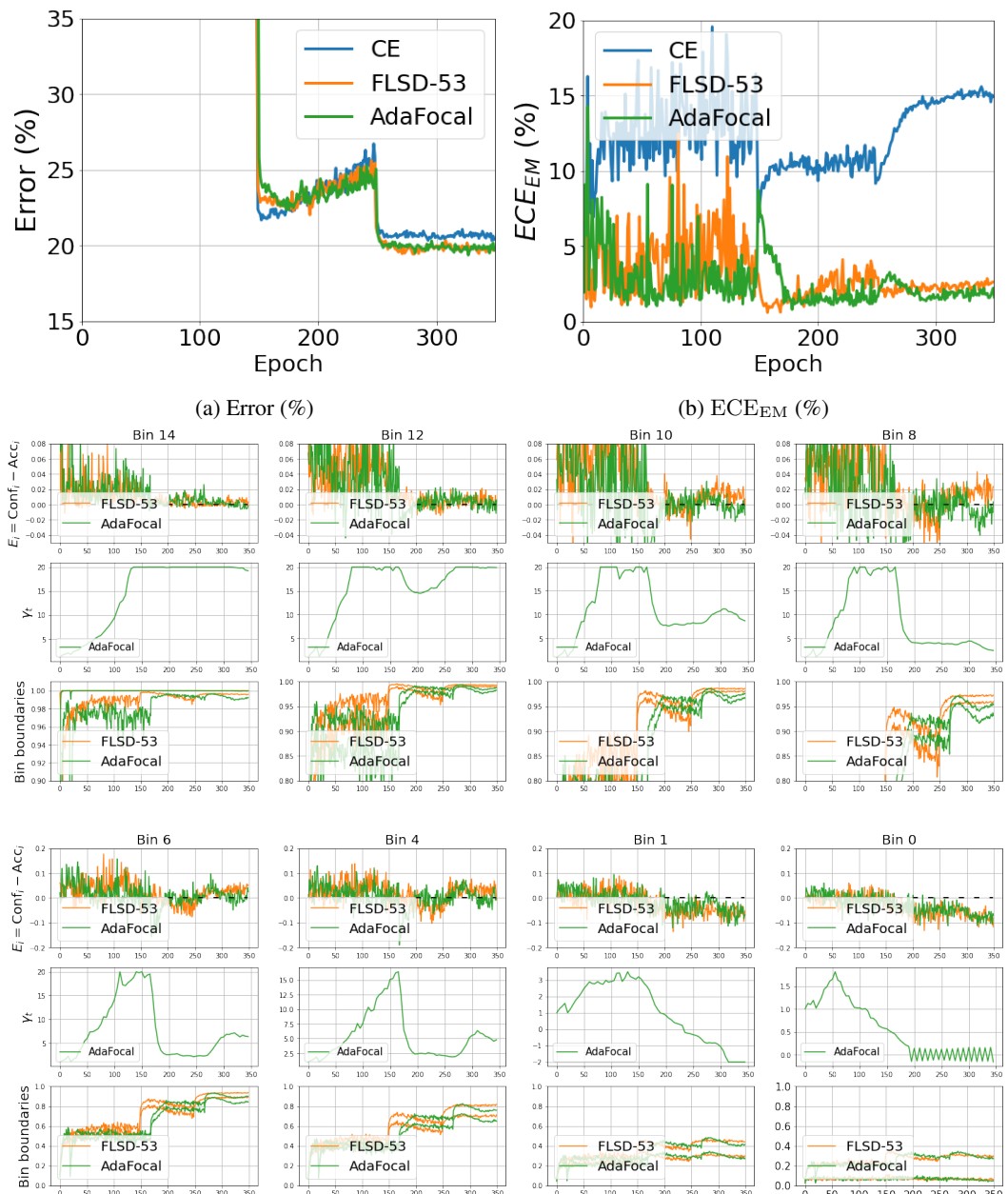

(a) Error (%)

(b) ECE$_{EM}$ (%)

(c) Dynamics of $\gamma$ and calibration behaviour in different bins. Each bin has three subplots: **top**: $E_{val,i} = C_{val,i} - A_{val,i}$, **middle**: evolution of $\gamma_t$, and **bottom**: bin boundaries. Black dashed line in top plot represent zero calibration error.

Figure 49: Wide-ResNet trained on CIFAR-100 with cross entropy (CE), FLSD-53, and AdaFocal.

## O.10    CIFAR-100, DenseNet-121

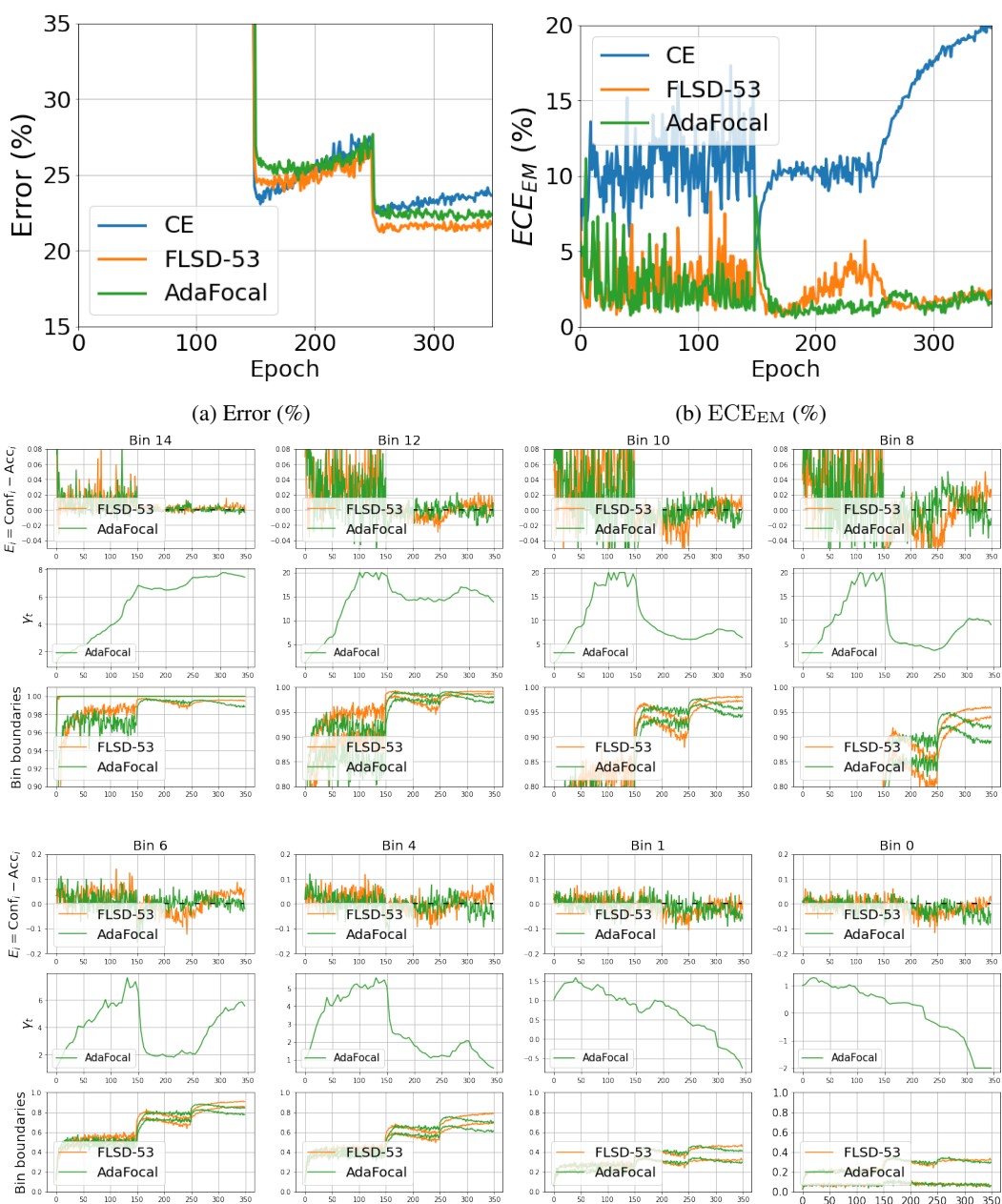

(a) Error (%)

(b) ECE_EM (%)

(c) Dynamics of $\gamma$ and calibration behaviour in different bins. Each bin has three subplots: **top**: $E_{val,i} = C_{val,i} - A_{val,i}$, **middle**: evolution of $\gamma_t$, and **bottom**: bin boundaries. Black dashed line in top plot represent zero calibration error.

Figure 50: DenseNet-121 trained on CIFAR-100 with cross entropy (CE), FLSD-53, and AdaFocal.

## O.11 Tiny-ImageNet, ResNet-50

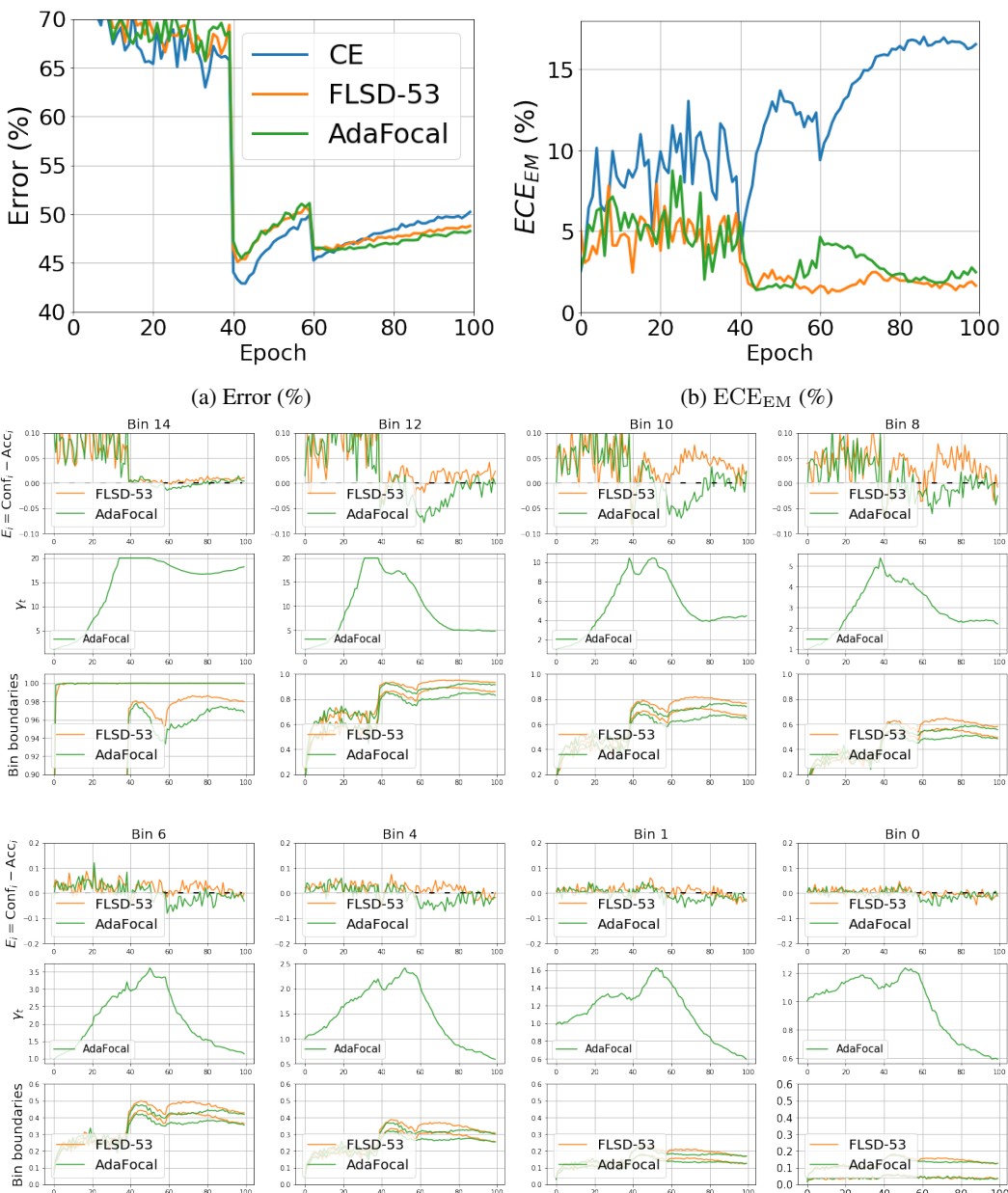

(a) Error (%)

(b) $ECE_{EM}$ (%)

(c) Dynamics of $\gamma$ and calibration behaviour in different bins. Each bin has three subplots: **top**: $E_{val,i} = C_{val,i} - A_{val,i}$, **middle**: evolution of $\gamma_t$, and **bottom**: bin boundaries. Black dashed line in top plot represent zero calibration error.

Figure 51: ResNet-50 trained on Tiny-ImageNet with cross entropy (CE), FLSD-53, and AdaFocal.

## O.12 Tiny-ImageNet, ResNet-110

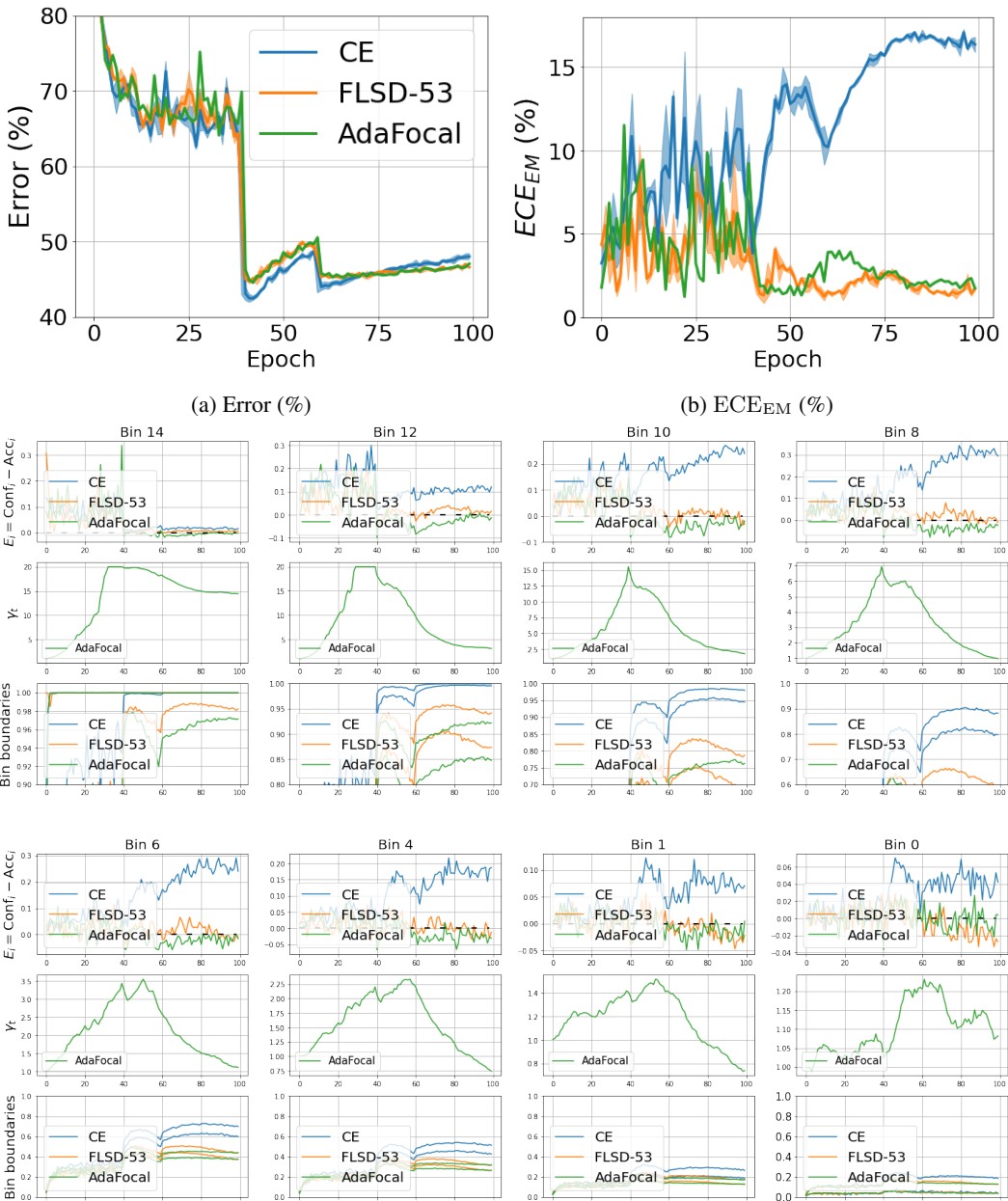

(a) Error (%)

(b) ECE$_{EM}$ (%)

(c) Dynamics of $\gamma$ and calibration behaviour in different bins. Each bin has three subplots: **top**: $E_{val,i} = C_{val,i} - A_{val,i}$, **middle**: evolution of $\gamma_t$, and **bottom**: bin boundaries. Black dashed line in top plot represent zero calibration error.

Figure 52: ResNet-110 trained on Tiny-ImageNet with cross entropy (CE), FLSD-53, and AdaFocal.

## O.13 SVHN, ResNet-110

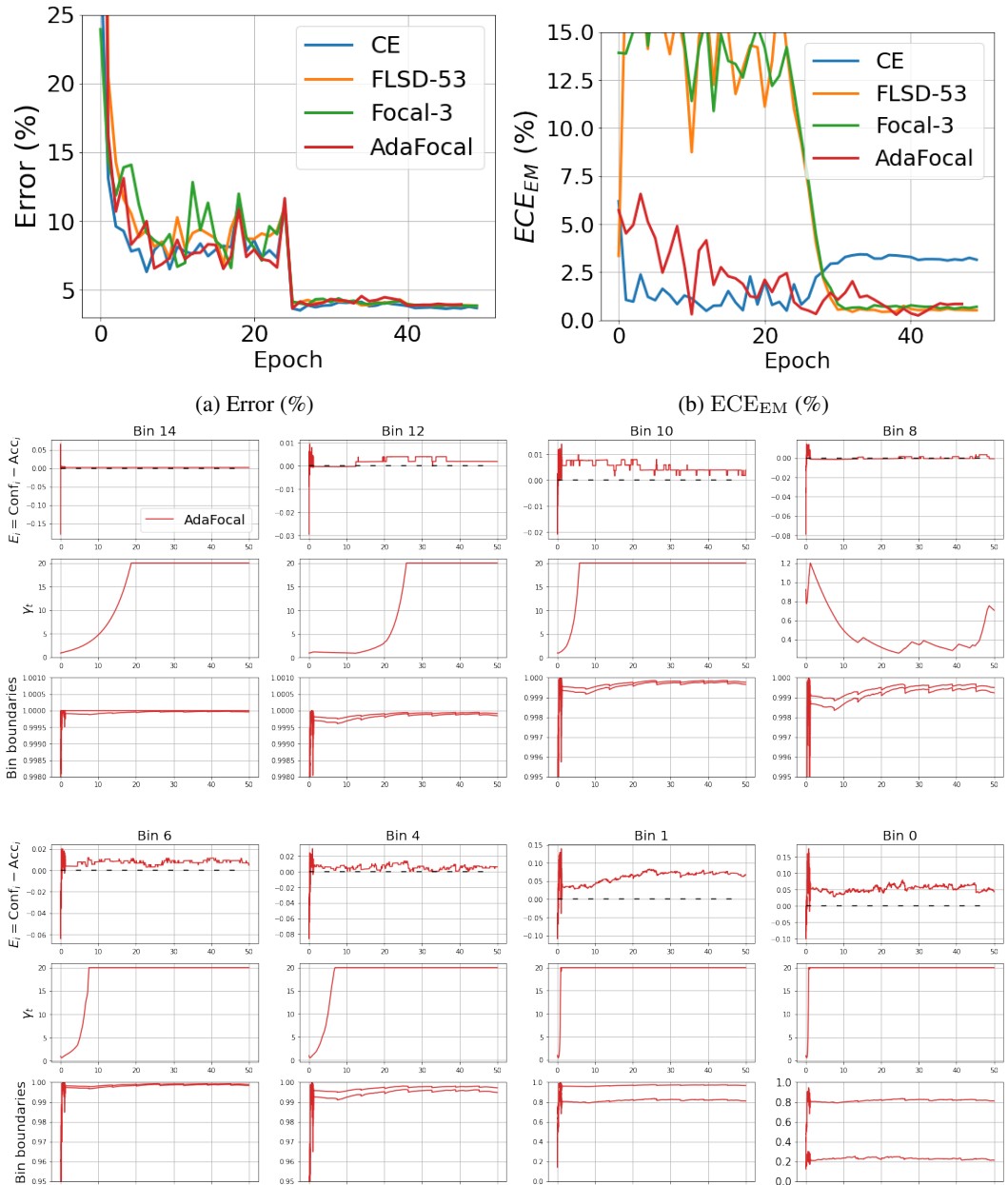

(a) Error (%)

(b) ECE$_{EM}$ (%)

(c) Dynamics of $\gamma$ and calibration behaviour in different bins. Each bin has three subplots: **top**: $E_{val,i} = C_{val,i} - A_{val,i}$, **middle**: evolution of $\gamma_t$, and **bottom**: bin boundaries. Black dashed line in top plot represent zero calibration error.

Figure 53: ResNet-110 trained on SVHN with cross entropy (CE), FLSD-53, AdaFocal, AdaFocal-shcedule. In AdaFocal-schedule, for the first 25 epochs $\gamma$ is updated every epoch, from epoch 25 to 40, $\gamma$ is updated every 100 mini-batches, and from epoch 40 to 50 (end of training), $\gamma$ is updated every mini-batch.

## O.14 20 Newsgroups, CNN

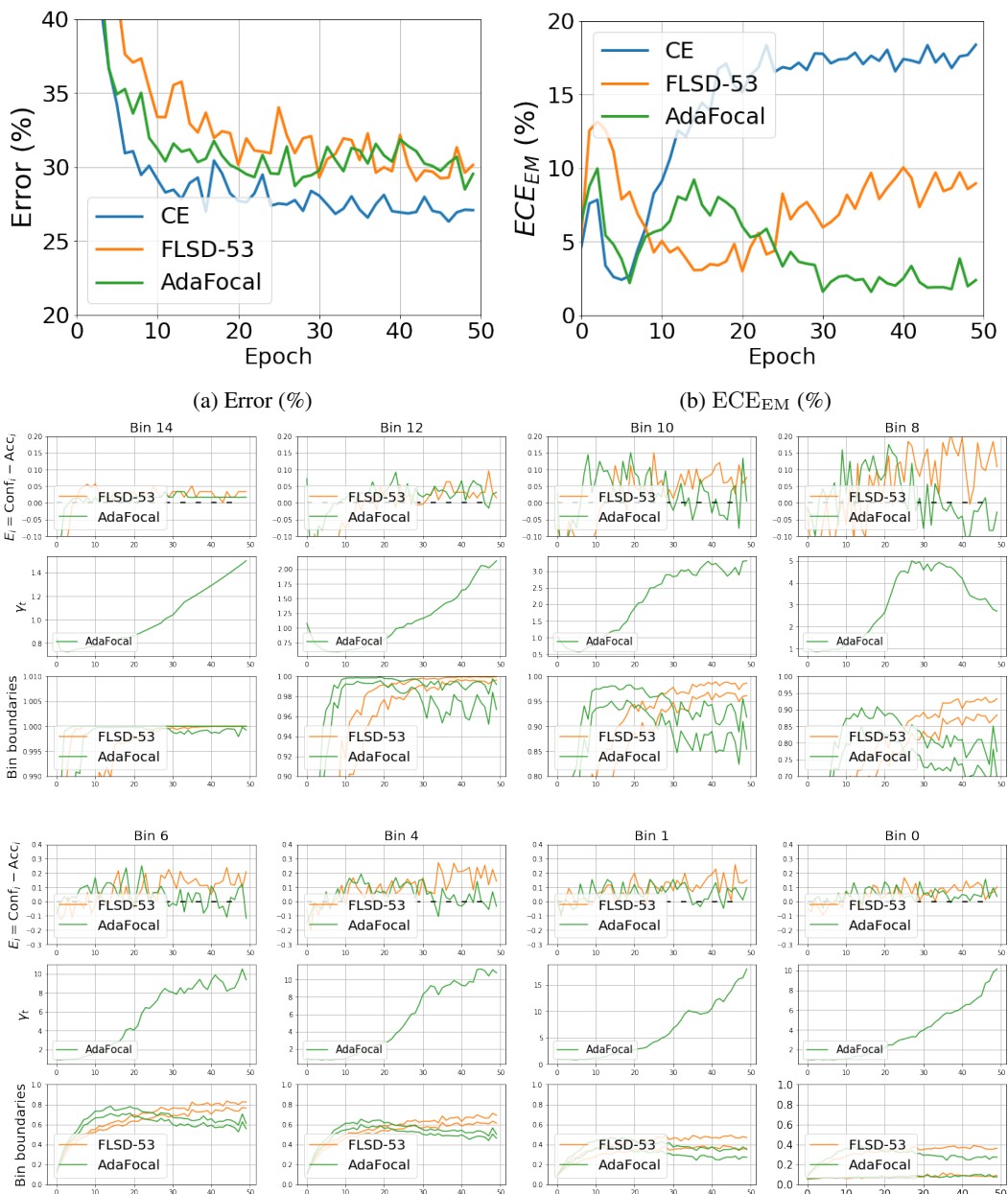

(a) Error (%)

(b) ECE$_{EM}$ (%)

(c) Dynamics of $\gamma$ and calibration behaviour in different bins. Each bin has three subplots: **top**: $E_{val,i} = C_{val,i} - A_{val,i}$, **middle**: evolution of $\gamma_t$, and **bottom**: bin boundaries. Black dashed line in top plot represent zero calibration error.

Figure 54: CNN trained on 20 Newsgroups with cross entropy (CE), FLSD-53, and AdaFocal.

## O.15   20 Newsgroups, BERT

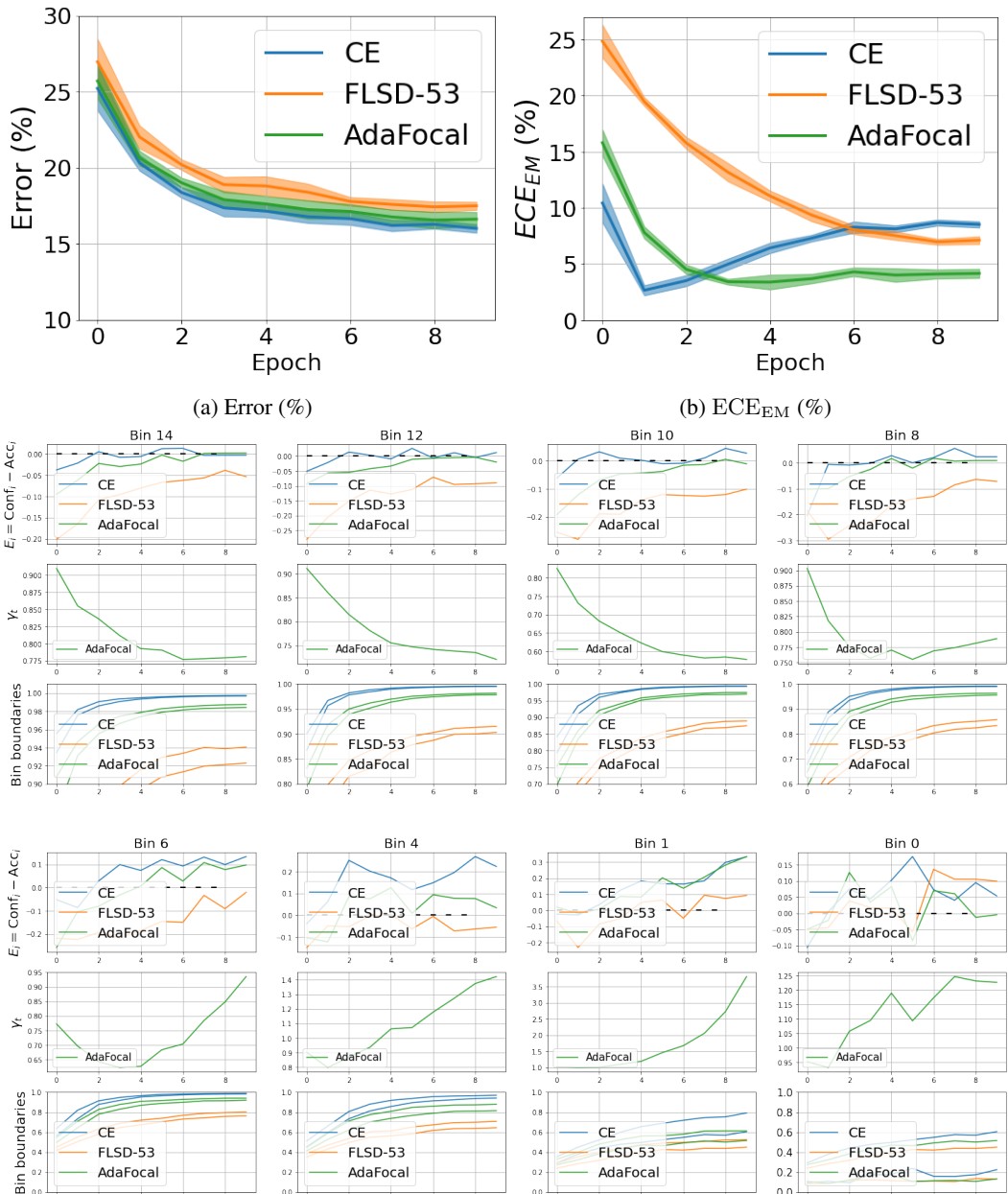

(a) Error (%)

(b) ECE$_{\mathrm{EM}}$ (%)

(c) Dynamics of $\gamma$ and calibration behaviour in different bins. Each bin has three subplots: **top**: $E_{val,i} = C_{val,i} - A_{val,i}$, **middle**: evolution of $\gamma_t$, and **bottom**: bin boundaries. Black dashed line in top plot represent zero calibration error.

Figure 55: Pre-trained BERT fine-tuned on 20 Newsgroups with cross entropy (CE), FLSD-53, and AdaFocal.

# P Comparison of CalFocal Loss Case 1 (Eq. 2) and Case 2 (Eq. 3)

For Fig. 56 below, please note the following legend:

- CE = Cross Entropy.
- $\mathcal{L}_{Exp,\lambda}$ = CalFocal loss case 1 (Eq. 2 in the main paper) which assigns $\gamma$s to each training sample.
- $\lambda$ = CalFocal loss case 2 (Eq. 3 in the main paper) which assigns a common $\gamma_b$ to all training samples that fall in validation-bin $b$.

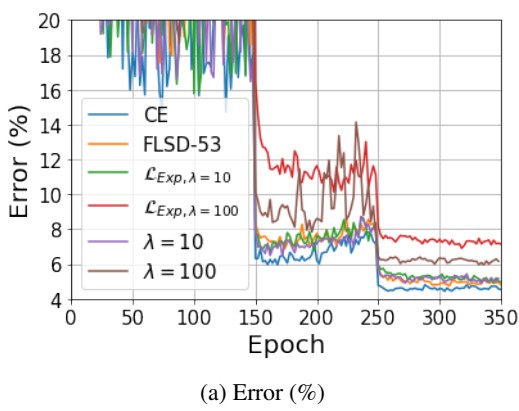

(a) Error (%)

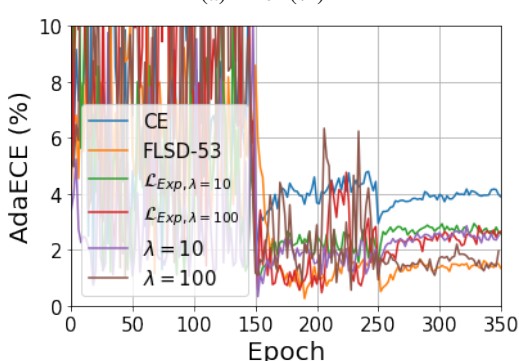

(b) $\text{ECE}_{EM}$ (%) (also called AdaECE in the literature [19]).

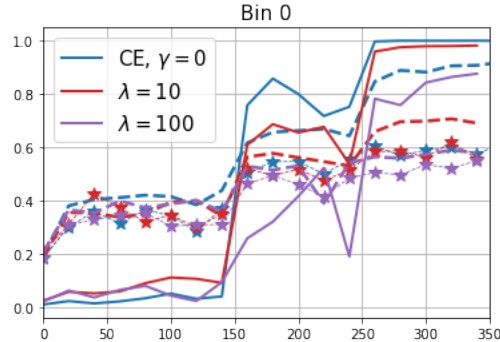

(c) For CalFocal loss case 2 (Eq. 3) marked in the legend by "$\lambda =$", the figure compares $C_{train}$ (solid line), $C_{val}$ (dashed line) and $A_{val}$ (starred lines) in validation bin-0 to show that when CalFocal brings $C_{train}$ closer to $A_{val}$, $C_{val}$ also approaches $A_{val}$.

Figure 56: ResNet-50 trained on CIFAR-10 using (1) cross entropy (CE), (2) FLSD-53 (3) CalFocal Case 1 loss function in Eq. 2 denoted by $\mathcal{L}_{Exp,\lambda}$, and (4) CalFocal Case 2 loss function in Eq. 3 denoted by "$\lambda =$".