# OpenReview forum: "AdaFocal: Calibration-aware Adaptive Focal Loss"
_NeurIPS.cc/2022/Conference — NeurIPS 2022 Accept_

### Official Review · Reviewer_2r5v · 2022-07-10

**Rating:** 6
**Confidence:** 4
**Soundness:** 3 good
**Presentation:** 4 excellent
**Contribution:** 3 good

**Summary:**

This paper proposes AdaFocal, a scheme to dynamically pick the $\gamma$ weight of the focal loss for the different groups of samples when training so that the expected calibration error is minimized w.r.t. validation set, during training a classifier. The weight $\gamma$ were adjusted so that the focal loss can suggest a classifier to be overconfident or underconfident in a certain region to calibrate the confidence to be closed to the mean accuracy. Experiments showed that AdaFocal outperformed many existing methods for calibrating the classifier, including the recent focal loss-based calibration method (FLSD-53) by Mukhoti et al.

**Questions:**

1. It is unclear to me what the weakness of AdaFocal is. Is it possible that training with AdaFocal may require a longer training time than using CE, etc.?
2. There is also a recent theoretical work [A], where they conducted a rigorous theoretical analysis of the behavior of $\gamma$ of focal loss and its relationship to calibration, which suggested that focal loss is not estimating class posterior probability. Also, it has been shown in their work that in a non-deep setting (Table 2 of [A] in the appendix: linear model or simple multilayered perception), using focal loss is worse than vanilla cross-entropy. I was wondering how is the performance of AdaFocal in such a simple setting.
3. SVHN is also a dataset that standard CE seems to be better than Focal loss in terms of calibration even in the deep learning setting (according to [A]). In AdaFocal paper and Mukhoti et al. papers, SVHN is not used. In AdaFocal paper, it is used in OOD detection but the ECE score is not presented in the main body of the paper. I was wondering if AdaFocal can mitigate this drawback of the focal loss (I feel AdaFocal may work well in SVHN but Mukhoti's method might not), because this loss use validation set to enforce the confidence is always calibrated w.r.t. calibration error.
4. Regarding the weakness of AdaFocal from the theoretical perspective highlighted in the "Weakness" section, have the authors considered the theoretical perspective of AdaFocal, perhaps something similar to the analysis in [Mukhoti et al.] or [A]?
5. Did the authors apply early stopping to the learner, since each method may have a different time to achieve best performance? For example, in Tiny Imagenet according to Figure 4, CE achieves the best performance around epoch 40 (where other methods also performed pretty well there). On the other hand, CE performance degraded as the epoch increases and temperature scaling may not be able to a reliable confidence score.

Minor
1: Table 8 in Appendix is an empty table. Please remove it or the authors may have forgotten to put the table information.


[A] Charoenphakdee, N., Vongkulbhisal, J., Chairatanakul, N., & Sugiyama, M. (2021). On focal loss for class-posterior probability estimation: A theoretical perspective. In Proceedings of the IEEE/CVF Conference on Computer Vision and Pattern Recognition (pp. 5202-5211).

**Limitations:**

The authors adequately addressed the limitations and potential negative societal impact of their work

**Strengths And Weaknesses:**

Significant and originality
- To my knowledge, the proposed method is novel, and its performance of the proposed method is impressive.

Strengths
- Proposed method is relatively simple to implement although there are several hyperparameters. It seems the standard choice of hyperparameters is applicable to many datasets at the same time (at least for image datasets in the experiments).
- Experiments on OOD detection have also been conducted, where AdaFocal is superior to other methods.

Weaknesses
- No theoretical aspect of the loss is discussed. It is uncleared what is happening to the minimizer of this loss function. In my understanding, the method is quite heuristic, the choice of weighting function in Algorithm is a little bit arbitrary (using exp(\lambda). Nevertheless, I think the intuition of the proposed method makes a lot of sense and quite interesting to try.
- AdaFocal introduces several hyperparameters to the model training process, e.g., number of bins to use, max/min of $\gamma$
- AdaFocal also introduces more training time since it has to evaluate the validation performance on each bin in every epoch (maybe not a big drawback in normal case)
- The accuracy of using AdaFocal is not guaranteed to be the same compared with standard training with CE + postprocessing methods like temperature scaling. Although it is unclear when AdaFocal can have better performance than CE (and vice versa: see CIFAR-10 ACC and CIFAR-100 Acc). For anyone who simply wants to use the same loss function (because the model has been using CE for a long time and many promising hyperparameters for CE may not be the one for AdaFocal) , using AdaFocal may not be the way to go.
- (minor) Reliability diagram result is not provided. I think looking at the reliability diagram can also be informative (at least in the appendix for future reference would be nice).

Clarity:
- The paper is well-written overall and not difficult to follow. However, I found that figure is a bit difficult to read and takes some time to understand. But the information provided in the paper is enough to understand the figure.

Overall:
I think the paper has sufficient evidence to suggest that using different $\gamma$ for different bins can be effective when using the Focal-loss style technique for improving the calibration of the classifier. The idea of using an over-confident version of Focal loss for calibration problems is not that straightforward in my opinion, and could potentially be used for a different problem in the future. Thus, I feel the paper has merits that outweigh its drawbacks, which are a lack of theoretical evidence and the increasing number of hyperparameters for training a model.

---

> ### Author Response · Authors · 2022-08-02
> **Author response to Reviewer 2r5v [1/2]**
>
> We thank the reviewer for taking the time to review our paper and provide helpful feedback and comments, especially for the reference to the paper that analyses focal loss and its relation to calibration theoretically. Please find our response to the comments and questions asked by the reviewer below.
>
> ### Reviewer comment 1:
> > - AdaFocal also introduces more training time since it has to evaluate the validation performance on each bin in every epoch (maybe not a big drawback in normal case)
> > - It is unclear to me what the weakness of AdaFocal is. Is it possible that training with AdaFocal may require a longer training time than using CE, etc.?
>
> **Author response:**
> To update $\gamma$ of AdaFocal, the extra operations that are required are
> 1. Forward pass on the validation set to compute the logits/softmaxes.
> 2. Compute bin statistics and update $\gamma$.
>
> In general, if we update $\gamma$ at the end of every epoch, then compared to the time it takes to train the model for one whole epoch, these two overheads are quite negligible. For example, for ResNet-50 trained on CIFAR-10 (train set contains $45000$ examples, val set contains $5000$ examples) using Nvidia Titan X Pascal GPU with 12GB memory,
> 1. Training for one epoch = $79,123$ ms = $79.1$ s
> 2. Forward pass on validation set = $2,886$ ms = $2.8$ s
> 3. Compute bin statistics and update  $\gamma$ = $8$ ms
>
> So if the standard training with cross entropy, without any involvement of a validation set, for 350 epochs, requires in total $79.1 \times \frac{350}{3600} = 7.7$ hours, then AdaFocal will add $2.808\times \frac{350}{60} = 16$ minutes on top of the entire training.
> Naturally, if we update $\gamma$ more often during the epoch then this overhead will increase and may become significant. However, for all our experiments we update $\gamma$ at the end of an epoch and that works quite well. Nonetheless, for a comparison of performance of AdaFocal when the update frequency of $\gamma$ is varied, please refer to Appendix J.
>
> ### Reviewer comment 2:
> > SVHN is also a dataset that standard CE seems to be better than Focal loss in terms of calibration even in the deep learning setting (according to [A]). I was wondering if AdaFocal can mitigate this drawback of the focal loss. (I feel AdaFocal may work well in SVHN but Mukhoti's method might not), because this loss use validation set to enforce the confidence is always calibrated w.r.t. calibration error.
>
> **Author response:**
> We have added the experiments on SVHN with ResNet-110 using cross entropy, Focal loss $\gamma=3$, FLSD-53, and AdaFocal to Appendix P.11. The reliability diagrams for these experiments are provided in Appendix M. We observe that:
> 1. Contrary to [A], Focal loss $\gamma=3$ and FLSD-53 perform better than CE on SVHN by a large margin.
> 2. AdaFocal, towards the end of the training, is able to match the performance of focal losses but not surpass them.
>
> From these experiments we see that  FLSD-53 itself is quite good for SVHN ResNet-110. It can very well be the case that for a particular dataset-model pair FLSD-53’s $\gamma$ lead to the best calibration, however knowing that beforehand is impossible without an extensive hyperparameter search. Using FLSD-53 for every dataset-model pair is also not recommended (FLSD-53 is highly miscalibrated on ImageNet ResNet-50). AdaFocal, on the other hand, promises to find the $\gamma$ values that result in either the best or very close to the best calibration (for example in this case).
>
>
> ### Reviewer comment 3:
> > Did the authors apply early stopping to the learner, since each method may have a different time to achieve best performance? For example, in Tiny Imagenet according to Figure 4, CE achieves the best performance around epoch 40 (where other methods also performed pretty well there). On the other hand, CE performance degraded as the epoch increases and temperature scaling may not be able to a reliable confidence score.
>
> **Author response:**
> For all experiments, except Tiny-ImageNet, we select the model at the end of the training mainly to be consistent with Mukhoti et al. (the work that we are trying to improve upon). Further, as confirmed by the authors of Mukhoti et al., they also use the model at the end of the training to report all their results. Therefore, for the following datasets, the error and ECE are reported for the model at the end of the training at
> 1. CIFAR-10: 350 epochs
> 2. CIFAR-100: 350 epochs
> 3. ImageNet: 90 epochs
> 4. NewsGroups: 50 epochs
>
> However, as rightly pointed out by the reviewer, for Tiny-ImageNet the model at the end is not the best. Therefore, in our paper (originally submitted) we have rightly reported the model that has the lowest error on the validation set. We do the same for the recently added BERT model trained on 20Newsgroup as this is a newly added experiment during rebuttal and was not included in Mukhoti et al. for comparison. This above information was already included in Appendix E.3

---

> ### Author Response · Authors · 2022-08-02
> **Author response to Reviewer 2r5v [2/2]**
>
> ### Reviewer comment 4:
> > AdaFocal introduces several hyperparameters to the model training process, e.g., number of bins to use, max/min of  gamma
>
> **Author response:**
> The hyperparameters introduced to AdaFocal are: $\lambda$, $\gamma_{min/max}$, $S_{th}$ but these hyperparameters do not require extensive hyperparameter search and are relatively easy to select.
> Based on the experiments in the paper
> 1. $\lambda$ is redundant as for all our experiments $\lambda=1$ worked very well. We could have easily omitted \lambda from AdaFocal’s Eq. 4 but we kept it as it was used in Eq. 3 of the toy loss function CalFocal. Further for an untested dataset-model pair that we have not explored, if after running AdaFocal with default settings one sees that increasing/decreasing the rate of change of$ \gamma$ might be helpful then one can use $\lambda$ to do so. However, in our experiments we didn’t need to do so.
> 2. $\gamma_{max}$ is introduced to stop $\gamma$ from exploding and keep the training stable. This can be drawn parallels with the common practice of gradient clipping in machine learning. Further, note that $\gamma_{max}$ do not require any special fine-tuning. Its sole purpose is to stop γ from exploding and any reasonable value around 15, 20, 25 should work well in practice. For all our experiments, we use $\gamma_{max} = 20$.
> 3. Similarly for $\gamma_{min}$, it was very easy to choose $\gamma_{min} = −2$ as our threshold point as for ImageNet we observed that values beyond $−2$ led to unstable training for ResNet-50. In the rest of the experiments the gamma values were always $ > -2$ and never faced this issue. For a new untested dataset-model pair, if $\gamma_{min} = −2$ does not work then also it should be fairly easy to select by looking at the $\gamma$ values in “evolution/dynamics of $\gamma$” plots (similar to Fig. 5 in the main paper) at the time step at which training becomes unstable and use the $\gamma$ values at that point as the new threshold.
> 4. $S_{th}$ also does not require an extensive hyperparameter search. It can be easily selected by observing how the $\gamma$ evolves for an initial test run of AdaFocal that does not switch to inverse focal loss. For example for Imagenet, if the bins are under-confident from the start then AdaFocal is going to push $\gamma$ (starting at $\gamma=1$) towards 0 and < 0. In this case it makes sense to have a higher $S_{th}$ as we know that using $\gamma < 0$ is beneficial for this dataset-model pair so why not switch to it early. This is the logic that we used to determine $S_{th}=0.2$ for Imagenet and it took us only one extra trial run. And to support our point about switching early we have shown the results with $S_{th}=0.5$ in our paper as well.
>
> Therefore, it is much more easy and beneficial to use AdaFocal with these hyperparameters than using focal loss and run a huge search for $\gamma$ for every bin (and for each time step).
>
> ### Reviewer comment 5:
> > Reliability diagram result is not provided. I think looking at the reliability diagram can also be informative (at least in the appendix for future reference would be nice).
>
> **Author response:**
> We have added reliability diagrams to Appendix M for all dataset-model pairs.
>
> ### Reviewer comment 6:
> > Regarding the weakness of AdaFocal from the theoretical perspective highlighted in the "Weakness" section, have the authors considered the theoretical perspective of AdaFocal, perhaps something similar to the analysis in [Mukhoti et al.] or [A]?
>
> **Author response:**
> We thank the reviewer for pointing us to [A] as we were not aware of this work before but is really interesting and of prime importance to us. During the rebuttal period, however, we were not able to fully analyze AdaFocal from all the theoretical perspectives of focal loss discussed in [A], therefore, as part of our future work we certainly plan to build on these theoretical results to provide a strong support and explanation for AdaFocal.
>
> Nonetheless, from a preliminary reading of [A], we think that if focal loss with $\gamma>0$ suffers from $\eta$UC (under-confidence) and $\eta$OC (over-confidence), then logically AdaFocal should suffer from these as well because AdaFocal still uses $\gamma>0$, with the difference that different training samples experience different $\gamma$s that further change over the training epochs. However, we will need further investigation with respect to how this adaptive change in $\gamma>0$ behave with respect to $\eta$UC and $\eta$OC and provide the improvement in calibration that we see in our paper. This is definitely part of our future work as we were looking for some theoretical foundations to analyse the complicated behaviour of AdaFocal brought upon by the dynamic change of $\gamma$ at every step. We again thank the reviewer for pointing us to the paper.

---

> > ### Comment · Reviewer_2r5v · 2022-08-07
> > **Thank you for your detailed reply!**
> >
> > I appreciate the authors' rebuttal reply to clarify my questions and concerns, providing feedback on experiments using SVHN and some future plans for theoretical analysis of AdaFocal, adding reliability diagrams, etc.  I am quite convinced that AdaFocal can improve upon FLSD-53, which is the work that this paper is built based on. I increased the soundness score from 2 to 3 and I keep my score to be on the positive side (6: Weak Accept).

---

### Official Review · Reviewer_J28C · 2022-07-10

**Rating:** 6
**Confidence:** 4
**Soundness:** 3 good
**Presentation:** 3 good
**Contribution:** 3 good

**Summary:**

The authors present an adaptation of Focal Loss that adapts the focal hyperparameter over the course of training to the data. This produces on average better calibrated models than traditional cross entropy or focal loss, especially when combined with post-hoc recalibration.



**Questions:**

- l. 76 what is $\mathbf{H}$ again?
- Can you reason why for ImageNet Post Temperature Scaling CE produces the best calibrated model?

**Strengths And Weaknesses:**

**Strengths**

- Incidentally, it also removes the focal hyperparameter from consideration.
- The authors introduce problem and related concepts well.
- The results section is comprehensive within their chosen focus and concise.
- The work overall is novel and relevant as model calibration is important for reliability.

**Weaknesses**

- The authors mostly focus on image classification. Some of their observations may not hold for other scenarios. e.g. their hyperparameter $\lambda$ might become relevant.
- Section 3 and following would benefit from some editing.
- There are a few naming inconsistencies e.g. calibration error and calibration gap. Consistency here makes it easier to read.
- The explanation of how the validation binning is applied to the training samples could be expanded.
- There are some typos (FLDS, parametera) and I think the qualities you describe as "innate" are more inherent.

---

> ### Author Response · Authors · 2022-08-02
> **Author response to Reviewer J28C**
>
> We thank the reviewer for taking the time to review our paper and provide helpful feedback and comments. We will use them to further solidify the message of the paper. Please find the changes that we have made to the paper during the rebuttal period in the comments to all reviewers. Our response to the comments and questions asked by the reviewer is given below.
>
> ## Reviewer comment 1:
> > l. 76 what is  again?
>
> **Author response:**
> In line 76 of the main text, $\mathbb{H}(\hat{\mathbf{p}})$ is the entropy of the prediction distribution (softmax) $\hat{\mathbf{p}}$. We will make it explicit in the main text.
>
> ## Reviewer comment 2:
> > Can you reason why for ImageNet Post Temperature Scaling CE produces the best calibrated model?
>
> **Author response:**
>
> | Method| No post-hoc| TS | ETS | Spline |
> | ----------- | ----------- | ----------- | ----------- | ----------- |
> | CE |  2.93 | 1.50 | 0.90 | 0.82 |
> | FLSD-53|  16.77 | 2.62 | 2.13 | 0.87 |
> | AdaFocal| 1.87 | 1.87 | 1.13 | **0.66** |
>
> Here AdFocal+Spline gives the best result. Similarly, we find the best performing combination for other experiments as
> - 20Newsgroup, CNN: 1.12 AdFocal+Spline
> - CIFAR-10, ResNet-50: 0.44 AdFocal+TS
> - CIFAR-10, ResNet-110: 0.57 AdFocal+ETS
> - CIFAR-10, Wide-ResNet: 0.44 AdFocal+TS
> - CIFAR-10, DenseNet-121: 0.53 AdFocal+Spline
> - CIFAR-100, ResNet-50: 1.01 AdFocal+Spline
> - CIFAR-100, ResNet-110: 1.24 AdFocal+ETS
> - CIFAR-100, Wide-ResNet: 1.08 FLSD-53+Spline
> - CIFAR-100, DenseNet-121: 1.03 FLSD-53+Spline
> - Tiny-ImageNet, ResNet-50: 1.23 AdFocal+ETS
>
> From these, it is not clear which post hoc calibration is the best choice to be combined with the “calibration during training” method (let’s call it pre-calibration). However, we observe that, in 9 out of 11 cases, it is the model that has been pre-calibrated by AdaFocal that gives the best result when combined with one of the post-calibration techniques. For an unknown dataset-model pair, the selection of the best pre-calibration+post-calibration methods might require more investigation (and a different study in itself) but based on the evidence in our paper we find that if we start with a better pre-calibrated model we get a better post-calibrated model as well, and AdaFocal is much better in generating the pre-calibrated models by a large margin.
>
> ## Reviewer comment 3:
> > - Section 3 and following would benefit from some editing.
> > - There are a few naming inconsistencies e.g. calibration error and calibration gap. Consistency here makes it easier to read.
> > - The explanation of how the validation binning is applied to the training samples could be expanded.
> > - There are some typos (FLDS, parameters) and I think the qualities you describe as "innate" are more inherent.
>
> **Author response:**
> Thank you for pointing these out, we will fix these issues in the paper.

---

> > ### Comment · Reviewer_J28C · 2022-08-08
> > **Thanks for the clarifications**
> >
> > Dear authors,
> >
> > thanks you for clarifying (most) of the raised points. I think it would be meaningful to clearly that the choice for TS/Splits/ETS boils down to an additional hyperparameter that needs to be found during a hyperparameter sweep.

---

> > > ### Author Response · Authors · 2022-08-08
> > > **Thank you for the suggestion.**
> > >
> > > We will add to our paper the above discussion about the choice of "during training calibration" + post-hoc calibration techniques to make it clear that the topic needs more investigation, which, however, does not affect the main contribution of the paper i.e. to produce very well calibrated models during training.
> > >
> > > Further, do let us know if we have missed out on any particular point that you wanted clarification on (as in your comment you have mentioned "most").
> > > 1. Regarding editing of section 3 and more explanation about the binning of training samples using validation bins, we will add more details to Appendix B and C describing the process of binning before diving into the correspondence of C_train and C_val.
> > > 2. We will fix the typos and replace "innately" with "inherently".

---

### Official Review · Reviewer_9C4n · 2022-07-11

**Rating:** 5
**Confidence:** 4
**Soundness:** 3 good
**Presentation:** 4 excellent
**Contribution:** 3 good

**Summary:**

This paper proposes a focal-loss-based method called AdaFocal for uncertainty calibration problem. When training models using AdaFocal, the hyper-parameter $\gamma$ used in focal loss can be dynamically varied during epochs and different samples.

The experimental results show the effectiveness of AdaFocal for calibration in both the cases of with and without post-hoc calibration. The experiments also shown AdaFocal can easily align mean confidence with accuracy of each bin in validation set.


**Questions:**

See weaknesses.

**Limitations:**

The authors have addressed the limitations.

**Strengths And Weaknesses:**

Strengths:

1. The proposed AdaFocal method exploits an adaptive strategy to determine the important hyperparameter $\gamma$ used in focal loss. AdaFocal can be considered as a generalized version of naive Focal loss and Focal-53.

2. The illustrative experiments before introducing the proposed method give clear motivation.

3. The datasets used in experiment part are from different domains, while have different difficulty degree. The results show the effectiveness of the proposed method.

Weaknesses:

1. The proposed method is used to adaptively determine the hyperparameter $\gamma$ used in Focal loss. However, other hyperparameters are induced in the proposed method like $\lambda$ and $S_{th}$. Would these hyperparameters significantly impact influence the final performance? Given this more complicated method, one naturally expects that the choosing of new hyperparameters like $\lambda$ and $S_{th}$ is much easier than that of $\gamma$.

2. Figure 2 shows the good correspondence between the average confidence of training set and validation set. In this case, the training samples are grouped using **validation-bin boundaries**. I think a fairer setting is that the training samples and validation samples are grouped independently into their respective training-bins and validation-bins. But in this case (as is shown in Figure 9) the correspondence between them is not good.

---

> ### Author Response · Authors · 2022-08-02
> **Author response to Reviewer 9C4n [1/2]**
>
> We thank the reviewer for taking the time to review our paper and provide helpful feedback and comments. We will use them to further solidify the message of the paper. Please find the changes that we have made to the paper during the rebuttal period in the comments to all reviewers. Our response to the comments and questions asked by the reviewer is given below.
>
> ## Reviewer comment 1:
> > The proposed method is used to adaptively determine the hyperparameter  used in Focal loss. However, other hyperparameters are induced in the proposed method like  and . Would these hyperparameters significantly impact influence the final performance? Given this more complicated method, one naturally expects that the choosing of new hyperparameters like  and  is much easier than that of .
>
> **Author response:**
> The hyperparameters introduced to AdaFocal are: $\lambda$, $\gamma_{min/max}$, $S_{th}$ but these hyperparameters do not require extensive hyperparameter search and are relatively easy to select.
> Based on the experiments in the paper
> 1. $\lambda$ is redundant as for all our experiments $\lambda=1$ worked very well. We could have easily omitted \lambda from AdaFocal’s Eq. 4 but we kept it as it was used in Eq. 3 of the toy loss function CalFocal. Further for an untested dataset-model pair that we have not explored, if after running AdaFocal with default settings one sees that increasing/decreasing the rate of change of$ \gamma$ might be helpful then one can use $\lambda$ to do so. However, in our experiments we didn’t need to do so.
> 2. $\gamma_{max}$ is introduced to stop $\gamma$ from exploding and keep the training stable. This can be drawn parallels with the common practice of gradient clipping in machine learning. Further, note that $\gamma_{max}$ do not require any special fine-tuning. Its sole purpose is to stop γ from exploding and any reasonable value around 15, 20, 25 should work well in practice. For all our experiments, we use $\gamma_{max} = 20$.
> 3. Similarly for $\gamma_{min}$, it was very easy to choose $\gamma_{min} = −2$ as our threshold point as for ImageNet we observed that values beyond $−2$ led to unstable training for ResNet-50. In the rest of the experiments the gamma values were always $ > -2$ and never faced this issue. For a new untested dataset-model pair, if $\gamma_{min} = −2$ does not work then also it should be fairly easy to select by looking at the $\gamma$ values in “evolution/dynamics of $\gamma$” plots (similar to Fig. 5 in the main paper) at the time step at which training becomes unstable and use the $\gamma$ values at that point as the new threshold.
> 4. $S_{th}$ also does not require an extensive hyperparameter search. It can be easily selected by observing how the $\gamma$ evolves for an initial test run of AdaFocal that does not switch to inverse focal loss. For example for Imagenet, if the bins are under-confident from the start then AdaFocal is going to push $\gamma$ (starting at $\gamma=1$) towards 0 and < 0. In this case it makes sense to have a higher $S_{th}$ as we know that using $\gamma < 0$ is beneficial for this dataset-model pair so why not switch to it early. This is the logic that we used to determine $S_{th}=0.2$ for Imagenet and it took us only one extra trial run. And to support our point about switching early we have shown the results with $S_{th}=0.5$ in our paper as well.
>
> Therefore, it is much more easy and beneficial to use AdaFocal with these hyperparameters than using focal loss and run a huge search for $\gamma$ for every bin (and for each time step). We will add this discussion to the Appendix of the paper.

---

> > ### Author Response · Authors · 2022-08-02
> > **Author response to Reviewer 9C4n [2/2]**
> >
> > ## Reviewer comment 2:
> > > Figure 2 shows the good correspondence between the average confidence of training set and validation set. In this case, the training samples are grouped using validation-bin boundaries. I think a fairer setting is that the training samples and validation samples are grouped independently into their respective training-bins and validation-bins. But in this case (as is shown in Figure 9) the correspondence between them is not good.
> >
> > **Author response:**
> > We would like to clarify that by correspondence we do not mean $C_{train}$ is equal to $C_{val}$, we simply mean that there is a good correspondence between the “change in $C_{train}$” and the “change in $C_{val}$” i.e. if we increase/decrease $C_{train}$, we do see an increase/decrease in $C_{val}$ as well. This is true for the case of independent binning as well as we see in Fig. 9 where as \gamma is increased from 0 (CE) to 3 to 5, we see that $C_{train}$ decreases and similarly the $C_{val}$ in that bin decreases as well.
> >
> > For AdaFocal to work, it does not matter how close $C_{train}$ is to $C_{val}$. All that matters is a way to control $C_{val}$ using $C_{train}$ so that we can push $C_{val}$ closer to $A_{val}$ (to ultimately reduce calibration error = $C_{val}$ - $A_{val}$). In that regard, for AdaFocal, the correspondence does not even have to be very strong i.e. it does not matter if an increase/decrease by $x$ amount does not see the same $x$ amount of change in $C_{val}$. AdaFocal’s aim is to step by step push $C_{train}$ in a direction which will push $C_{val}$ towards $A_{val}$. Note that $C_{train}$ does not need to be close to $A_{val}$ which gives the freedom to keep pushing $C_{train}$ further until $C_{val}$ is close to $A_{val}$. For e.g. let’s assume $C_{val} - A_{val} = -2x$, then by pushing $C_{train}$ by $2x$ amount upwards let’s say we only see $C_{val}$ change by $1.5x$ i.e. now $C_{val} - A_{val} = -0.5x$, in that case we push $C_{train}$ further to see if $C_{val} - A_{val} = 0$. In case of, overshoot $C_{val} - A_{val} > 0$, $C_{train}$ is brought down to bring $C_{val}$ lower as well.
> >
> > This is the working principle of AdaFocal which neither requires $C_{train}$ to be equal to $C_{val}$ nor does it require $C_{train}$ and $C_{val}$ to have a really strong correspondence. A loose or good enough correspondence is sufficient for AdaFocal to function and produce well calibrated models. We will highlight this point more clearly in the main text of the paper

---

### Official Review · Reviewer_gWGg · 2022-07-11

**Rating:** 4
**Confidence:** 3
**Soundness:** 3 good
**Presentation:** 3 good
**Contribution:** 2 fair

**Summary:**

The paper proposes AdaFocal, an adaptive focal loss that modifies the focal loss hyperparameter $\gamma$ for different sample bins based on historic values of $\gamma$ and the model's per-bin calibration properties. The authors evaluate AdaFocal on several image classification benchmarks (CIFAR-10, CIFAR-100, TinyImageNet, ImageNet) and one text classification dataset (20 Newsgroup).

**Implementation details**
The $\gamma$ update rule adjusts $\gamma_{t+1}$ based on the value of $\gamma_{t}$ and the difference between the model's accuracy and confidence in a particular validation bin. Training examples are assigned to particular validation bin based on validation bin boundaries that are re-computed once per epoch. The adjustment for $\gamma_{t+1}$ is then computed based on the validation bin miscalibration that the training example is assigned to.

**Questions:**

- The validation bin boundaries are only updated once per training epoch. What happens if they are updated more frequently (i.e. every minibatch or at least a few times per training epoch)?


**Limitations:**

No, the authors should address limitations and potential negative social impact of their work.

**Strengths And Weaknesses:**

Strengths:
- The update rule for gamma and the algorithm for AdaFocal is fairly straightforward. The overall simplicity will ease implementation and increase adoption of the method.
- The authors give strong empirical evidence that the confidence of the neural network on the training data roughly matches the confidence of the validation data for the image classification datasets, which helps motivate the AdaFocal method.
- The empirical results for AdaFocal are strong across all experimental setups for pre-temperature scaling and strong for CIFAR-10/CIFAR-100 for post-temperature scaling.


Weaknesses:
- The method still involves some heuristics which may not generalize to other datasets/tasks and the paper uses empirical evidence collected only on a limited selection of datasets/tasks to justify the method.
  - For example, one such heuristic is thresholding $\gamma_{t}$ based on a pre-set hyperparameter $S_{th}$ to switch between the inverse-focal and focal loss. Another heuristic is the number of bins. Both of these hyperparameters may vary in optimal value based on the dataset, model, and task.
- Only one model architecture is tested for TinyImageNet, ImageNet, and Newsgroup 20.
- Post temperature scaling, other methods outperform AdaFocal on both ImageNet and Newsgroup 20 tasks.
- Error bars should be included in Figure 4 if the plots are showing an average over 5 runs.
- The plots comparing confidence of training and validation samples should be included for the Newsgroup 20 classification task.
- Missing baseline comparison to NeurIPS '21 calibration-during-training method "Soft Calibration Objectives for Neural Networks" (https://arxiv.org/abs/2108.00106)

My main reason for leaning towards a borderline reject rather than accept is that the method is justified purely based on empirical evaluation but the empirical evaluation has several weaknesses (mentioned above).

---

> ### Author Response · Authors · 2022-08-02
> **Author response to Reviewer gWGg [1/2]**
>
> We thank the reviewer for taking the time to review our paper and provide helpful feedback and comments. We will use them to further solidify the message of the paper. Please find the changes that we have made to the paper during the rebuttal period in the comments to all reviewers. Our response to the comments and questions asked by the reviewer is given below..
>
> ### Reviewer comment 1:
> > The validation bin boundaries are only updated once per training epoch. What happens if they are updated more frequently (i.e. every minibatch or at least a few times per training epoch)?
>
> **Author response:**
> In order to check the behaviour of AdaFocal with more frequent $\gamma$ updates, we ran experiments on 20newsgroup with BERT and CNN (as they are faster to train in this rebuttal period) and have presented the results in Appendix J. From these experiments, we observe that
> 1. Updating $\gamma$ more often leads to better tracking of the calibration behaviour of the validation set at all time steps and therefore overall better ECE performance of the model.
> 2. However, this comes at the cost of increased training time.
>
> A detailed discussion is presented in Appendix J of the revised paper.
>
> ### Reviewer comment 2:
> > Only one model architecture is tested for TinyImageNet, ImageNet, and Newsgroup 20.
>
> **Author response:**
> For Tiny-Imagenet we have added additional experiments for ResNet110 in Table 1 and 2 of the main paper. Similarly for 20Newsgroup, we have added results with the BERT model as well. Both the experiments show AdaFocal as the best performing model. For Imagenet, we were not able to finish the experiments given the limited time and the resources that had to be shared with other experiments during the rebuttal. We will test other architectures as time permits to further confirm the performance of AdaFocal on Imagenet.
>
> ### Reviewer comment 3:
> > - Error bars should be included in Figure 4 if the plots are showing an average over 5 runs.
> > - The plots comparing confidence of training and validation samples should be included for the Newsgroup 20 classification task.
>
> **Author response:**
> We have updated Fig. 4 with error bars in the resubmitted paper and have included the plot for $C_{train}$ vs $C_{val}$ for 20Newsgroup as Fig 14 in Appendix C.6. We see a good correspondence between $C_{train}$ and $C_{val}$ in this case as well.
>
> ### Reviewer comment 4:
> > Post temperature scaling, other methods outperform AdaFocal on both ImageNet and Newsgroup 20 tasks.
>
> **Author response:**
> In the main paper we have presented results with only temperature scaling. For a full comparison with other post-calibration methods, we have also included Ensemble Temperature Scaling (ETS) and Spline fitting in Table 3 and 4 in Appendix F. Combining the results from Table1, 3 and 4:
>
> ImageNet:
> | Method| No post-hoc cal| TS | ETS | Spline |
> | ----------- | ----------- | -----| -----| -----|
> CE | 2.93 | 1.50 | 0.90 | 0.82 |
> FLSD-53 | 16.77 | 2.62 | 2.13 | 0.87 |
> AdaFocal | 1.87 | 1.87 | 1.13 |**0.66**|
>
> 20 newsgroup:
> | Method| No post-hoc cal| TS | ETS | Spline |
> | ----------- | ----------- | -----| -----| -----|
> CE | 18.57 | 4.08 |2.46 |1.97|
> FLSD-53| 8.86 | 2.13 | 2.50 | 1.38|
> AdaFocal | 2.62 | 2.46 | 2.29 | **1.12**|
>
> Further, the best combinations across different dataset-model pairs are:
> - ImageNet, ResNet-50: 0.66 AdFocal+Spline
> - 20Newsgroup, CNN: 1.12 AdFocal+Spline
> - CIFAR-10, ResNet-50: 0.44 AdFocal+TS
> - CIFAR-10, ResNet-110: 0.57 AdFocal+ETS
> - CIFAR-10, Wide-ResNet: 0.44 AdFocal+TS
> - CIFAR-10, DenseNet-121: 0.53 AdFocal+Spline
> - CIFAR-100, ResNet-50: 1.01 AdFocal+Spline
> - CIFAR-100, ResNet-110: 1.24 AdFocal+ETS
> - CIFAR-100, Wide-ResNet: 1.08 FLSD-53+Spline
> - CIFAR-100, DenseNet-121: 1.03 FLSD-53+Spline
> - Tiny-ImageNet, ResNet-50: 1.23 AdFocal+ETS
>
> We observe that, in 9 out of 11 cases, it is the model that has been pre-calibrated by AdaFocal that gives the best result when combined with one of the post-calibration. This is consistent with the main focus of the paper that pre-calibrated models from training (calibration during training method) are more advantageous than uncalibrated models. We do not advocate that AdaFocal+TS is the best choice but that AdaFocal+”some post hoc calibration” is better than other “calibration during training methods+post hoc calibration”. For an unknown dataset-model pair, the selection of the best pre-calibration+post-calibration methods might require more investigation (and a different study in itself) but based on the evidence in our paper we find that if we use a better pre-calibrated model we get a better post-calibrated model as well, and AdaFocal is the best technique in generating the best pre-calibrated model by a large margin.

---

> > ### Author Response · Authors · 2022-08-02
> > **Author response to Reviewer gWGg [2/2]**
> >
> > ### Reviewer comment 5:
> > > The method still involves some heuristics which may not generalise to other datasets/tasks and the paper uses empirical evidence collected only on a limited selection of datasets/tasks to justify the method. For example, one such heuristic is thresholding γt  based on a pre-set hyperparameter Sth to switch between the inverse-focal and focal loss. Another heuristic is the number of bins. Both of these hyperparameters may vary in optimal value based on the dataset, model, and task.
> >
> > **Author response:**
> > The hyperparameters introduced to AdaFocal are: $\lambda$, $\gamma_{min/max}$, $S_{th}$ but these hyperparameters do not require extensive hyperparameter search and are relatively easy to select.
> > Based on the experiments in the paper
> > 1. $\lambda$ is redundant as for all our experiments $\lambda=1$ worked very well. We could have easily omitted \lambda from AdaFocal’s Eq. 4 but we kept it as it was used in Eq. 3 of the toy loss function CalFocal. Further for an untested dataset-model pair that we have not explored, if after running AdaFocal with default settings one sees that increasing/decreasing the rate of change of$ \gamma$ might be helpful then one can use $\lambda$ to do so. However, in our experiments we didn’t need to do so.
> > 2. $\gamma_{max}$ is introduced to stop $\gamma$ from exploding and keep the training stable. This can be drawn parallels with the common practice of gradient clipping in machine learning. Further, note that $\gamma_{max}$ do not require any special fine-tuning. Its sole purpose is to stop γ from exploding and any reasonable value around 15, 20, 25 should work well in practice. For all our experiments, we use $\gamma_{max} = 20$.
> > 3. Similarly for $\gamma_{min}$, it was very easy to choose $\gamma_{min} = −2$ as our threshold point as for ImageNet we observed that values beyond $−2$ led to unstable training for ResNet-50. In the rest of the experiments the gamma values were always $ > -2$ and never faced this issue. For a new untested dataset-model pair, if $\gamma_{min} = −2$ does not work then also it should be fairly easy to select by looking at the $\gamma$ values in “evolution/dynamics of $\gamma$” plots (similar to Fig. 5 in the main paper) at the time step at which training becomes unstable and use the $\gamma$ values at that point as the new threshold.
> > 4. $S_{th}$ also does not require an extensive hyperparameter search. It can be easily selected by observing how the $\gamma$ evolves for an initial test run of AdaFocal that does not switch to inverse focal loss. For example for Imagenet, if the bins are under-confident from the start then AdaFocal is going to push $\gamma$ (starting at $\gamma=1$) towards 0 and < 0. In this case it makes sense to have a higher $S_{th}$ as we know that using $\gamma < 0$ is beneficial for this dataset-model pair so why not switch to it early. This is the logic that we used to determine $S_{th}=0.2$ for Imagenet and it took us only one extra trial run. And to support our point about switching early we have shown the results with $S_{th}=0.5$ in our paper as well.
> >
> > Therefore, it is much more easy and beneficial to use AdaFocal with these hyperparameters than using focal loss and run a huge search for $\gamma$ for every bin (and for each time step).
> >
> > ### Reviewer comment 6:
> > > Missing baseline comparison to NeurIPS '21 calibration-during-training method "Soft Calibration Objectives for Neural Networks" (https://arxiv.org/abs/2108.00106)
> >
> > **Author response:**
> > We thank the reviewer for pointing to the paper as we had missed it in our literature survey. However, at the moment we are not able to present comparisons to this work (S-AVUC loss function) due to the following issues:
> > 1. The method requires massive hyperparameter search over 4 parameters: threshold parameter $\kappa$, temperature parameter $T$, weight of the secondary loss parameter $\beta$, and weight of the L2-normalisation parameter $\lambda$.
> > 2. Further, the authors have also not shared the exact value of the hyperparameters that worked best for them.
> >
> > Given the time and resources it takes to conduct each experiment, such a hyperparameter search was not possible during the rebuttal period.

---

> > > ### Comment · Reviewer_gWGg · 2022-08-08
> > > **Response**
> > >
> > > I've read the author response and the other reviews. I still have some concerns regarding the empirical evaluation. There is still only one model being tested on ImageNet, though I do appreciate the addition of the BERT model for Newsgroup 20 and the additional ResNet for TinyImageNet. I am also still concerned about the overall reliance on heuristics, lack of theoretical motivation, and introduction of new hyperparameters.
> > >
> > > I increased the "soundness" score from 2 to 3 in light of the additional experiments.

---

### Author Response · Authors · 2022-08-02
**Additions to the main paper and appendix during the rebuttal phase.**

To all the reviewers,

We have made the following changes to the paper during the rebuttal phase:

**Main paper:**
1. Results for Tiny-ImageNet, ResNet-110 and 20newsgroup, BERT are added to Table 1 and 2. We find that AdaFoal performs the best in these cases as well.
2. Fig. 4 is updated with error bars (mean and standard deviation).

**Appendix:**
1. Experiments regarding “frequency of $\gamma$-update” are added to Appendix J.
2. Plots for $C_{train}$ vs $C_{val}$ for 20Newsgroup are added to Fig 14 in Appendix C.6.
3. Reliability diagrams for all dataset-model pairs have been added to Appendix M.
4. Discussion on computational overhead for AdaFocal is presented in Appendix K.
5. Experiments on SVHN with ResNet-110 using cross entropy, focal loss $\gamma=3$, FLSD-53, and AdaFocal are aded to Appendix P.11.

---

### Meta-Review · Area_Chair_rQf5 · 2022-08-30

**Recommendation:** Accept
**Confidence:** Less certain

**Metareview:**

Prior work has shown that training with focal loss improves the uncertainly calibration of the models. In this work, authors show that the adoptive choice of the hyper-parameter used in the focal loss would further improve the calibration. Extensive empirical results showed in the paper suggest that the proposed method, AdaFocal, improves over the baselines in several domains/datasets and architectures in terms of calibration error and OOD detection.

The paper is well-written and easy to follow. The proposed method is novel and its relationship with prior work is properly discussed. Furthermore, empirical results show that this method outperforms existing baselines. Therefore, I am recommending acceptance. Given lack of any theoretical insights or motivation for the proposed method, it would certainly help to improve the empirical results particularly by adding results for a few other architectures trained on ImageNet. Furthermore, please add discussions and ablations on the sensitivity of hyper-parameter choices for the camera-ready version.

**Award:**

No

---

### Decision · Program_Chairs · 2022-09-14

Accept